

# Increased West Antarctic and unchanged East Antarctic ice discharge over the last 7 years

Alex S. Gardner[1], Geir Moholdt[2], Ted Scambos[3], Mark Fahnstock[4], Stefan Ligtenberg[5], Michiel van den Broeke[5], and Johan Nilsson[1]

[1]Jet Propulsion Laboratory, California Institute of Technology, Pasadena, CA 91109, USA
[2]Norwegian Polar Institute, Fram Centre, 9296 Tromsø, Norway
[3]National Snow and Ice Data Center (NSIDC), University of Colorado at Boulder, Boulder, CO 80303, USA
[4]Geophysical Institute, University of Alaska Fairbanks, Fairbanks, AK 99775, USA
[5]Institute for Marine and Atmospheric research Utrecht (IMAU), Utrecht University, Utrecht, the Netherlands

**Correspondence:** Alex S. Gardner (alex.s.gardner@jpl.nasa.gov)

**Abstract.**  Ice discharge from large ice sheets plays a direct role in determining rates of sea-level rise. We map present-day Antarctic-wide surface velocities using Landsat 7 and 8 imagery spanning 2013–2015 and compare to earlier estimates derived from synthetic aperture radar, revealing heterogeneous changes in ice flow since $\sim 2008$. The new mapping provides complete coastal and inland coverage of ice velocity north of 82.4° S with a mean error of $< 10\,\mathrm{m\,yr^{-1}}$, resulting from multiple overlapping image pairs acquired during the daylight period. Using an optimized flux gate, ice discharge from Antarctica is $1929 \pm 40$ Gigatons per year ($\mathrm{Gt\,yr^{-1}}$) in 2015, an increase of $36 \pm 15\,\mathrm{Gt\,yr^{-1}}$ from the time of the radar mapping. Flow accelerations across the grounding lines of West Antarctica's Amundsen Sea Embayment, Getz Ice Shelf and Marguerite Bay on the western Antarctic Peninsula, account for 89 % of this increase. In contrast, glaciers draining the East Antarctic Ice Sheet have been remarkably constant over the period of observation. Including modeled rates of snow accumulation and basal melt, the Antarctic ice sheet lost ice at an average rate of $183 \pm 94\,\mathrm{Gt\,yr^{-1}}$ between 2008 and 2015. The modest increase in ice discharge over the past 7 years is contrasted by high rates of ice sheet mass loss and distinct spatial patters of elevation lowering. The West Antarctic Ice Sheet is experiencing high rates of mass loss and displays distinct patterns of elevation lowering that point to a dynamic imbalance. We find modest increase in ice discharge over the past 7 years, which suggests that the recent pattern of mass loss in Antarctica is part of a longer-term phase of enhanced glacier flow initiated in the decades leading up to the first continent-wide radar mapping of ice flow.

## 1 Introduction

The Antarctic ice sheet receives roughly $2000\,\mathrm{Gt}$ ($\sim 5.5\,\mathrm{mm}$ sea-level equivalent) of precipitation each year with $> 90\,\%$ of this mass leaving as solid ice discharge to the ocean and the remaining $< 10\,\%$ leaving in the form of sublimation, wind-driven snow transport, meltwater runoff and basal melt. Recent studies indicate significant mass loss from the Antarctic ice sheet that is likely accelerating (Harig and Simons, 2015; Helm et al., 2014; Martín-Español et al., 2016; McMillan et al., 2014; Rignot et al., 2011b; Shepherd et al., 2012; Velicogna, 2009). Understanding how this imbalance evolves is critical to providing meaningful projections of sea-level change. A major hurdle for improved attribution of mass changes determined from gravimetry and/or altimetry, and in determining mass changes themselves from the mass balance approach, is the difficulty in resolving continent-wide changes in ice discharge at high precision and accuracy for multiple epochs. This requires circum-Antarctic measurements of surface velocity on fine spatial scale and with sufficient accuracy ($\sim 10\,\mathrm{m\,yr^{-1}}$) to observe regionally coherent changes in flow.

**Published by Copernicus Publications on behalf of the European Geosciences Union.**

Earlier circum-Antarctic mappings of surface velocity have been based on synthetic aperture radar (SAR) data with incomplete coverage for 1996–2000 (Jezek et al., 2003; Rignot, 2006) and near-complete coverage for 2007–2009 (Rignot et al., 2011 TS2). Applications of optical imagery for surface velocity mapping have heretofore been limited to more local scales (e.g., Bindschadler and Scambos, 1991; Scambos et al., 1992) due to limited sensor capabilities, cloudiness and too few repeat-image acquisitions. Improvements in sensor technology (particularly in radiometric resolution) and far higher image acquisition rates for Landsat 8, launched in 2013, largely overcome these limitations (Fahnestock et al., 2015; Jeong and Howat, 2015; Mouginot et al., 2017) and provide the ability to generate near-complete yearly mappings of surface velocity with high accuracy ($\sim 10\,\mathrm{m\,yr^{-1}}$).

Here we describe the application of two newly developed and independent feature tracking methodologies (JPL and NSIDC) that we applied to hundreds of thousands of Landsat image pairs covering the entire Antarctic ice sheet north of 82.4° S, producing six near-complete mappings of ice sheet surface velocities in both the 2013–2014 and 2014–2015 austral polar daylight periods. By differencing these velocity fields with the earlier SAR mapping (Rignot et al., 2011a) we resolve changes in ice surface velocity for the 7-year period between circa 2008 and 2015. Velocity changes are then used to estimate ice discharge on the basin scale and its change through time. For the determination of ice discharge we provide a novel approach to defining the cross-sectional area of ice flow (flux gate; Sect. 2.2) that greatly reduces uncertainties in estimates of ice discharge. By differencing estimates of ice discharge and basal melt rates (Van Liefferinge and Pattyn, 2013) from published estimates of the surface mass balance (van Wessem et al., 2016, 2014) we are able to estimate the net mass balance of the ice sheet on the basin scale, revealing recent patters of ice sheet imbalance.

## 2   Methods

### 2.1   Surface velocity

Glacier velocities were determined by feature tracking of matching path-row Landsat Collection 0 L1T and L1GT image pairs in the panchromatic Band 8 (15 m pixel size) using normalized cross correlation (NCC). To assess the sensitivity of our results to choices in Landsat processing methodology (e.g., search template size, spatial resolution, geolocation offset correction, data filtering, image-pair date separation and compositing) we examine multiple velocity mosaics derived from two independent processing methodologies developed by JPL and NSIDC (Fig. 1). Uncertainties in velocities were determined by comparing Landsat and SAR velocities measured at flux-gate nodes for basins with minimal change in ice discharge (B1–19 and B27), i.e., where velocity differences are assumed to be indicative measurement uncertainty.

Uncertainties in velocities can be as high as 20–30 m yr$^{-1}$ locally but are largely uncorrelated on basin scales ($> 1000$ km; see Appendix A for validation of the velocity fields). All velocity mosaics are freely downloadable from the NSIDC. JPL and NSIDC processing chains share many of the same characteristics, with main differences being how the image-pair data are corrected for geolocation errors, how the imagery is searched for matching features and the choice of search parameters such as template size and spacing.

#### 2.1.1   JPL auto-RIFT

**Image-pair pixel offsets**

The autonomous Repeat Image Feature Tracking (auto-RIFT v0.1) processing scheme was applied to all Landsat 7 and 8 images acquired between August 2013 and May 2016 with 80 % cloud cover or less. Images were preprocessed using a 5 by 5 Wallis operator to normalize for local variability in image radiance caused by shadows, topography and sun angle. All image pairs with less than 910-day separation were searched. Preprocessed image pairs were searched for matching features by finding local NCC maxima at subpixel resolution using Taylor refinement (Paragios et al., 2006) within a specified search distance. A sparse (1/16 of full search) NCC search was first used to determine areas of coherent correlation between image pairs. Results from the sparse search guide a dense search with search centers spaced such that there is no overlap between adjacent template search chips (i.e., the distance between template centers is equal to the template size). Highest-quality image pairs ($< 20$ % cloud and $< 1$-year separation) were searched using this approach, with a large search distance centered at zero pixel offset with a 32 by 32 pixel template chip. Spatially resolved statistics (mean and standard deviation of $x$ and $y$ displacements) are then used to guide a dense image search of all imagery with $16 \times 16$ or $32 \times 32$ pixel template chips depending on expected gradients in surface velocities. Areas of unsuccessful retrievals were searched with progressively increasing template chip sizes of 32, 64 and 128 that increase the signal to noise at the expense of spatial resolution.

Successful matches were identified using a novel normalized displacement coherence (NDC) filter. In this approach filtering is applied on search-normalized displacements, i.e., displacements divided by the NCC search distance. Normalized displacements are accepted if 7 or more of the values within a 5 by 5 pixel centered window are within one-quarter of a search distance for both $x$ and $y$ displacement components. This acceptance criterion is iterated on three times. Finally an iterative (two times) filter is applied to remove the few number of displacements that are retained by random agreement with neighbors. For this filter, displacements are compared to the centered 5 by 5 window median. Only values that agree within 4 times the centered 5 by 5 window mean absolute deviation are retained. The NDC filtering ap-

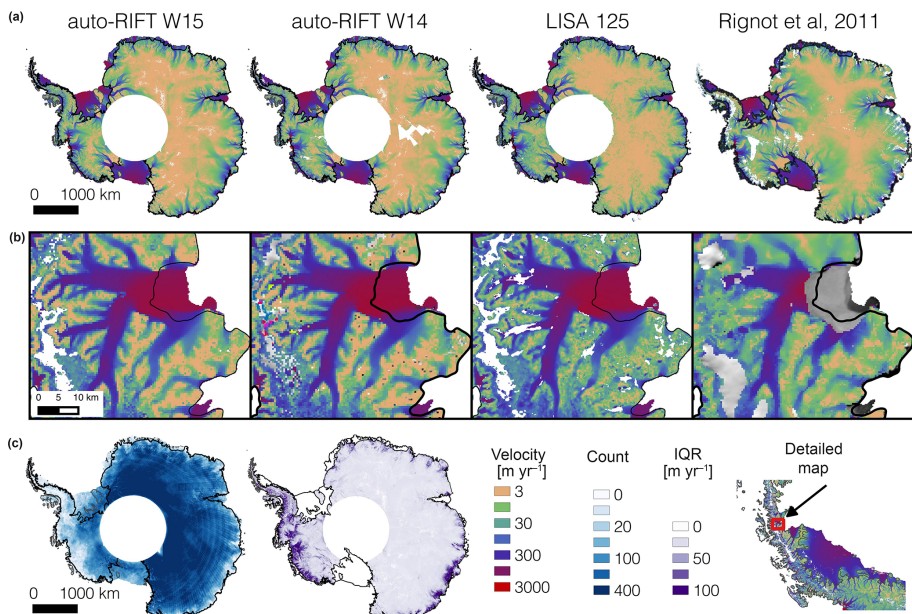

**Figure 1.** Comparison between JPL auto-RIFT weighted average, NSIDC LISA 125 m and Rignot et al. (2011 TS3) surface velocities. Panel **a** shows Antarctic-wide velocities; panel **b** shows close-ups of the Hektoria Glacier, located on the eastern side of the Antarctic Peninsula for spatial detail; and panel **c** shows valid image-pair velocity counts and their interquartile range (IQR) for the auto-RIFT W15 mosaic. Formal errors produced by auto-RIFT are unrealistically low so we display the IQR as a proxy for the per-pixel random error.

proach is highly generic and very effective at removing random image-pair matches but not at removing match blunders that can result in spatially coherent errors. Remaining blunders are filtered during the merging process using information from all image pairs.

Image-pair pixel displacements were calculated from georeferenced images that are in Antarctic Polar Stereographic (EPSG 3031) projection. This introduces scale distortions that increase with distance from the latitude of origin (71° S). We corrected for this scale distortion when converting from pixel displacement to velocity following the equations presented in Snyder (1987).

Image geometry between image pairs is highly stable, but images suffer from large $x$ and $y$ geolocation errors ($\sim 15$ m). This resulted in good gradients in velocity but poor absolute velocity. Displacement fields were also contaminated by match blunders (e.g., matching along shadow edges or of surfaces obscured by cloud in one of the two images). Therefore, displacement fields required heavy post-processing to isolate the geophysical signal. This was done by stacking all time-normalized displacements (velocities), co-registering them over stationary or slow flowing surfaces and filtering based on the interquartile range (IQR) determined for each pixel of the displacement stack. All $x$ and $y$ displacements that fell outside of the range $Q_1 - T \times \mathrm{IQR}$ to $Q_3 + T \times \mathrm{IQR}$ were culled from the data set, where $Q_1$ and $Q_3$ are the first and third quartile, respectively, and $T$ is a scalar that defines the acceptance threshold.

## Reference velocity

A reference velocity ($V x_0$, $V y_0$) field was generated from all individual image-pair velocities. As a first step, gross outliers were removed from the unregistered data by setting $T$ equal to 3. Stacked displacement fields were then coregistered by iteratively correcting for the median $x$ and $y$ velocity difference between individual image-pair velocities and static reference velocity fields ($V x_{\mathrm{ref}}$ and $V y_{\mathrm{ref}}$) over stationary or slow flowing surfaces, stopping after five iterations. For each iteration, coregistered displacements were filtered setting $T$ equal to 1.5, and the effective template chip size (resolution of the velocity field) was coarsened for low-velocity gradients ($< 10$ m yr$^{-1}$ between adjacent search chips) to minimize high-frequency noise while retaining spatial gradients.

Initial $V x_{\mathrm{ref}}$ and $V y_{\mathrm{ref}}$ were defined as all grounded ice pixels with median velocities $< 10$ m yr$^{-1}$ and with $> 100$ valid retrievals. Where these conditions were not met, $V x_{\mathrm{ref}}$ and $V y_{\mathrm{ref}}$ were supplemented with Rignot et al. (2011a) velocities $< 10$ m yr$^{-1}$. Additionally, all pixels containing exposed rock were initially assigned a $V x_{\mathrm{ref}}$ and $V y_{\mathrm{ref}}$ of 0 m yr$^{-1}$. Exposed rock was identified using the SCAR Antarctic Digital Database (Thomson and Cooper, 1993; Fig. 2). The initial template chip size was set to the minimum chip size for which 40 % of the valid displacements in the stack were determined using a chip of that size or smaller. After each coregistration of the data, $V x_{\mathrm{ref}}$ and $V y_{\mathrm{ref}}$ were set equal to the error-weighted velocity for those pixels that have velocities $< 50$ m yr$^{-1}$ and a $V x$ and $V y$

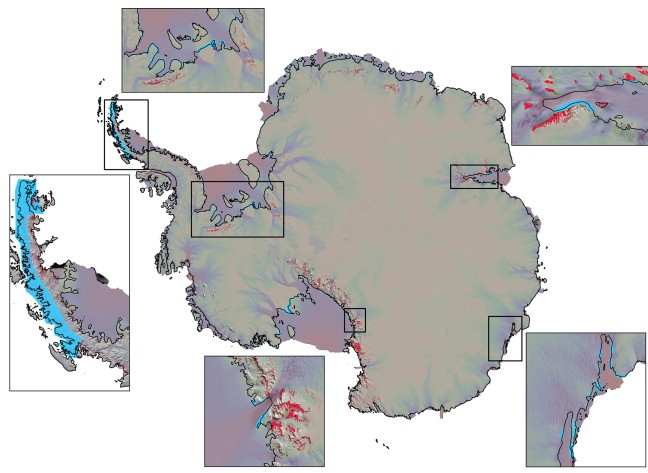

**Figure 2.** Antarctic ice sheet velocities overlain on the MODIS Mosaic of Antarctica (Scambos et al., 2007). Areas of imposed zero change in velocity are shown in cyan. Areas of prescribed zero surface velocity (rock outcrops) are shown in red as defined according to the Antarctic Digital Database (http://www.add.scar.org).

$IQR < 40\,m\,yr^{-1}$. All pixels containing exposed rock are reassigned a $Vx_{ref}$ and $Vy_{ref}$ of $0\,m\,yr^{-1}$. The uncertainty of each image-pair velocity field was determined as the standard deviation of the residuals to $Vx_{ref}$ and $Vy_{ref}$. When there were fewer than 320 coregistration pixels within an image pair, the uncertainty was set to the RSS of the pointing uncertainty of each image.

**JPL auto-RIFT annual fields**

All image-pair velocities for a given year $Y$ (center date of image pair $> 15$ July, $Y - 1$ and $< 15$ July $Y$) were coregistered using the reference velocity field ($Vx_0$, $Vy_0$), where $Vx_{ref}$ and $Vy_{ref}$ were set equal to the error-weighted velocity ($Vx_0$, $Vy_0$) for those pixels that have velocities $< 50\,m\,yr^{-1}$ and $Vx_0$ and $Vy_0$ $IQR < 40\,m\,yr^{-1}$. Annual error-weighted averages and median velocities were first calculated setting the filter limits based on the quartile ranges of $Vx_0$ and $Vy_0$ and setting $T = 3$. Velocities were further refined by setting the filter limits based on the quartile ranges of initial annual values and using a more stringent acceptance threshold of $T = 1.5$.

Using this approach we calculated four nearly complete Landsat 8 velocity maps: median (M) and error-weighted average (W) velocities for years 2014 and 2015. The 2014 and 2015 velocities were derived from $\sim 100\,000$ and $\sim 200\,000$ unique image pairs, respectively (Fig. 1).

### 2.1.2   NSIDC LISA

NSIDC's Landsat ice speed for Antarctica processing (LISA v1.0) used the Python image correlation, PyCorr v1.10, described in detail by Fahnestock et al. (2015). PyCorr

was applied to Landsat 8 data separated by 16 to 400 days, spanning 26 September 2013 to 1 April 2015 using a reference template size of $300 \times 300\,m$ with $300\,m$ spacing between search templates. Images were manually selecting based on the proportion of cloud-free surface coverage from the group of images with less than 70 % cloud cover. A high-pass filter of approximately $250\,m$ spatial scale was applied to the images to enhance surface detail and suppress topographic shading.

PyCorr outputs a quality metric delcorr, which is the difference between the regression coefficient of the peak match and the second-highest match outside of a $3 \times 3$ cell area around the peak. All displacement values with a delcorr value less than 0.15 were eliminated. Velocities are further filtered by examining the difference between the velocities at the assessed pixel with the eight surrounding values. Velocities with no neighbors were masked. Velocities with one neighbor were masked when the absolute difference between the two values was greater than $365\,m\,yr^{-1}$. Velocities with two neighbors were masked if they exceeded 3 standard deviations of the mean. Finally the standard deviation of each $3 \times 3$ region was computed, and the center pixel of each region was masked when the corresponding standard deviation is greater than $365\,m\,yr^{-1}$.

Image-pair geolocation errors were corrected using three sets of $x-y$ velocity offsets. Each set of offsets were computed over rock (http://www.add.scar.org) and near-zero ice ($< 20\,m\,yr^{-1}$) and low ice velocity ($< 40$ and $> 20\,m\,yr^{-1}$) areas according to Rignot et al. (2011a). Offset corrections were then weighted by count and applied to individual image-pair results.

Resulting velocities for each image pair were bilinearly resampled to the target grid spacing of either 750 or 125 m. These grids were then composited using a weighting scheme that favors the more accurate long-interval velocity determinations (16-day pairs, 0.3 weighting; 32-day pairs, 0.6; 48-day pairs, 0.9; $> 48$-day pairs, 1.0). Additionally, a weighting factor was applied to each cell based on the mean NCC and delcorr values. Mosaics were then corrected for projection scale distortion. The velocity grids were then stacked and combined in a weighted average scheme. The number of image pairs in the LISA v1.0 grid ranges from $\sim 10$ to over 200 (Fig. 1).

### 2.2   Flux gates

Estimation of ice flux from measurements of surface velocity requires knowledge of the vertical density profile, flow cross-sectional area (flux gate) and an assumption of the relationship between surface and depth-averaged velocity. The most accurate estimates of ice thickness come from radio-echo-sounding (RES) measurements, but RES data only exist for about 19 % of the ice sheet grounding line. For the calculation of discharge, we choose to compromise proximity to the grounding line for inclusion of more upstream RES data

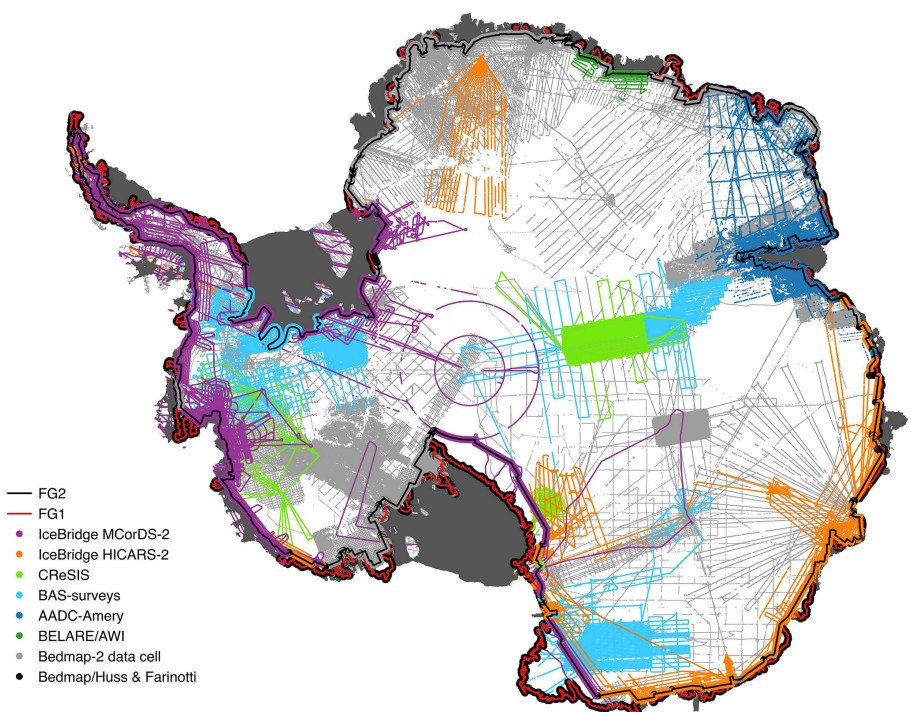

**Figure 3.** Radio-echo-sounding data used to compile flux gates FG1 and FG2. An overview of the use and of each data set and their references is provided in Table 1.

**Table 1.** Data sources and percentages for radio-echo-sounding data used to compile flux gates.

| Data set | GL0 | FGI | FG2 | Reference |
|---|---|---|---|---|
| IceBridge MCoRDS-2 | 5.3 % | 16.1 % | 31.5 % | Leuschen et al. (2010) |
| IceBridge HiCARS-2 | 2.1 % | 5.4 % | 20.2 % | Blankenship et al. (2012) |
| CReSIS | 0.4 % | 0.3 % | 0.3 % | Gogineni (2012) |
| BAS surveys | 1.7 % | 7.4 % | 9.0 % | https://legacy.bas.ac.uk/data/aerogeo/ |
| AADC Amery | 2.6 % | 2.5 % | 12.1 % | Allison and Hylland (2010) |
| BELARE/AWI | 0.1 % | 2.4 % | 4.1 % | Callens et al. (2014, 2015) |
| Bedmap-2 data cell | 6.9 % | 8.3 % | 18.3 % | Fretwell et al. (2013) |
| Sum | 19.2 % | 42.4 % | 95.6 % | |

and for avoiding glacier shear zones with poorly constrained velocities. We do so by modifying the best-known grounding line to go inland of major shear zones and to follow nearby RES flight lines from which valid ice thickness data can be extracted. We prioritize the nearest and most recent RES data available from seven freely available data sets (Fig. 3 and Table 1). For flux gates with no RES data within 1 km distance, ice thickness values are extracted by bilinear interpolation from the ice thickness grid of Huss and Farinotti (2014) over the Antarctic Peninsula and Bedmap-2 (Fretwell et al., 2013) for the rest of Antarctica. We generate three alternative flux gates: a grounding-line flux gate (GL0) based on a synthesis of mappings of the grounding line, ~~a grounded ice set of flux gates near upstream of the grounding line improved by following RES profiles (FG1) and a flux-gate outline based solely on RES profiles in favorable positions (FG2).~~

GL0 is a best-assessment grounding-line position from a synthesis of incomplete data first presented in Depoorter et al. (2013) that has been updated here by more recent grounding-line mappings in the Amundsen Sea region (Rignot et al., 2014, 2011b) and for the Totten Glacier in East Antarctica (Li et al., 2015; Rignot et al., 2013); two highly dynamic regions with considerable ice fluxes and changes in grounding-line position. Ice thickness was mainly extracted from the gridded products of Bedmap-2 (67 %) and the Antarctic Peninsula (9 %), but also a considerable amount of RES data that were within 1 km (applied threshold) of the grounding line (19 %). For that, we also considered grid cells in Bedmap-2 that have been derived directly from RES data

(7 %), as indicated in a data coverage mask. These thickness values have a much lower uncertainty (mean 68 m) than the interpolated thicknesses in areas not covered by RES (mean 168 m).

FG1 is a modified version of GL0 that follows RES flight lines (Fig. 3) or Bedmap-2 data cells that are in the vicinity of the grounding line. Whether or not to divert from the grounding line in favor of RES profiles was determined ad hoc rather than applying a strict distance threshold. Long, continuous RES profiles further apart were more likely to be followed than short, scattered RES data closer to the grounding line. In general, the modified parts of FG1 are within a few tens of kilometers from the GL0 and even less so in the Amundsen and Bellingshausen Sea coasts and the Filchner-Ronne ice shelf regions, where RES flight lines are often aligned with the grounding line. Almost all of these important regions are covered by RES data in FG1, and for Antarctica as a whole the RES coverage is 42 % (Table 1). We found that FG1 was the most suitable flux-gate line for estimating changes in ice discharge due to its close proximity to the grounding line and high coverage of RES data.

FG2 is a ~~further~~ modified version of FG1 that further prioritizes RES flight lines over proximity to the grounding line around the entire continent. Only slight modifications were made in regions like the Amundsen and Bellingshausen Sea coasts, the Filchner-Ronne ice shelf and Dronning Maud Land for which many near-grounding-line RES data exist, but for parts of East Antarctica and along the Transantarctic Mountains the modification can be several hundred kilometers (Fig. 3). The total coverage of RES data along FG2 is 96 % (Table 1). We used this flux-gate line to estimate absolute discharge for the ice sheet, but not for assessing temporal changes in discharge, because they are often most pronounced near the grounding line that is better sampled by FG1.

The average point spacing along the three flux-gate lines is 198–265 m, with a maximum spacing of 400 m to ensure sufficiently dense sampling of ice thickness and surface velocity for ice flux calculations (see Appendix A for a detailed discussion of resolution-dependent errors in flux calculations). Flux-gate points without RES data and within the rock mask of the SCAR Antarctic Digital Database (< 4 %; Thomson and Cooper, 1993; Fig. 2) were assigned a zero ice thickness. Since the thickness data were provided as physical ice thicknesses, we subtracted modeled average (1979–2015) firn air content (FAC; see Sect. 2.5) to obtain ice-equivalent thicknesses, assuming ice has a density of 917 kg m$^{-3}$, relevant for ice flux calculations.

For further analyses, we also extracted point attributes for source data and year, surface elevation, FAC and all available thickness data. Histograms of ice thickness, uncertainties in ice thickness, date of thickness measurement, FAC, uncertainty in FAC, surface velocity, ice thickness change rate and uncertainty ice thickness change rate for all three flux gates

are shown in Fig. D. Flux gates and extracted ancillary data are provided as a Supplement TS5.

## 2.3 Ice discharge

We calculate ice flux ($F$) by multiplying the $x$ and $y$ velocity component ($Vx/y$) by the width of the flux gate projected in the $x$ and $y$ coordinates ($Wx/y$) and ice-equivalent thickness ($H$) at each flux node ($i$) and summing

$$F = \sum_{i=1}^{nn} (Vx_i Wx_i + Vy_i Wy_i) H_i, \qquad (1)$$

where nn is the number of nodes at which ice flux is calculated. Here we defined the flux gate following polygon convention with the upstream side of the flux gate being defined as to the right-hand side of the polygon gate vector as one moves from node $n$ to node $n+1$. In this convention $Wx$ is negative when $y_{n+1} > y_n$ and $Wy$ is negative when $x_{n+1} < x_n$. Ice discharge ($D$) at the grounding line of the ice sheet corresponds to $F$ for the GL0 flux gate. Applying mass conserving principles (Morlighem et al., 2011), $D$ is equal to $F + \text{SMB} + dV_{\text{dyn}}/dt$ for the FG1 and FG2 flux gates. SMB is the unmeasured flux due to a positive surface mass budget of the area between the flux gate and the grounding line and is estimated from RACMO2.3 climatology (1979–2015; see Sect. 2.4). SMB is corrected (reduced) for basal melt occurring between the flux gate and the grounding line which does not contribute to solid ice discharge (Van Liefferinge and Pattyn, 2013). $dV_{\text{dyn}}/dt$ is the unmeasured flux due to ice flow convergence and divergence between the flux gate and the grounding line, which we refer to as the dynamic volume change. This is accounted for by assuming that firn corrected CryoSat-2 elevation change rates (Sect. 2.6) measured over ice moving at $> 200$ m yr$^{-1}$ that lies between the flux-gate and the grounding line can be attributed to dynamic volume change. Rates of volume change in 2008 and 2015 were extrapolated using the measured acceleration in the rate of elevation change over the period of CryoSat-2 data (2011–2015). Measured dynamic volume loss is considered to increase total discharge and vice versa. Uncertainty in the dynamic volume change can not be rigorously quantified and are therefore conservatively assumed to be 0.1 m yr$^{-1}$ times the area between the grounding line and the flux gate having a surface velocity $> 200$ m yr$^{-1}$ or 30 % of the magnitude of the estimated dynamic volume change, whichever is larger. A velocity cutoff of 200 m yr$^{-1}$ was selected to separate volume changes resulting from changes surface mass balance and those resulting from changes in dynamics. This threshold is arbitrary. Even so, the dynamic volume change correction is very small and insensitive to the selected cutoff velocity.

Calculation of discharge is highly sensitive to the definition of the flux gate and to any vertical gradient in the ice flow (Chuter et al., 2017; Mouginot et al., 2014; Rignot, 2006; Rignot and Thomas, 2002). When calculating ice

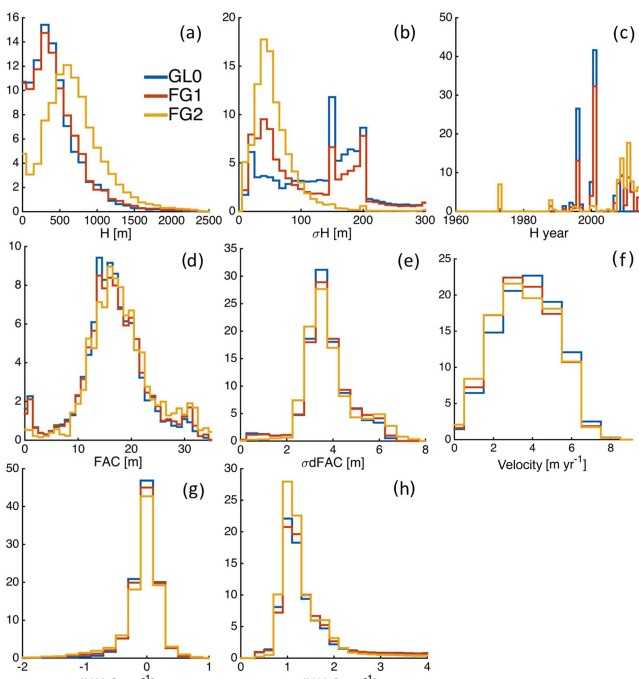

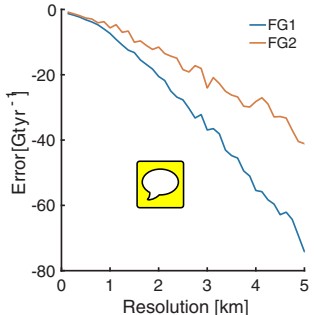

**Figure 5.** Error in total Antarctic discharge (relative to best estimate) when velocity and ice thickness are averaged for increasing along-flux-gate resolutions prior to computing flux.

**Figure 4.** Histograms of ice-equivalent thickness (**a**), uncertainty in ice-equivalent thickness (**b**), year of ice thickness measurement (**c**), firn air content (**d**), uncertainty in firn air content (**e**), surface velocity (**f**), change rate of ice-equivalent thickness (**g**) and uncertainty in change rate of ice-equivalent thickness (**h**) for GL0, FG1 and FG2 flux gates. The $y$ axis is the percentage of flux nodes that fall within each histogram bin.

flux, we assume that there are no vertical gradients in ice velocity. This assumption introduces a small positive bias ($< 0.4\%$) but is negligible relative to other sources of error. See Appendix A for the calculation of the expected vertical gradient in ice velocity. One known issue is the systematic underestimation of ice flux with the coarsening of the resolution of the basal topography and/or the surface velocity (Fig. 5). This happens because fast-moving ice is concentrated in basal troughs: higher velocities multiplied by larger ice thickness and lower velocities multiplied by smaller ice thickness do not equate to average thickness multiplied by average velocity. FG2, which follows high-resolution RES profiles around almost the entire continent at the expense of proximity to the grounding line, provides the cross-sectional area with the lowest uncertainty and is most appropriate for estimating the total discharge, even after having to account for additional mass input between the gate and the grounding line. FG1 strikes a balance between proximity to the grounding line (GL0) and the distance from ice thickness observations. This gate is best suited for estimating changes in ice discharge. Our best estimate of total discharge is computed using the 2015 error-weighted average auto-RIFT velocities, FG2 and ~~estimated~~ additional mass flux between FG2 and GL0. We then compute the change in discharge between the

2015 and 2008 period at FG1 and subtract this from our best estimate of total discharge, accounting for dynamic volume change and changes in ice thickness between periods. This multi-flux-gate approach greatly reduces errors in estimates of ice discharge.

For areas south of the Landsat observation limit, we first calculate the total flux across gates located $> 82.4°$ S using the 1997 and 2009 SAR velocity mappings of Scheuchl et al. (2012). To determine a representative 2015 flux rate we extrapolate the 2009 estimate assuming the same rate of change in discharge as observed for the 1997–2009 period.

Changes in flux ($dF$) were calculated at all flux-gate nodes ($i$) where both velocity mappings were valid and assumed to be unchanged elsewhere. In our analysis of velocities we found that there were some geocoding issues between the SAR (Rignot et al., 2011 TS6) and Landsat velocities, which are most likely due to errors in the elevation model used to convert from radar slant range coordinates to a location on the Earth surface. We also found the SAR velocities unreliable for most of the northwest Antarctic Peninsula, where velocities near the grounding lines of narrow outlet glaciers were unrealistically low and likely the results of interpolation to areas of missing data. To minimize the impact of these artifacts in our flux-change analyses, we prescribed areas of zero change in flux along shear margins where changes are expected to be small and for much of the northwest Antarctic Peninsula (Fig. 2). Any residual geocoding errors are expected to introduce noise into our analysis but are unlikely to significantly bias our estimates of flux or flux change as errors will somewhat cancel when integrated across the entire glacier cross section (errors are typically of similar magnitude but opposite sign along right and left flow margins). See Appendix A for a comprehensive discussion of the uncertainty quantification.

One known limitation of our analysis is that the SAR velocity mosaic (Rignot et al., 2011a) that we difference our Landsat velocities with is derived from data spanning the period 1996–2009 with no information provided on the effective date of the data. We assume that the SAR mosaic

has a representative date of circa 2008 as most data used in the mosaic was collected between 2007 and 2009. This data has been used previously to estimate total Antarctic discharge in Rignot et al. (2013) with a reference date of 2007 to 2008 and in Depoorter et al. (2013) with a reference date of 2007 to 2009. Individual year composites of the data used in older mosaic were recently made available (Mouginot et al., 2017). These new data come with more precise time stamps but at the expense of reduced horizontal resolution (1 km vs. 450 m), reduced spatial coverage and larger uncertainties. To ensure that our stated time period of circa 2008 is appropriate we resample (linear interpolation) the original SAR velocity mosaic to 1 km and compare to the error averaged 2007–2008 TS7 and 2008–2009 velocities from the new data set. Differences in flux across the FG1 are less than $2\,\mathrm{Gt\,yr^{-1}}$ for all basins except for basins 12, 13, 14 and 24 that differ by $-4$, $-5$, $-6$ and $4\,\mathrm{Gt\,yr^{-1}}$, respectively. Some of the difference can be attributed to real differences in flow but also from differences in uncertainties between products (the original SAR mosaic having lower errors, particularly for the East Antarctic) and from differences in horizontal resolution. From this analysis we concluded that the best estimate of flux for the $\sim 2008$ period is produced by the earlier mosaic that has higher spatial resolution and the lower uncertainty, which is derived from the same underlying data contained in the annual mosaics. We also determine the period "circa 2008" characterizes well the effective date of the earlier ~~MEASURES~~ mosaic.

## 2.4 Surface mass budget

Here we estimate SMB for the 2008–2015 period from Regional Atmospheric Climate Model version 2.3 (RACMO2.3) output at a horizontal resolution of 5.5 km for the Antarctic Peninsula (van Wessem et al., 2016) and 27 km elsewhere (van Wessem et al., 2014). In RACMO2.3, SMB is calculated as the total precipitation (from snow and rain) minus total sublimation (directly from the surface and from drifting snow), wind-driven snow erosion and meltwater runoff. For the six Antarctic Peninsula basins (B1, B23–27), entirely or partially covered by the high-resolution model, we use the 27 km model output for the missing years of 2014 and 2015. For these basins, the 27 km model output was scaled to better agree with the 5.5 km output using the delta scaling approach. Uncertainty in SMB is taken to be 20 % and is treated as uncorrelated between basins. The reader is referred to the works of van Wessem et al. TC (2014 and 2016) for a thorough discussion of the model setup, model validation and SMB uncertainties.

## 2.5 Firn air content

To convert volume fluxes to mass fluxes, the depth-averaged ice-sheet density is needed. FAC is a measure of the residual column that would remain if the firn column were compressed to the density of glacier ice, assumed to be $917\,\mathrm{kg\,m^{-3}}$. We estimate FAC using the firn densification model IMAU-FDM (Ligtenberg et al., 2011, 2014). IMAU-FDM simulates firn densification by dry compaction and through meltwater processes (percolation, retention and refreezing) and is forced at the surface by 3-hourly resolution output of RACMO2.3 (van Wessem et al., 2016, 2014): surface temperature, 10 m wind speed, precipitation (solid and liquid), sublimation, wind-driven snow erosion/deposition and surface melt. The simulation over the entire Antarctic continent (at 27 km grid resolution) covers 1979–2015, while the Antarctic Peninsula simulation (at 5.5 km grid resolution) only covers 1979–2013. Both simulations output FAC at 2-day temporal resolution. The IMAU-FDM is calibrated using 48 depth–density observations from across Antarctica (Ligtenberg et al., 2011), and results have been successfully used to convert satellite altimetry (e.g., Gardner et al., 2013; Scambos et al., 2014; Shepherd et al., 2012) and ice thickness measurements (e.g., Depoorter et al., 2013; Fretwell et al., 2013) into estimates of ice mass change and ice-equivalent thickness. Although time-evolving FAC is simulated throughout 1979–2015, we use the climatological average FAC as the most robust correction of our flux-gate thicknesses that are based on source data from many different times, sometimes unknown.

Uncertainties in the simulated FAC originate from either the observations used in the IMAU-FDM calibration process or the RACMO2.3 forcing data. This has been quantified at 10 % (Supplement of Depoorter et al., 2013), composed of measurements errors in the observations of the pinning points in a depth–density profile: surface density, depth of $550\,\mathrm{kg\,m^{-3}}$ level and depth of $830\,\mathrm{kg\,m^{-3}}$ level. The RACMO2.3 uncertainty is primarily caused by the assumption used for model initialization; to initialize the IMAU-FDM, it is assumed that the climate over the past 100–1000 years was equal to the 1979–2013/15 average climate (Ligtenberg et al., 2011). Therefore, errors in the climatic forcing during the initialization period have a direct effect on the simulated firn density profile and subsequent FAC. Using sensitivity simulations, it was found that a 1 % perturbation in accumulation during the initialization period causes a 0.75 % error in FAC. Similarly, a 1 % perturbation in the melt / accumulation ratio results in a 0.27 m error in FAC. The melt / accumulation ratio was used instead of the total melt, as the amount of annual snow that melts away in summer (i.e., the ratio between annual melt and annual accumulation) mainly determines how much firn pore space remains rather than the total amount of melt.

Along the ice-sheet grounding line the mean and standard deviation of FAC are $16.3 \pm 6.1\,\mathrm{m}$ with associated uncertainties of $3.7 \pm 1.0\,\mathrm{m}$. The combined uncertainties of the firn observations and the RACMO2.3 forcing of accumulation and surface melt showed the highest uncertainties on the western side of the Antarctic Peninsula, where high accumulation is combined with high melt. In areas where the modeled FAC

uncertainty was higher than the actual FAC, the uncertainty was re-set to the same value as the FAC.

## 2.6 Surface elevation and elevation change

To account for thickness changes between the times of discharge calculation (2008 and 2015) and to correct for dynamic volume change between the flux gate and the grounding line, we use surface elevation rates estimated from CryoSat-2 radar altimetry between January 2011 and January 2015 (Fig. 6). CryoSat-2 elevations were derived from the ESA L1c product using the methodology by Nilsson et al. (2016) ~~for the time period of January 2011 to January 2015 over the Antarctic ice sheet~~. For each CryoSat-2 observation mode (LARM and SARIn), the derived surface elevations were separated into grounded and floating ice using the grounded and floating ice definitions from Depoorter et al. (2013) gridded to a 240 m in stereographic (EPSG: 3031) projection. Geophysical range corrections were applied to all data according to Bouzinac (2015). For floating ice, the tidal corrections (ocean tide and ocean loading) were replaced with values generated from the CATS2008 tidal model (Padman et al., 2008).

Surface elevation changes and rates of acceleration were generated using the surface fit method, described in Nilsson et al. (2016), onto a 1 km polar-stereographic grid (EPSG: 3031) for each mode. The derived elevation change distribution was edited to remove solutions with a magnitude larger than $\pm 15\,\mathrm{m\,yr^{-1}}$, similar to the approach taken by Wouters et al. (2015). The edited data was then interpolated onto a 1 km grid using the weighted average of the 16 closest grid points, weighted by their standard error from the least-squares solution and distance. The standard error of the rate of change is assumed to be indicative of the formal error of each measurement. No correction for potential trends in FAC and glacial isostatic adjustment are applied, which may cause surface elevation rates to deviate from ice-equivalent thickness rates.

## 2.7 Mass budget

To assess the net ice sheet mass budget during the 2008–2015 period, we combine our new estimates of discharge (Sect. 2.3) with estimates of surface mass budget (Sect. 2.4) and basal melt rates (Pattyn, 2010; Van Liefferinge and Pattyn, 2013). Discharge and surface mass budget for the northern Antarctic Peninsula (B25–26) are highly uncertain and only included for reference in Table 2. The complex basal topography, narrow glacial valleys and highly crevassed ice, make interpretation of the bed reflection in radar data difficult in this region. Estimating the surface mass budget is equally challenging with large interannual variability and steep spatial gradients in both precipitation and melt due to extreme surface topography over a large latitudinal range. For B25–26, we therefore rely on net mass budgets determined from glacier elevation changes within the 2003–2011 period that

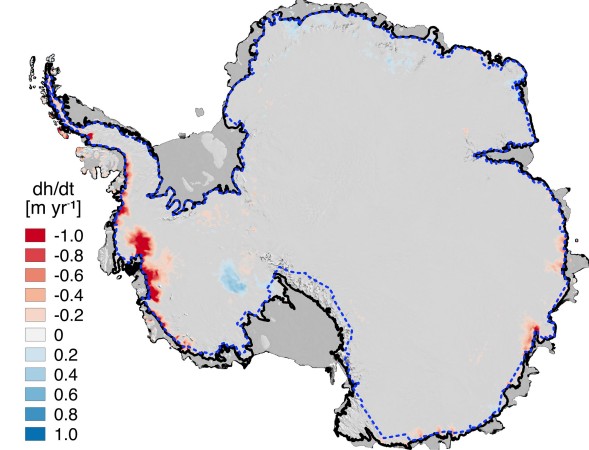

**Figure 6.** Surface elevation change for the period 2011 to 2015. Flux gate FG2 shown with blue dashed line and GL0 shown with heavy black line.

we update with estimated discharge changes for 2008–2015 (Scambos et al., 2014; Berthier et al., 2012 TS12; Shuman et al., 2011 TS13). A full discussion of the updated Antarctic Peninsula mass budget estimate is provided in Appendix B.

## 3 Results

### 3.1 Changes in surface velocity and ice discharge

By combining uncertainties of ice velocity and its relation to depth-averaged velocity, ice thickness, dynamic volume change and SMB for each flux-gate configuration, we estimate a total discharge uncertainty of 5.6 % for GL0, 4.5 % for FG1 and 2.1 % for FG2. The lower uncertainty for FG2 is due to the extensive use of RES data for ice thickness along the flux gate (Fig. 4). Hence, we use FG2 in combination with the Landsat velocity field to estimate total discharge. Obtaining continent-wide discharge for $\sim 2008$ using the SAR-based velocity field (Rignot et al., 2011a) at the FG2 flux gate is not possible due to data gaps inland of the grounding line. Instead, we estimate discharge change between the 2008 and Landsat mappings at FG1 and then subtract that from the Landsat estimate of discharge to obtain a total estimate for 2008. This approach reduces the impact of ice thickness errors at FG1 since they get scaled by velocity differences rather than by velocity magnitudes that are typically much larger. Thickness changes at FG1 and changes in the rate of dynamic volume change between FG1 and the grounding line 2008 and Landsat mappings were accounted for in the estimates of discharge change using the derived CryoSat-2 elevation change rates for 2011–2015 (see Sect. 2.6). Rates of volume change in 2008 and 2015 were extrapolated using the measured acceleration over the 2011–2015 period. Calculating flux in this way reduced the uncertainty in the total

**Table 2.** Surface area, cross sectional area for flux gate FG2, discharge corrected for dynamic volume change and surface mass balance between flux gate FG2 and the grounding line, basal melting, surface mass balance (SMB) and net mass balance for the 27 basins defined by Zwally et al. (2002). Cumulative numbers are provided for the East Antarctic Ice Sheet (EAIS: B2–17), the West Antarctic Ice Sheet (WAIS: B1, B18–23) and the Antarctic Peninsula (AP: B24–B27). Basal melt rates are from Van Liefferinge and Pattyn (2013) and calculated according to Pattyn (2010). SMB is calculated using the RACMO2.3 regional climate model at 5.5 km (van Wessem et al., 2016) resolution over the Antarctic Peninsula and 27 km elsewhere (van Wessem et al., 2014) and averaged over the 2008–2015 period. The net mass balance is calculated as the 2008–2015 SMB minus the average rate of discharge minus basal melt. ~~Net mass balance for the northern Antarctic Peninsula (basins 25 and 26) is not determined using calculated discharge and SMB because of large and poorly constrained uncertainties in ice thickness and modeled SMB. Instead the net mass balance for basins 25 and 26 are determined by updating the mass balance estimate of Scambos et al. (2014) with changes in discharge determined here (see Appendix B).~~ Discharge for 2008 is derived from Rignot et al. (2011 [TS9]) and for 2015 from the mean of the JPL 2015 error-weighted Landsat 8-velocity mapping. [TS10] [11]

| Area | Flux gate | Discharge (Gt yr⁻¹) | | | Basal melt | SMB (Gt yr⁻¹) | Net mass change | |
|---|---|---|---|---|---|---|---|---|
| km² | km² | 2008 | 2015 | Δ | Gt yr⁻¹ | 2008–2015 | Gt yr⁻¹ | kg m⁻² yr⁻¹ |
| 474 800 | 987 ± 53 | 110 ± 8 | 112 ± 7 | 2 ± 3 | 3 ± 0 | 121 ± 24 | 7 ± 25 | 16 ± 54 |
| 765 400 | 305 ± 33 | 48 ± 6 | 47 ± 4 | −1 ± 4 | 3 ± 1 | 52 ± 10 | 2 ± 12 | 2 ± 16 |
| 1 556 600 | 213 ± 18 | 59 ± 4 | 60 ± 4 | 1 ± 2 | 5 ± 2 | 74 ± 15 | 9 ± 15 | 6 ± 10 |
| 241 200 | 351 ± 55 | 41 ± 8 | 43 ± 7 | 2 ± 3 | 1 ± 0 | 45 ± 9 | 2 ± 12 | 8 ± 50 |
| 185 300 | 196 ± 30 | 30 ± 5 | 31 ± 4 | 1 ± 2 | 1 ± 0 | 36 ± 7 | 5 ± 9 | 26 ± 47 |
| 607 700 | 501 ± 59 | 60 ± 7 | 60 ± 6 | −1 ± 3 | 3 ± 0 | 81 ± 16 | 17 ± 17 | 28 ± 29 |
| 492 500 | 495 ± 62 | 68 ± 8 | 70 ± 8 | 2 ± 2 | 2 ± 0 | 93 ± 19 | 23 ± 20 | 46 ± 41 |
| 161 200 | 277 ± 32 | 17 ± 4 | 18 ± 3 | 1 ± 2 | 1 ± 0 | 36 ± 7 | 18 ± 8 | 111 ± 50 |
| 146 000 | 219 ± 18 | 17 ± 3 | 16 ± 2 | −1 ± 2 | 1 ± 0 | 17 ± 3 | 0 ± 5 | −1 ± 31 |
| 919 300 | 55 ± 5 | 34 ± 4 | 33 ± 3 | −1 ± 2 | 3 ± 1 | 42 ± 8 | 6 ± 9 | 6 ± 10 |
| 255 200 | 187 ± 14 | 13 ± 3 | 12 ± 2 | −1 ± 2 | 1 ± 1 | 16 ± 3 | 1 ± 4 | 6 ± 17 |
| 727 100 | 610 ± 74 | 102 ± 11 | 101 ± 10 | 0 ± 3 | 5 ± 1 | 128 ± 26 | 21 ± 28 | 29 ± 38 |
| 1 130 800 | 667 ± 50 | 226 ± 19 | 223 ± 18 | −2 ± 5 | 7 ± 1 | 201 ± 40 | −31 ± 45 | −27 ± 39 |
| 718 500 | 714 ± 48 | 130 ± 10 | 130 ± 10 | 0 ± 3 | 5 ± 1 | 125 ± 25 | −10 ± 27 | −14 ± 38 |
| 123 800 | 190 ± 11 | 26 ± 6 | 26 ± 5 | 0 ± 2 | 1 ± 0 | 25 ± 5 | −2 ± 8 | −16 ± 62 |
| 262 000 | 159 ± 13 | 13 ± 2 | 14 ± 2 | 0 ± 2 | 1 ± 0 | 10 ± 2 | −5 ± 3 | −18 ± 12 |
| 1 825 800 | 646 ± 51 | 67 ± 8 | 67 ± 7 | −1 ± 3 | 5 ± 2 | 78 ± 16 | 5 ± 18 | 3 ± 10 |
| 261 400 | 125 ± 16 | 9 ± 3 | 8 ± 2 | −1 ± 2 | 2 ± 1 | 23 ± 5 | 13 ± 5 | 49 ± 21 |
| 367 700 | 258 ± 34 | 44 ± 6 | 45 ± 6 | 1 ± 2 | 3 ± 1 | 37 ± 7 | −11 ± 10 | −30 ± 26 |
| 180 100 | 490 ± 54 | 171 ± 15 | 183 ± 14 | 12 ± 4 | 2 ± 0 | 112 ± 22 | −67 ± 27 | −375 ± 149 |
| 207 500 | 179 ± 12 | 180 ± 12 | 189 ± 12 | 9 ± 4 | 2 ± 1 | 98 ± 20 | −89 ± 23 | −428 ± 111 |
| 210 200 | 112 ± 7 | 127 ± 8 | 134 ± 8 | 7 ± 2 | 2 ± 0 | 84 ± 17 | −49 ± 19 | −231 ± 89 |
| 74 600 | 249 ± 20 | 83 ± 8 | 83 ± 7 | 0 ± 3 | 1 ± 0 | 65 ± 13 | −18 ± 15 | −242 ± 204 |
| 1?0 600 | 211 ± 15 | 94 ± 7 | 95 ± 7 | 2 ± 3 | 1 ± 0 | 86 ± 17 | −9 ± 19 | −94 ± 186 |
| *2? 700* | *78 ± 15* | *88 ± 13* | *91 ± 12* | *4 ± 5* | *0 ± 0* | *100 ± 20* | *−10 ± 21* | *−297 ± 605* |
| *42 000* | *116 ± 12* | *23 ± 4* | *25 ± 3* | *2 ± 2* | *1 ± 0* | *29 ± 6* | *−17 ± 7* | *−406 ± 174* |
| 52 000 | 89 ± 9 | 12 ± 3 | 12 ± 2 | 0 ± 2 | 0 ± 0 | 18 ± 4 | 6 ± 5 | 120 ± 88 |
| 10 118 500 | 5786 ± 165 | 952 ± 31 | 952 ± 29 | −1 ± 11 | 45 ± 4 | 1058 ± 66 | 61 ± 73 | 6 ± 7 |
| 1 776 200 | 2400 ± 88 | 724 ± 24 | 754 ± 23 | 30 ± 8 | 16 ± 1 | 541 ± 45 | −214 ± 51 | −120 ± 29 |
| *229 200* | *493 ± 26* | *217 ± 15* | *223 ± 14* | *7 ± 6* | *2 ± 0* | *234 ± 27* | *−31 ± 29* | *−133 ± 128* |
| 12 123 900 | 8679 ± 189 | 1894 ± 43 | 1929 ± 40 | 36 ± 15 | 63 ± 4 | 1834 ± 84 | −183 ± 94 | −15 ± 8 |

flux estimate generated from SAR velocities from 99 Gt yr⁻¹ when calculating total discharge only at the grounding line to 40 Gt yr⁻¹, a 60 % reduction in uncertainty, when applying this combined approach .

Comparing differences in discharge estimates between 6 Landsat velocity mappings (Fig. 7, 4 auto-RIFT v0.1, 2 LISA v1.0) shows good agreement despite differences in feature tracking methodologies, template chip size, horizontal resolution and time periods. The standard deviations be-tween flux-change estimates are below the stated uncertainty in discharge listed in Table 2 for all 27 basins. Differences that do exist can be attributed to product errors. Auto-RIFT W15 has the lowest uncertainties, followed by auto-RIFT M15 then auto-RIFT W14 and M14 with the LISA 125 and 750 m products having the highest uncertainties (See Fig. A1). auto-RIFT uncertainties are lowest for the 2015 mapping simply due to a much larger number of available image pairs. The reason for higher uncertainties of the LISA

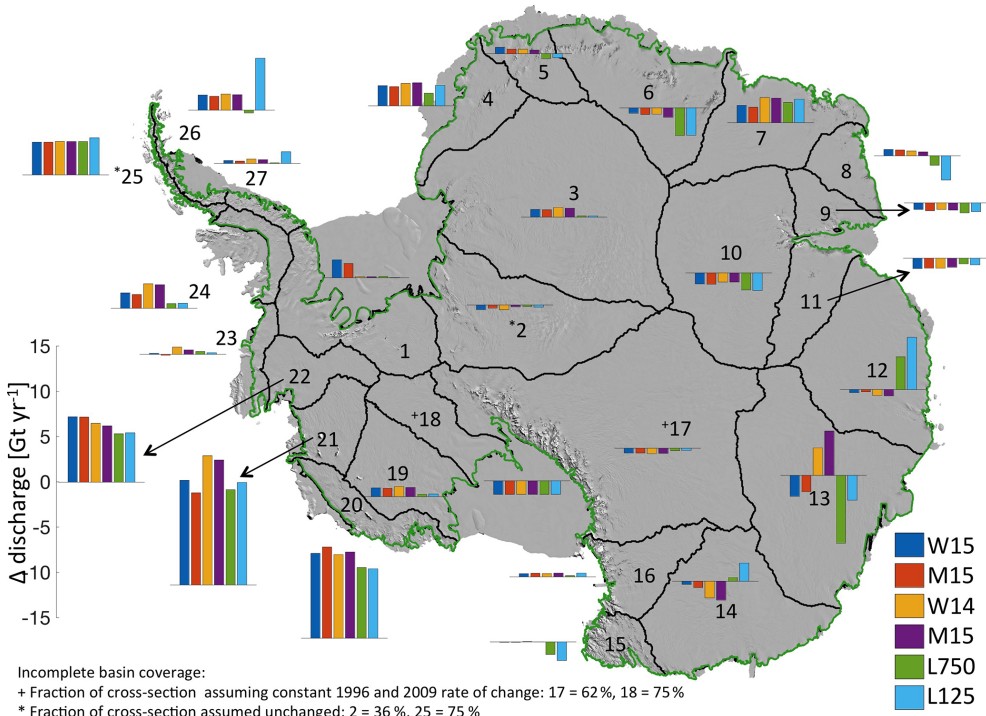

**Figure 7.** Change in flux across FG1 flux gate (shown with green line; see Methods) for the 27 basins defined by Zwally et al. (2002) calculated by differencing the pan-Antarctic SAR mapping of Rignot et al. (2011a, circa 2008) with six different Landsat 8 velocity mappings (M14/15 = JPL median of all 2014/15 image pairs; W14/15 = JPL weighted average of all 2014/15 image pairs; L750 = NSIDC 750 m average of all 2014–2015 image pairs; L125 = NSIDC 125 m average of all 2014–2015 image pairs). Basins 2, 17 and 18 are complimented with differences in 1997 and 2009 SAR velocities poleward of 82.5° S (Scheuchl et al., 2012). Much of the difference between velocity mappings can be attributed to product errors. W15 has the lowest uncertainties (used in this study), followed by M15, then W14 and M14, with the LISA products having the highest uncertainties (See Fig. A1).

products is not entirely known but is likely due to differences in geolocation offset correction and merging procedures. Some difference between mappings can also be expected due to real changes in ice flow between effective dates. This good agreement between products gives us confidence that our results are not sensitive to the Landsat processing methodology. From here forward we only present results generated using auto-RIFT W15 that provides the lowest uncertainties and longest period over which change in discharge is calculated.

### 3.1.1 Amundsen Sea

For the B21 and B22 catchments, containing Pine Island, Thwaites, Haynes, Pope, Smith and Kohler glaciers (Fig. 8), we find a 6 % increase in ice discharge or $17 \pm 4$ Gt yr$^{-1}$ (Table 2). This implies an average discharge increase of 2.4 Gt yr$^{-2}$ for 2008–2015 that is considerably lower than the 6.5 Gt yr$^{-2}$ previously estimated for 1994–2008 (Mouginot et al., 2014). This recent slowing in the rate of acceleration is in excellent agreement with the previously published temporally dense history of ice discharge that gave a rate of discharge increase for this region of 2.3 Gt yr$^{-2}$ for overlap-

ping but shorter period of 2010–2013 period (Mouginot et al., 2014). Pine Island and Thwaites glaciers both show clear signs of persistent dynamic drawdown, with velocities increasing by $> 100$ m yr$^{-1}$ up to 80–100 km inland from the grounding line (Fig. 9). Figure 9 shows a peak in Pine Island velocity change at 50 km and a secondary peak at 110 km upstream of the grounding line. We see no such peak when comparing between Landsat products, which makes us confident that the secondary peak is not an artifact of the Landsat processing. One possible non-geophysical explanation is that the radar mosaic includes data from a period significantly earlier than 2008 for area of the second peak. East Kohler and Smith glaciers also show extensive speedups throughout their length, with increases of $> 100$ m yr$^{-1}$ reaching more than 40 km inland likely driven by increased ocean melt rates and subsequent grounding-line retreat (Khazendar et al., 2016; Scheuchl et al., 2016). Patterns of velocity change for Pope and Kohler glaciers are more complex, with slowing of up to 100 m yr$^{-1}$ near the grounding line and increased speed by $\sim 50$ m yr$^{-1}$ upstream reaching 40–80 km inland. This pattern of change is suggestive of an earlier period of dynamic drawdown that is slowly propagating inland contrasted by more recent deceleration near the grounding line. Glaciers

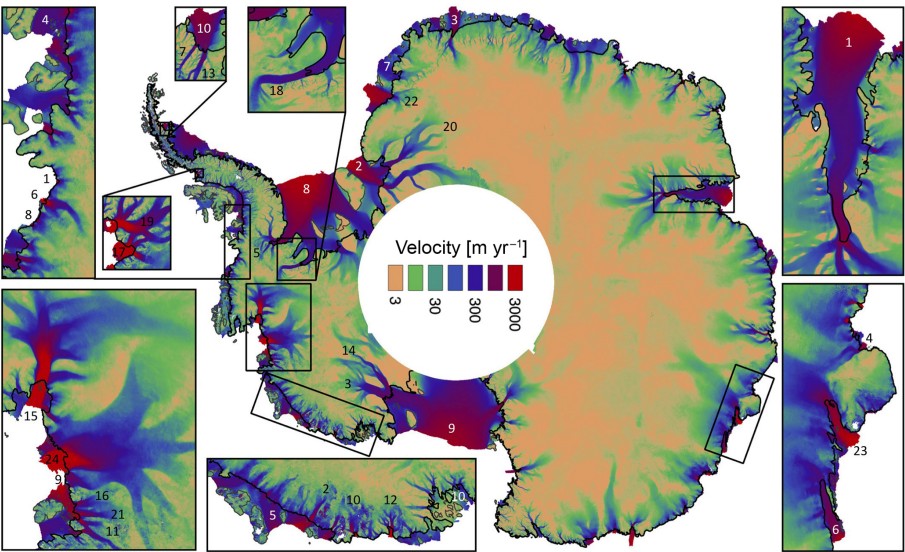

**Figure 8.** 2015 Antarctic ice sheet surface velocities shown in log scale determined from feature tracking of > 200 000 Landsat image pairs. Glacier and ice streams discussed in text labeled with black numbering: (1) Alison, (2) Berry, (3) Bindschadler, (4) Bond, (5) Evans, (6) Ferrigno, (7) Flask, (8) Fox, (9) Haynes, (10) Hull, (11) Kohler, (12) Land, (13) Leppard, (14) MacAyeal, (15) Pine Island, (16) Pope, (17) Prospect, (18) Rutford, (19) Seller, (20) Slessor, (21) Smith, (22) Stancomb-Wills, (23) Totten, and (24) Twaites. Ice shelves labeled with white numbering: (1) Amery, (2) Filchner, (3) Fimbul, (4) George VI, (5) Getz, (6) Moscow U., (7) Riiser-Larsen, (8) Ronne, (9) Ross, (10) Scar Inlet, and (10) Sulzberger.

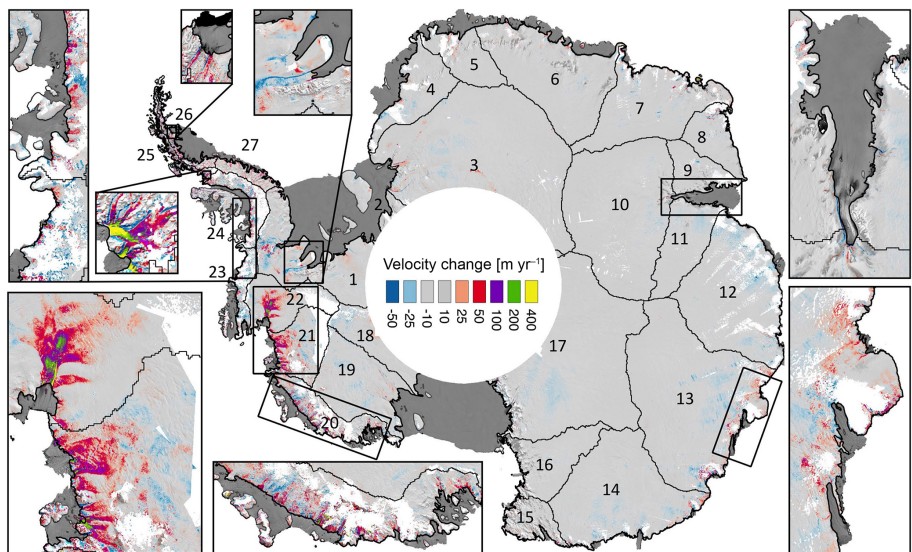

**Figure 9.** Change in surface velocities between date of pan-Antarctic SAR mapping (Rignot et al., 2011a, circa 2008) and new 2015 velocity mapping produced here from feature tracking of Landsat 8 imagery. Change in velocities shown for grounded ice only. Missing data shown in white and the 27 basin boundaries defined by Zwally et al. (2002) are shown in black.

feeding the Getz and Sulzberger ice shelves (B20; including Berry, Hull and Land glaciers) increased in speed by 10 to 100 m yr$^{-1}$ at their grounding lines, increasing discharge by 6 % (Table 2). This result is in broad agreement with Chuter et al. (2017) that observed increases in ice velocity during the 2007–2013 period alongside 2010–2013 dynamic thin-

ning rates of 0.7 m yr$^{-1}$ for the glaciers feeding the Abbot and Getz ice shelves.

### 3.1.2 Bellingshausen coast

Localized accelerations of 50–200 m yr$^{-1}$ are observed near grounding lines for several of the major glaciers along the

Bellingshausen Coast (B23 and B24) including the Ferrigno, Fox and Alison ice streams and glaciers feeding into the southern George VI Ice Shelf. Despite some areas of flow acceleration, increases in discharge are highly localized. For many glaciers, the flux-gate cross section is decreasing from regional thinning, resulting in negligible changes in discharge. This result is unexpected, but with high confidence, as this region has experienced high rates of ice shelf thinning (Paolo et al., 2015) and grounding-line retreat (Christie et al., 2016), both of which were inferred to have resulted in accelerated dynamic thinning that contributed to a $56 \pm 8\,\mathrm{Gt\,yr^{-1}}$ increase in the rate of mass loss that began around 2009 and persisted until at least April 2014 (Wouters et al., 2015). From our analysis we conclude that any changes in discharge contributing to observed rates of thinning must have occurred prior to the SAR mapping of ice velocities. This result agrees with a recent investigation of longer-term (1995–2016) changes in ice discharge for this region (Hogg et al., 2017) that found that the region's glacier experienced an increase in ice discharge between 1995 and 2008 and almost no change in discharge between 2008 and 2016.

### 3.1.3 Northern Antarctic Peninsula

Along the west coast of the northern Antarctic Peninsula (B25) glaciers feeding into Marguerite Bay (Seller and Prospect) sped up by $400\text{–}800\,\mathrm{m\,yr^{-1}}$ at their grounding lines, the largest speedup of all Antarctic glaciers, with an increase of $> 100\,\mathrm{m\,yr^{-1}}$ reaching 10–15 km upstream. This speedup was recently attributed increased ocean melt rates resulting from SOI/ENSO-driven ocean warming (Walker and Gardner, 2017). The majority of the west-coast glaciers to the north of Marguerite Bay are not sufficiently sampled in the earlier SAR mapping and are assumed to be unchanged between 2008 and 2015 (Fig. 2). Along the east coast of the northern Antarctic Peninsula (B26) most glaciers feeding into the former Larsen A and B ice shelves that collapsed in 1995 and 2002, respectively, either have not changed significantly or show signs of slowing near their grounding lines (Wuite et al., 2015) with the exception of Leppard and Flask glaciers. These two glaciers have sped up by $50\text{–}100\,\mathrm{m\,yr^{-1}}$ at their grounding lines, likely in response to reduced ice shelf buttressing and a resulting speedup of the abutting Scar Inlet Ice Shelf (Khazendar et al., 2015). Overall, this region shows a modest increase in ice discharge of $6 \pm 6\,\mathrm{Gt\,yr^{-1}}$, most of which comes from the glaciers flowing into Marguerite Bay. Small changes in rates of discharge between periods are in good agreement with constant rates of RACMO-derived surface mass budget and mass changes derived from GRACE data (Appendix B).

### 3.1.4 Ice streams feeding large ice shelves

Our analysis suggests a $5\text{–}20\,\mathrm{m\,yr^{-1}}$ slowdown of a broad region upstream of both Bindschadler and MacAyeal ice streams, which feed the Ross Ice Shelf. Ice streams feeding the Ronne-Filchner Ice Shelf show heterogeneous changes with slowing of $15\text{–}40\,\mathrm{m\,yr^{-1}}$ upstream of the Rutford and Evans ice stream grounding lines and $\sim 20\,\mathrm{m\,yr^{-1}}$ speedup of the Slessor Ice Stream. Slowing in the Rutford Ice Stream is consistent with the slowing observed between 1997 and 2009 (Scheuchl et al., 2012), but the apparent increase in velocity of the Slessor Ice Stream is of equal magnitude but of opposite sign to the changes observed between 1997 and 2009 (Scheuchl et al., 2012). Further to the east, the Stancomb-Wills Glacier increased in speed by $20\text{–}40\,\mathrm{m\,yr^{-1}}$, just upstream of the grounding line, with glaciers feeding the Riiser-Larsen, Fimbul and Amery ice shelves showing little change. Overall, changes in surface velocity along grounding lines of ice streams and glaciers feeding the major ice shelves of East and West Antarctica have not been large enough to significantly impact the net ice discharge for their respective basins (Table 2).

### 3.1.5 East Antarctic glaciers

Ice discharge has remained remarkably steady for the East Antarctic glaciers, particularly along the coasts of Dronning Maud Land and Enderby Land. These basins (B5–B8) showed very little change in ice discharge. The region to the west of Law Dome, including Underwood and Bond glaciers, shows subtle evidence of some increased flow speed and ice discharge, though the signal is near the limit of detection in part due to larger errors in the earlier radar mosaic for this region. However, the much larger Totten Glacier and the tributaries of the Moscow University Ice Shelf (B13) that drain a large fraction of the East Antarctic Ice Sheet show localized areas of ice speed variations but little change in discharge (Fig. 1). This result is consistent with recent findings of Li et al. (2016) showing that the Totten Glacier increased in velocity between 2001 and 2007, likely in response to elevated ocean temperature, but has been relatively unchanged since.

### 3.1.6 Antarctic discharge

In total we estimate that between the SAR and auto-RIFT W15 velocity mappings, the Antarctic ice sheet increased its solid ice discharge to the ocean from $1897 \pm 41$ to $1932 \pm 38\,\mathrm{yr^{-1}}$. This represents a $36 \pm 15\,\mathrm{Gt\,yr^{-1}}$ increase in total discharge between 2008 and 2015; 79 % of the increases in discharge concentrated to glaciers flowing into the Amundsen Sea and another 11 % coming from glaciers flowing into Marguerite Bay. Breaking it down to the main ice-sheet regions, the discharge of the West Antarctic Ice Sheet (B1, B18–23) increased by $30 \pm 15\,8\,\mathrm{Gt\,yr^{-1}}$ and the Antarctic Peninsula (B24–27) by $7 \pm 6\,\mathrm{Gt\,yr^{-1}}$, representing a 4 and 3 % increase in discharge, respectively. The discharge of the East Antarctic Ice Sheet (B2–17) was remarkably unchanged with a total discharge of $956 \pm 31$ and $952 \pm 29\,\mathrm{Gt\,yr^{-1}}$ in 2008 and 2015, respectively.

Our estimate of 2008 total Antarctic ice discharge ($1894 \pm 43\,yr^{-1}$) is smaller than earlier estimates of $2048 \pm 146$ and $2049 \pm 86\,Gt\,yr^{-1}$ by Rignot et al. (2013) and Depoorter et al. (2013), respectively. Both earlier studies use the same SAR velocity mosaic as used here (Rignot et al., 2011a). Our estimate agrees with that of Rignot et al. (2013) within stated errors but not with that of Depoorter et al. (2013). Rignot et al. TS15 used Operation Ice Bridge and BEDMAP-2 ice thickness data at InSAR derived grounding lines to determine a total Antarctic discharge, with upscaling accounting for $352\,Gt\,yr^{-1}$ of the total discharge. The most obvious reason for the difference in the central estimates is the definition of the flux gates. Rignot et al. (2013) mostly rely on BEDMAP-2 data while our study draws almost entirely from flight data. Another possible reason for the difference is the upscaling of results for unmeasured basins. For these basins the total discharge is assumed to be the modeled climatological average surface mass balance integrated over the upstream basin. Such estimates have not been adjusted for losses due to basal melt, and they are sensitive to errors in the modeled SMB and to the delineation of the contributing basin area over which SMB is integrated. Upscaling for unmeasured areas by Depoorter et al. (2013) accounted for $476\,Gt\,yr^{-1}$ of their estimated discharge. The Depoorter et al. (2013) study uses a different definition of groundling but otherwise uses the same data as used in Rignot et al. (2013). Again, much of the difference between estimates can be attributed to the definition of ice thickness and upscaling to unmeasured basins. It should also be noted that Depoorter et al. (2013) and Rignot et al. (2013) both used output from an earlier version of RACMO that produced larger total SMB than the version of the model used in our study. Since SMB is used to upscale discharge, this likely contributes some to the larger discharge estimates. Similar conclusions were made for updated Greenland Ice Sheet discharge estimates that were lower than previous estimates (Enderlin et al., 2014).

### 3.2 Changes in net mass balance

For the West Antarctic Ice Sheet, the 2008–2015 net mass budgets were negative for all but two basins (B1 and B18) (Fig. 10), summing to a total imbalance of $-214 \pm 51\,Gt\,yr^{-1}$ with largest rates of loss collocated with increased glacier velocities along the Amundsen Sea Embayment (B21 and B22) and Getz Ice Shelf (B20). The ~~mass~~ large loss for the Getz Ice Shelf region is in contrast to the near balance conditions recently reported by Chuter et al. (2017) for the 2006–2008 period but is in agreement with the 2010–2013 estimate of net mass change by Martín-Español et al. (2016). The East Antarctic Ice Sheet is found to have increased slightly in mass at a rate of $61 \pm 73\,Gt\,yr^{-1}$ with largest gains in Dronning Maud (B6) and Enderby Land (B7 and B8) that can be partially attributed to increase in precipitation rate ($+28\,Gt\,yr^{-1}$ relative to 1979–2007 mean) during the study period, which is consistent with earlier

findings (Boening et al., 2012; King et al., 2012; Shepherd et al., 2012). For the whole of Antarctica, we estimate an average mass budget of $-183 \pm 94\,Gt\,yr^{-1}$ for the 2008–2015 period. Other recent estimates of Antarctic mass change include those derived from CryoSat-2 altimetry of $-159 \pm 48\,Gt\,yr^{-1}$ for the period 2010–2013 (McMillan et al., 2014) and $-116 \pm 76\,Gt\,yr^{-1}$ for the period 2011–2014 (Helm et al., 2014, assuming density of ice) and a recent estimate from the joint inversion of gravity, altimetry and GPS data of $-159 \pm 22\,Gt\,yr^{-1}$ for the period 2010–2013 (Martín-Español et al., 2016). All three studies show near balance to slightly positive mass changes for the East Antarctic Ice Sheet and large losses for the West Antarctic Ice Sheet and the Antarctic Peninsula, all of which agree well with the results presented here when considering uncertainties and differences in study periods.

### 4 Discussion

Areas of accelerated surface velocity (Fig. 9) and increased ice discharge are in good agreement with basin-scale assessment of changes in ice flow and ice discharge (Li et al., 2016; Mouginot et al., 2014) and with patterns of ice sheet thinning determined from laser and radar altimetry (Flament and Rémy, 2012; Helm et al., 2014; Pritchard et al., 2009). These show broad regions of surface lowering for glaciers feeding into the Amundsen Sea Embayment and Getz Ice Shelf and rapid drawdown of smaller glacier systems in the Antarctic Peninsula. Glaciers and ice streams feeding major ice shelves were remarkably steady with small heterogeneous changes in velocity. Apparent upstream slowing of Bindschadler and MacAyeal ice streams are at the limit of detectability and difficult to interpret. Recent assessments show varying changes in ice stream velocities for this region (Hulbe et al., 2016; Scheuchl et al., 2012), suggesting that measured trends may be influenced by rapid changes in the sub-ice-stream hydrology (Hulbe et al., 2016).

Strongly negative net mass budgets are apparent for the West Antarctic Ice Sheet and are largely due to mean rates of ice discharge greatly exceeding rates of snow accumulation. The basin-averaged results (Fig. 10) match remarkably well with patterns of pan-Antarctic multi-decadal (1994–2012) changes in ice shelf thickness (Paolo et al., 2015): high rates of mass loss from glaciers feeding into the Amundsen Sea are collocated with high rates of ice shelf thinning and near balance conditions for Wilkes Land glaciers and basins feeding the Filchner-Ronne, Ross and Amery ice shelves are collocated with ice shelves that have experienced little change in ice thickness over the past two decades. This result further supports the strong link between oceanic melting of ice shelves and ice sheet mass budget (Pritchard et al., 2012).

The link between basin mass budget and change in discharge is less obvious. This is primarily due to differences in representative periods as mass budgets represent the cumu-

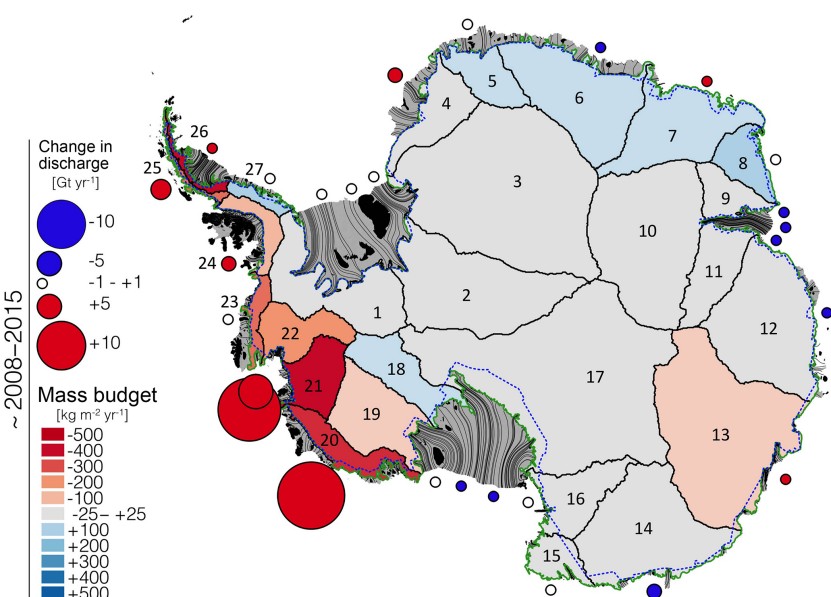

**Figure 10.** Mass budget and change in discharge for the 27 basins defined by Zwally et al. (2002). Mass budget is calculated as described in Table 2 using 2008–2015 average surface mass balance in the main and inset figures. Change in discharges (circles) calculated by differencing the pan-Antarctic SAR mapping of Rignot et al. (2011a; circa 2008) with weighted average of all 2015 image-pair displacements supplemented with 2009 SAR velocities to fill missing Landsat coverage poleward of 82.5° S (Scheuchl et al., 2012) with a correction for acquisition time differences to provide an estimate of total discharge for the interior basins (2, 17 and 18; see Table 2). Flux gates FG1 and FG2 are shown with solid green and dashed blue lines, respectively.

lative imbalance away from equilibrium state while changes in discharge are only representative of change in discharge between two periods in time; e.g., a glacier can decelerate but still be discharging ice at a rate that exceeds the surface mass budget minus basal melt. Increased ice discharge from the Amundsen Sea Embayment and subsequent partial re-stabilization have been attributed to changes in ice shelf buttressing (Jacobs et al., 1996; Macgregor et al., 2012) that resulted from increased ice shelf basal melt rates (Jacobs et al., 2011; Jenkins et al., 1997) and more recently to a decrease in ocean melting resulting from changes in the temperature of intermediate depth waters (Dutrieux et al., 2014). Increased discharge from glaciers feeding into the Getz Ice Shelf is likely in response to rapid thinning of the ice shelf due to changes in ocean circulation and the depth of warmer modified Circumpolar Deep Water (Jacobs et al., 2013).

## 5 Conclusion

Applying novel feature tracking methods to hundreds of thousands of Landsat image pairs we are now able construct a detailed and comprehensive record of recent changes in Antarctic-wide ice flow. When combined with optimized flux-gate definitions and an earlier mapping of surface velocity (Rignot et al., 2011a), such measurements allow for accurate reconstructions of ice discharge and changes in ice discharge through time. Applying these new capabili-

ties, we determine that the Antarctic ice sheet discharged 1897 ± 41 yr⁻¹ of solid ice into the ocean in 2008, increasing to 1932 ± 38 yr⁻¹ in 2015 with 79 % of the increase in discharge concentrated to glaciers flowing into the Amundsen Sea and another 11 % comes from glaciers flowing into Marguerite Bay. Glaciers and ice streams feeding major ice shelves were remarkably steady with small heterogeneous changes in velocity. Strongly negative net mass budgets are apparent for the West Antarctic Ice Sheet and are largely due to mean rates of ice discharge greatly exceeding rates of snow accumulation. The East Antarctic Ice Sheet experienced near-balance conditions with modest gains in Dronning Maud and Enderby Land driven by increased rates of precipitation.

Over the last decade, it is evident that larger-scale changes in discharge are relatively modest (< 7 % for all basins) compared to the fractional imbalance between discharge and surface mass budget (up to several tens of percent). This suggests that the recent pattern of mass loss in Antarctica, dominated by the Amundsen Sea sector, is likely a part of a longer-term phase of enhanced glacier flow initiated in the 1990s as indicated by satellite records (Konrad et al., 2017; Mouginot et al., 2014) or as early as the 1940s as proposed from sub-ice-shelf sediment records (Smith et al., 2017).

Glaciology is rapidly transitioning from an observationally constrained environment to one with ample high-quality, high-volume satellite data sets suitable for mapping ice flow on continental scales (e.g., Landsat 8, Sentinel 2a/b, Sen-

tinel 1a/b). This study provides a foundation for continued assessment of ice sheet flow and discharge that will allow researches to observe both large and subtle changes ice sheet flow that may indicate early signs of ice sheet instability with low latency. Such a capability would help to diagnose unstable flow behavior and, in conjunction with high accuracy measurements of ice sheet elevation and mass change, would lead to improved assessment ice sheet surface mass balance and ice shelf melt rates. Low-latency monitoring of ice flow and discharge would also allow field programs, flight planning and satellite tasking to coordinate the collection complimentary observations in areas of changing ice behavior. These advances will ultimately lead to a deeper understanding of the causal mechanisms resulting in observed and future ice sheet instabilities. Any substantial improvement in our assessment of ice sheet discharge will require more detailed knowledge of ice thickness just upstream of the grounding line, particularly for areas of complex flow such as the Antarctic Peninsula and Victoria Land. Errors in discharge estimates can be greatly reduced if thickness profiles are acquired perpendicular to ice flow. Improved estimates of net mass change calculated using the mass budget approach will come from continued refinement of regional climate models and better estimates of basal melt.

*Data availability.* ~~All velocity mosaics are available from NSIDC. Grounding lines, flux gates and ancillary data are provided as Supplementary Data~~ TS16.

## Appendix A: Uncertainty quantification

### A1 Ice discharge

The uncertainty in flux estimates were calculated for each of the 27 basins as

$$\sigma F = \left( \sigma F_H^2 + \sigma F_{dH}^2 + \sigma F_V^2 + \sigma F_{SMB}^2 + \sigma F_{dV_{dyn}/dt}^2 + \sigma F_{bm}^2 \right.$$
$$\left. + \sigma F_{\overline{V}}, \right. \tag{A1}$$

where $\sigma F_H$ is due to uncertainties in ice-equivalent thickness, $\sigma F_{dH}$ is due to uncertainties in the change of ice-equivalent thickness between the measurement times of ice thickness and surface velocity, and $\sigma F_V$ is due to uncertainties in measured velocity. $\sigma F_{\overline{V}}$ is due to the assumption that the depth-averaged velocity ($\overline{V}$) is equal to the surface velocity and is added as a bias (outside of the quadrature sum) to both sides of the error envelope for simplicity. $\sigma F_{dV_{dyn}/dt}$, $\sigma F_{SMB}$ and $\sigma F_{bm}$ are uncertainties introduced by dynamic volume change, surface mass balance and basal melt corrections applied between the flux gate ~~the true~~ the grounding-line. $\sigma F_{dV_{dyn}/dt}$ was taken to be $0.1\,\text{m yr}^{-1}$ for surfaces moving faster than $200\,\text{m yr}^{-1}$. $\sigma F_{SMB}$ was taken to be 20 % of the SMB. Uncertainties in flux resulting from uncertainties in ice thickness, changes in ice thickness and surface velocity were propagated assuming a conservative correlation length along the flux gate as follows:

$$\sigma F_H = \sqrt{\sum_1^{n_H} \left( \sum_{i=1}^{m_H} \sigma H_i W_i V_i \right)^2}, \tag{A2}$$

$$\sigma F_{dH} = \sqrt{\sum_1^{n_{dH}} \left( \sum_{i=1}^{m_{dH}} \sigma \frac{dH}{dt}_i dt_i W_i V_i \right)^2}, \tag{A3}$$

$$\sigma F_V = \sqrt{\sum_1^{n_V} \left( \sum_{i=1}^{m_V} \sigma V_i W_i H_i \right)^2 + \sum_{i=1}^{nn} \sigma V_{0i} W_i H_i}, \tag{A4}$$

where $m$ is the number of point estimates of flux ($x$) for each correlation length distance along the flux gate and $n$ is the number of discrete uncorrelated lengths for each basin for measurements of ice thickness ($H$), changes in ice thickness ($dH$) and the surface velocity normal to the flux gate ($V$). Uncertainties in ice thickness ($\sigma H_i$) are taken as the RSS of the thickness estimate and the FAC. Uncertainties in changes in ice thickness ($\sigma \frac{dH}{dt}$) are determined as the RSS of uncertainty due to changes in FAC and surface elevation. $dt$ is the difference in time between the measurement of ice thickness and the measurement of surface velocity. $\sigma F_V$ is modeled using a velocity uncertainty component $\sigma V_0$ that is fully correlated at lengths smaller than an estimated correlation length and uncorrelated at larger lengths ($\sigma V$). Comparing Landsat and SAR velocities measured at flux-gate nodes for basins with minimal change in ice discharge (B1–19 and B27); i.e., where velocity differences are assumed to be indicative measurement uncertainty, we were able to model the observed

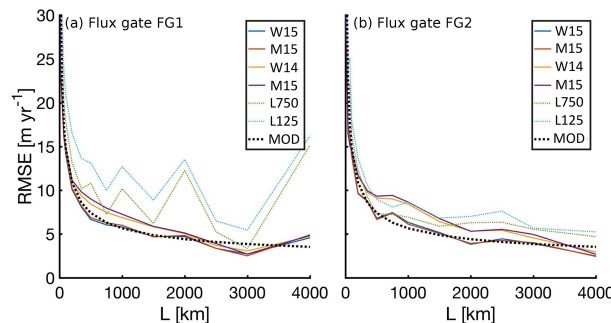

**Figure A1.** RMSE of the Landsat component of velocity that is normal to the flux-gate cross section at FG1 **(a)** and FG2 **(b)** flux nodes relative to $\sim 2008$ SAR velocities (Rignot et al., 2011 [TS17]) as a function of averaging distance ($L$). MOD is the modeled uncertainty assuming a fully correlated uncertainty of $1\,\text{m yr}^{-1}$ plus a $3\,\text{m yr}^{-1}$ uncertainty that is uncorrelated at distances greater than 1000 km.

RMSE between Landsat and SAR observations (Fig. A1) setting $\sigma V_0 = 3\,\text{m yr}^{-1}$ and $\sigma V = 1\,\text{m yr}^{-1}$ with a correlation length of 1000 km for both the SAR and Landsat mappings. Uncertainties in velocities can be as high as $20$–$30\,\text{m yr}^{-1}$ locally but are largely uncorrelated on basin scales. There are insufficient data to determine rigorous estimates of the correlation lengths for ice thickness, change in ice thickness and surface velocity, all of which are likely spatially variable. Instead we took a conservative approach and assigned a correlation length of 1000 km to all three measurements.

When calculating ice flux we assumed that the surface velocity was equal to the depth-averaged velocity. This approach neglects vertical gradients in ice velocity that result from the stress-dependent plastic deformation (creep) of ice. Since surface velocities are always larger than the depth-averaged velocity this introduced a positive bias into our estimates of ice flux. Neglecting sliding and assuming a depth constant creep parameter ($A$) the depth-averaged velocity is 80 % of the surface velocity (Cuffey and Paterson, 2010). Assuming parallel flow and a linear increase in shear stress with depth, the surface velocity due to creep ($V_s$ [TS18]) can be calculated as follows:

$$V_s = \frac{2A}{1+n} t_b^n H, \tag{A5}$$

where $n$ is the creep exponent, $H$ is the ice thickness and $t_b$ is the driving stress at the bed. $n$ is typically assumed to be 3 and so is done here. $t_b$ is calculated using the surface slope and ice depth (Cuffey and Paterson, 2010). The creep parameter $A$ (Fig. A2a) is taken from Ice Sheet System Model (ISSM) output generated as part of the Sea-level Response to Ice Sheet Evolution (SeaRISE) experiments (Bindschadler et al., 2013). We calculated surface slope from a CryoSat-2 DEM that was smoothed on a scale of several times the ice thickness (20 km). Ice thickness was taken from Bedmap-2 (Fretwell et al., 2013). $V_s$ varied between $0\,\text{m yr}^{-1}$ at the ice

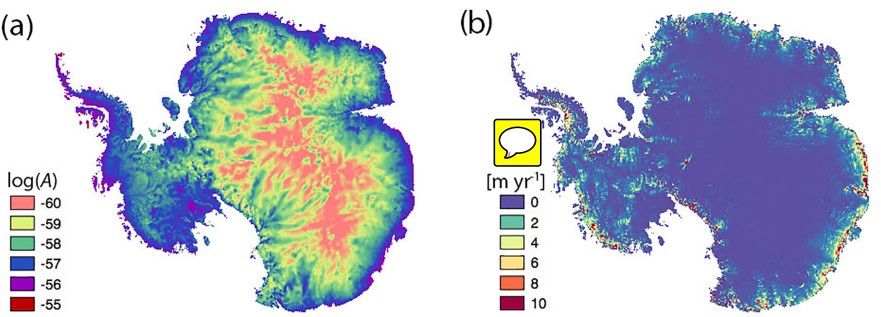

**Figure A2.** Creep parameter ($A$: $\mathrm{s^{-1}\,Pa^{-3}}$) shown in log scale **(a)**. Estimated surface velocity due to ice creep ($U_{s_x}$).

divides and $10\,\mathrm{m\,yr^{-1}}$ in steeply sloped outlet glaciers near the coast (Fig. A2b). We considered 20 % of Vs to be the upper bound of the bias introduced into our flux estimates due to vertical gradients in the velocity field ($\sigma F_{\overline{V}}$), calculated as

$$\sigma F_{\overline{V}} = 0.2 \sum_{x=1}^{nn} \mathrm{Vs}_i\, W_i\, H_i, \tag{A6}$$

where nn is the number nodes along the basin flux gate. This is an upper bound scenario, as $A$ increases rapidly with temperature, and ice sheet temperature is typically at a maximum near the bed. This results in a higher concentration of shear deformation near the base of the ice sheet than inferred from a depth-constant $A$.

Uncertainties in flux estimates were assumed to be uncorrelated between basins. A detailed accounting of each flux term and their associated error is provided in Tables A1 through A3. Table A1 provides detailed breakdown for the total discharge calculated using FG2 as the flux gate. This approach produces the discharge estimate with the lowest error and is the approach used in the main paper. For comparison, Tables A2 and A3 provide detailed breakdowns for the total discharge calculated using FG1 and GL0, respectively.

## A2   Change in ice discharge

Uncertainty in flux-change estimates ($\sigma dF$) are calculated as

$$\sigma dF = \sqrt{\sigma dF_H^2 + \sigma dF_{dH}^2 + \sigma dF_V^2 + \sigma dF_{no\_data}^2}, \tag{A7}$$

where $\sigma dF_H$ is the thickness-related uncertainty and is calculated as

$$\sigma dF_H = \sigma F_{H0} \frac{dF}{F}, \tag{A8}$$

where $dF$ is the change in flux and $F$ is the total flux. $\sigma dF_{dH}$ is calculated in the same way as $\sigma F_{dH}$ but setting $dt$ to the time separation between repeat measurements of velocity. $\sigma dF_v$ is the flux-change uncertainty from the measured velocity and is determined as

$$\sigma dF_v = \sqrt{\sigma F_{v1}^2 + \sigma dF_{v2}^2}, \tag{A9}$$

where $\sigma F_v$ is the uncertainty in flux introduced from uncertainties in surface velocity for two measurement epochs (1 and 2). $\sigma dF_{no\_data}$ is the flux-change uncertainty introduced by the assumption of zero change in flux for areas lacking reliable repeat measurements ($\sigma F_{no\_data}$) and for areas between the flux gate and the grounding line ($\sigma F_{SMB}$) and is calculated as

$$\sigma dF_{no\_data} = 0.1 \left( \sigma F_{SMB} + \sigma F_{no\_data} \right). \tag{A10}$$

Uncertainties in flux-change estimates were assumed to be uncorrelated between basins. A detailed accounting of each change in flux term and their associated error is provided in Table A4.

**Table A1.** Detailed breakdown of total discharge estimate ($D$) presented in Table 2 using JPL auto-RIFT 2015 weighted average velocity (W15), FG2 flux gate and GL0 grounding line. Surface areas are for the total basin area upstream of the grounding line, flux gate and the area between the flux gate and the grounding line. $F$ is the total flux across the flux gate. SMB, BM and $dV_{dyn}/dt$ are the surface mass balance, basal melt and dynamic volume change integrated over the area between the flux gate and the grounding line. All error terms and their propagation are describe in Sect. A1.

| ID basin | Surface area (km²) | | | Length (km) | Flux through FG2 (Gt yr⁻¹) | | | | | | Additional flux between FG2 and GL0 [Gt yr⁻¹] | | | | | | Total discharge (Gt yr⁻¹) | |
|---|---|---|---|---|---|---|---|---|---|---|---|---|---|---|---|---|---|---|
| | GL0 | FG2 | GL0 − FG2 | FG2 | $F$ | $\sigma F_H$ | $\sigma F_{dH}$ | $\sigma F_V$ | $\sigma F_{V\_bar}$ | $\sigma F$ | SMB | $\sigma F_{SMB}$ | BM | $\sigma F_{BM}$ | $dV_{dyn}/dt$ | $\sigma F_{dV_{dyn}/dt}$ | $D$ | $\sigma D$ |
| 1 | 474 821 | 463 896 | 10 926 | 1170 | 108.8 | 5.6 | 0.3 | 3.3 | 0.2 | 6.8 | 2.7 | 0.5 | 0.0 | 0.0 | 0.1 | 0.2 | 111.7 | 6.8 |
| 2 | 765 381 | 757 558 | 7824 | 359 | 46.0 | 3.2 | 0.0 | 1.8 | 0.1 | 3.7 | 0.9 | 0.2 | 0.0 | 0.0 | 0.2 | 0.1 | 47.1 | 3.7 |
| 3 | 1 556 551 | 1 545 527 | 11 024 | 253 | 58.1 | 3.4 | 0.1 | 1.6 | 0.0 | 3.8 | 1.6 | 0.3 | 0.0 | 0.0 | 0.0 | 0.1 | 59.8 | 3.8 |
| 4 | 241 158 | 226 961 | 14 197 | 680 | 39.5 | 7.0 | 0.4 | 2.0 | 0.1 | 7.4 | 3.8 | 0.8 | −0.1 | 0.0 | −0.1 | 0.2 | 43.3 | 7.4 |
| 5 | 185 337 | 182 253 | 3084 | 415 | 29.3 | 4.0 | 0.4 | 1.5 | 0.1 | 4.4 | 1.6 | 0.3 | 0.0 | 0.0 | −0.1 | 0.0 | 30.8 | 4.4 |
| 6 | 607 737 | 583 513 | 24 224 | 740 | 53.8 | 5.3 | 0.2 | 2.4 | 0.2 | 6.0 | 6.0 | 1.2 | −0.2 | 0.0 | −0.2 | 0.2 | 59.7 | 6.1 |
| 7 | 492 518 | 451 677 | 40 841 | 799 | 49.1 | 5.9 | 0.3 | 2.4 | 0.2 | 6.6 | 20.6 | 4.1 | −0.4 | 0.0 | −0.3 | 0.6 | 69.9 | 7.9 |
| 8 | 161 243 | 144 458 | 16 785 | 485 | 11.9 | 1.9 | 0.2 | 1.8 | 0.1 | 2.7 | 6.2 | 1.2 | −0.2 | 0.0 | −0.5 | 0.1 | 17.7 | 3.0 |
| 9 | 146 003 | 140 392 | 5612 | 386 | 15.0 | 1.1 | 0.2 | 1.6 | 0.1 | 2.0 | 1.3 | 0.3 | 0.0 | 0.0 | 0.0 | 0.1 | 16.4 | 2.1 |
| 10 | 919 320 | 918 041 | 1278 | 56 | 33.0 | 2.9 | 0.2 | 1.2 | 0.7 | 3.2 | 0.0 | 0.0 | 0.0 | 0.0 | 0.0 | 0.1 | 33.1 | 3.2 |
| 11 | 255 178 | 249 424 | 5753 | 275 | 11.9 | 0.7 | 0.1 | 1.5 | 1.0 | 2.4 | 0.4 | 0.1 | −0.1 | 0.0 | 0.0 | 0.1 | 12.3 | 2.4 |
| 12 | 727 088 | 698 205 | 28 883 | 882 | 89.0 | 8.8 | 0.6 | 2.7 | 1.0 | 10.2 | 11.0 | 2.2 | −0.6 | 0.0 | 0.7 | 0.8 | 101.2 | 10.5 |
| 13 | 1 130 843 | 1 095 837 | 35 006 | 804 | 192.7 | 15.9 | 0.9 | 2.8 | 1.4 | 17.5 | 27.5 | 5.5 | −0.7 | 0.0 | 2.5 | 1.0 | 223.3 | 18.5 |
| 14 | 718 511 | 667 028 | 51 483 | 734 | 102.2 | 6.7 | 0.5 | 3.0 | 0.5 | 7.9 | 26.9 | 5.4 | −0.8 | 0.0 | 0.1 | 0.8 | 130.0 | 9.7 |
| 15 | 123 780 | 14 735 | 109 044 | 187 | 1.3 | 0.1 | 0.0 | 1.5 | 0.1 | 1.7 | 24.5 | 4.9 | −0.5 | 0.1 | 0.2 | 0.4 | 26.4 | 5.3 |
| 16 | 262 005 | 236 067 | 25 937 | 339 | 10.7 | 0.7 | 0.1 | 1.4 | 0.0 | 1.6 | 2.9 | 0.6 | −0.2 | 0.0 | 0.0 | 0.1 | 13.7 | 1.7 |
| 17 | 1 825 799 | 1 667 954 | 157 845 | 775 | 46.0 | 3.8 | 0.2 | 2.8 | 0.4 | 5.2 | 22.9 | 4.6 | −0.4 | 0.0 | −2.3 | 1.8 | 67.0 | 7.3 |
| 18 | 261 357 | 234 457 | 26 900 | 185 | 7.4 | 1.1 | 0.0 | 1.3 | 0.0 | 1.7 | 2.0 | 0.4 | 0.0 | 0.0 | −1.9 | 1.2 | 7.5 | 2.2 |
| 19 | 367 678 | 347 019 | 20 659 | 353 | 42.9 | 5.6 | 0.1 | 1.7 | 0.0 | 5.8 | 2.4 | 0.5 | −0.1 | 0.0 | 0.1 | 0.8 | 45.5 | 5.9 |
| 20 | 180 072 | 139 929 | 40 143 | 714 | 140.6 | 11.4 | 0.9 | 2.3 | 0.5 | 12.2 | 33.9 | 6.8 | −0.5 | 0.0 | 8.2 | 2.5 | 183.2 | 14.3 |
| 21 | 207 491 | 203 265 | 4227 | 276 | 180.8 | 11.1 | 0.7 | 1.5 | 0.2 | 11.4 | 4.7 | 0.9 | −0.1 | 0.0 | 3.7 | 1.1 | 189.3 | 11.5 |
| 22 | 210 237 | 206 351 | 3886 | 177 | 130.9 | 7.8 | 0.3 | 1.3 | 0.0 | 7.9 | 2.8 | 0.6 | 0.0 | 0.0 | 0.8 | 0.2 | 134.5 | 7.9 |
| 23 | 74 562 | 53 294 | 21 269 | 396 | 60.6 | 5.1 | 0.6 | 1.7 | 0.3 | 5.7 | 20.2 | 4.0 | −0.3 | 0.0 | 1.5 | 0.5 | 82.5 | 7.0 |
| 24 | 100 567 | 93 479 | 7088 | 429 | 86.3 | 5.9 | 0.5 | 1.6 | 0.2 | 6.3 | 7.5 | 1.5 | −0.1 | 0.0 | 1.3 | 0.4 | 95.2 | 6.5 |
| 25 | 34 657 | 18 312 | 16 345 | 475 | 34.4 | 3.5 | 0.8 | 1.2 | 0.2 | 3.9 | 54.9 | 11.0 | −0.1 | 0.0 | 2.0 | 0.6 | 91.4 | 11.8 |
| 26 | 42 025 | 30 162 | 11 863 | 469 | 17.3 | 1.7 | 0.4 | 1.3 | 0.5 | 2.7 | 7.2 | 1.4 | 0.0 | 0.0 | 0.2 | 0.1 | 24.7 | 3.1 |
| 27 | 51 962 | 40 319 | 11 643 | 297 | 6.9 | 0.6 | 0.1 | 1.2 | 0.0 | 1.4 | 5.0 | 1.0 | 0.0 | 0.0 | 0.0 | 0.0 | 11.9 | 1.7 |
| Total | 12 123 881 | 11 410 110 | 713 770 | 13 108 | 1606.3 | 31.8 | 2.2 | 10.2 | 2.2 | 35.2 | 301.5 | 17.9 | −5.3 | 0.1 | 16.1 | 4.1 | 1929.2 | 39.9 |

**Table A2.** Same as Table A1 but using the FG1 for the flux gate.

| ID basin | Surface area (km²) | | | Length (km) | Flux through FG1 (Gt yr⁻¹) | | | | | | Additional flux between FG1 and GL0 [Gt yr⁻¹] | | | | | | Total discharge (Gt yr⁻¹) | |
|---|---|---|---|---|---|---|---|---|---|---|---|---|---|---|---|---|---|---|
| | GL0 | FG1 | GL0 – FG1 | FG1 | $F$ | $\sigma F_H$ | $\sigma F_{dH}$ | $\sigma F_V$ | $\sigma F_{V\_bar}$ | $\sigma F$ | SMB | $\sigma F_{SMB}$ | BM | $\sigma F_{BM}$ | $dV_{dyn}/dt$ | $\sigma F_{dV_{dyn}/dt}$ | $D$ | $\sigma D$ |
| 1 | 474821 | 466855 | 7967 | 1287 | 109.8 | 5.3 | 0.3 | 3.2 | 0.2 | 6.5 | 1.8 | 0.4 | 0.0 | 0.0 | 0.1 | 0.2 | 111.7 | 6.5 |
| 2 | 765381 | 761534 | 3847 | 370 | 46.9 | 2.9 | 0.1 | 1.9 | 0.0 | 3.6 | 0.5 | 0.1 | 0.0 | 0.0 | 0.2 | 0.1 | 47.5 | 3.6 |
| 3 | 1556551 | 1553115 | 3437 | 299 | 59.6 | 3.8 | 0.1 | 1.6 | 0.0 | 4.2 | 0.3 | 0.1 | 0.0 | 0.0 | 0.1 | 0.1 | 60.0 | 4.2 |
| 4 | 241158 | 239208 | 1950 | 843 | 44.5 | 10.7 | 0.5 | 1.9 | 0.0 | 10.9 | 0.6 | 0.1 | 0.0 | 0.0 | 0.0 | 0.0 | 45.1 | 10.9 |
| 5 | 185337 | 184737 | 600 | 489 | 30.4 | 4.6 | 0.5 | 1.6 | 0.1 | 5.0 | 0.2 | 0.1 | 0.0 | 0.0 | 0.0 | 0.0 | 30.6 | 5.0 |
| 6 | 607737 | 604178 | 3559 | 1080 | 59.0 | 9.1 | 0.2 | 2.3 | 0.1 | 9.4 | 0.8 | 0.2 | 0.0 | 0.0 | 0.1 | 0.0 | 59.7 | 9.4 |
| 7 | 492518 | 492159 | 359 | 1253 | 45.1 | 23.0 | 1.4 | 1.6 | 0.1 | 23.1 | 0.2 | 0.2 | 0.0 | 0.0 | -0.2 | 0.1 | 45.3 | 23.1 |
| 8 | 161243 | 160984 | 259 | 554 | 14.8 | 7.3 | 0.4 | 1.4 | 0.0 | 7.5 | 0.2 | 0.0 | 0.0 | 0.0 | 0.0 | 0.0 | 14.9 | 7.5 |
| 9 | 146003 | 145979 | 24 | 466 | 14.5 | 2.0 | 0.1 | 1.7 | 0.0 | 2.7 | 0.0 | 0.0 | 0.0 | 0.0 | 0.0 | 0.0 | 14.5 | 2.7 |
| 10 | 919320 | 919149 | 171 | 36 | 37.3 | 3.6 | 0.1 | 1.2 | 0.2 | 4.0 | 0.0 | 0.0 | 0.0 | 0.0 | 0.0 | 0.0 | 37.3 | 4.0 |
| 11 | 255178 | 255033 | 145 | 333 | 9.0 | 1.3 | 0.1 | 1.6 | 0.8 | 2.8 | 0.0 | 0.0 | 0.0 | 0.0 | 0.0 | 0.0 | 9.0 | 2.8 |
| 12 | 727088 | 726521 | 567 | 1072 | 124.6 | 38.6 | 1.8 | 2.0 | 0.1 | 38.8 | 0.0 | 0.1 | 0.0 | 0.0 | 0.0 | 0.0 | 124.9 | 38.8 |
| 13 | 1130843 | 1125684 | 5159 | 1005 | 202.3 | 35.7 | 2.0 | 2.6 | 1.1 | 36.9 | 4.1 | 0.8 | -0.1 | 0.0 | 1.3 | 0.4 | 207.8 | 36.9 |
| 14 | 718511 | 716677 | 1834 | 1129 | 134.0 | 24.6 | 1.6 | 2.2 | 0.1 | 24.8 | 1.2 | 0.2 | -0.1 | 0.0 | 0.1 | 0.1 | 135.3 | 24.8 |
| 15 | 123780 | 123620 | 160 | 1102 | 19.7 | 14.0 | 0.4 | 1.7 | 0.7 | 14.8 | 0.0 | 0.0 | 0.0 | 0.0 | 0.0 | 0.0 | 19.7 | 14.8 |
| 16 | 262005 | 261418 | 587 | 554 | 7.2 | 1.7 | 0.2 | 1.2 | 0.0 | 2.1 | 0.0 | 0.1 | 0.0 | 0.0 | 0.0 | 0.0 | 7.2 | 2.1 |
| 17 | 1825799 | 1823861 | 1938 | 1235 | 53.8 | 10.9 | 0.3 | 2.1 | 0.1 | 11.2 | 0.3 | 0.1 | 0.0 | 0.0 | -0.1 | 0.1 | 53.9 | 11.2 |
| 18 | 261357 | 259869 | 1488 | 444 | 10.6 | 2.0 | 0.1 | 1.6 | 0.0 | 2.6 | 0.1 | 0.0 | 0.0 | 0.0 | -0.1 | 0.0 | 10.6 | 2.6 |
| 19 | 367678 | 366585 | 1094 | 419 | 44.9 | 6.2 | 0.1 | 1.7 | 0.0 | 6.4 | 0.1 | 0.0 | 0.0 | 0.0 | 0.0 | 0.1 | 45.0 | 6.4 |
| 20 | 180072 | 173181 | 6891 | 1382 | 146.9 | 23.4 | 1.1 | 2.1 | 0.4 | 23.9 | 7.4 | 1.5 | -0.2 | 0.0 | 6.1 | 1.8 | 160.6 | 24.0 |
| 21 | 207491 | 205221 | 2271 | 326 | 181.9 | 11.9 | 0.8 | 1.5 | 0.2 | 12.2 | 2.3 | 0.5 | -0.1 | 0.0 | 3.7 | 1.1 | 188.0 | 12.3 |
| 22 | 210237 | 208363 | 1874 | 219 | 132.1 | 8.4 | 0.4 | 1.3 | 0.0 | 8.6 | 1.3 | 0.3 | 0.0 | 0.0 | 0.8 | 0.2 | 134.2 | 8.6 |
| 23 | 74562 | 72800 | 1763 | 881 | 78.2 | 17.3 | 1.1 | 1.9 | 0.0 | 17.4 | 2.7 | 0.5 | 0.0 | 0.0 | 0.7 | 0.2 | 81.6 | 17.5 |
| 24 | 100567 | 97297 | 3271 | 511 | 88.1 | 8.9 | 0.7 | 1.6 | 0.2 | 9.3 | 2.9 | 0.6 | 0.0 | 0.0 | 1.0 | 0.3 | 92.0 | 9.3 |
| 25 | 34657 | 33834 | 823 | 1360 | 62.9 | 25.6 | 1.9 | 1.3 | 0.0 | 25.8 | 1.7 | 0.3 | 0.0 | 0.0 | 0.8 | 0.2 | 65.4 | 25.8 |
| 26 | 42025 | 41888 | 138 | 1300 | 26.1 | 7.7 | 1.9 | 1.5 | 0.5 | 8.4 | 0.0 | 0.0 | 0.0 | 0.0 | 0.2 | 0.0 | 26.2 | 8.4 |
| 27 | 51962 | 51562 | 400 | 703 | 8.9 | 2.4 | 0.6 | 1.5 | 0.0 | 2.8 | 0.2 | 0.0 | 0.0 | 0.0 | 0.0 | 0.0 | 9.1 | 2.8 |
| Total | 12123881 | 12071309 | 52572 | 20653 | 1792.9 | 80.3 | 4.6 | 9.5 | 1.7 | 82.0 | 29.3 | 2.0 | -0.6 | 0.0 | 14.4 | 2.3 | 1837.3 | 82.1 |

**Table A3.** Same as Table A1 but using the GL0 for the flux gate.

| ID basin | Surface area (km²) | | | Length (km) | Flux through FG1 (Gt yr⁻¹) | | | | | | Additional flux between FG1 and GL0 [Gt yr⁻¹] | | | | | | Total discharge (Gt yr⁻¹) | |
|---|---|---|---|---|---|---|---|---|---|---|---|---|---|---|---|---|---|---|
| | GL0 | GL0 | GL0 − GL0 | GL0 | $F$ | $\sigma F_H$ | $\sigma F_{dH}$ | $\sigma F_V$ | $\sigma F_{V\_bar}$ | $\sigma F$ | SMB | $\sigma F_{SMB}$ | BM | $\sigma F_{BM}$ | $dV_{dyn}/dt$ | $\sigma F_{dV_{dyn}/dt}$ | $D$ | $\sigma D$ |
| 1 | 474 821 | 474 821 | 0 | 1651 | 112.1 | 8.2 | 0.3 | 3.9 | 0.4 | 9.5 | 0.0 | 0.0 | 0.0 | 0.0 | 0.0 | 0.0 | 112.1 | 9.5 |
| 2 | 765 381 | 765 381 | 0 | 457 | 39.4 | 4.0 | 0.1 | 2.3 | 0.0 | 4.6 | 0.0 | 0.0 | 0.0 | 0.0 | 0.0 | 0.0 | 39.4 | 4.6 |
| 3 | 1 556 551 | 1 556 551 | 0 | 325 | 50.9 | 7.4 | 0.2 | 1.7 | 0.0 | 7.6 | 0.0 | 0.0 | 0.0 | 0.0 | 0.0 | 0.0 | 50.9 | 7.6 |
| 4 | 241 158 | 241 158 | 0 | 920 | 47.3 | 14.2 | 0.6 | 2.0 | 0.0 | 14.4 | 0.0 | 0.0 | 0.0 | 0.0 | 0.0 | 0.0 | 47.3 | 14.4 |
| 5 | 185 337 | 185 337 | 0 | 530 | 29.2 | 5.0 | 0.4 | 1.7 | 0.1 | 5.3 | 0.0 | 0.0 | 0.0 | 0.0 | 0.0 | 0.0 | 29.2 | 5.3 |
| 6 | 607 737 | 607 737 | 0 | 1144 | 58.5 | 13.2 | 0.3 | 2.2 | 0.0 | 13.4 | 0.0 | 0.0 | 0.0 | 0.0 | 0.0 | 0.0 | 58.5 | 13.4 |
| 7 | 492 518 | 492 518 | 0 | 1265 | 44.5 | 22.5 | 1.4 | 1.6 | 0.0 | 22.6 | 0.0 | 0.0 | 0.0 | 0.0 | 0.0 | 0.0 | 44.5 | 22.6 |
| 8 | 161 243 | 161 243 | 0 | 574 | 13.7 | 6.9 | 0.4 | 1.4 | 0.0 | 7.0 | 0.0 | 0.0 | 0.0 | 0.0 | 0.0 | 0.0 | 13.7 | 7.0 |
| 9 | 146 003 | 146 003 | 0 | 554 | 14.1 | 2.0 | 0.2 | 1.9 | 0.0 | 2.8 | 0.0 | 0.0 | 0.0 | 0.0 | 0.0 | 0.0 | 14.1 | 2.8 |
| 10 | 919 320 | 919 320 | 0 | 35 | 33.4 | 3.2 | 0.1 | 1.2 | 0.2 | 3.5 | 0.0 | 0.0 | 0.0 | 0.0 | 0.0 | 0.0 | 33.4 | 3.5 |
| 11 | 255 178 | 255 178 | 0 | 343 | 8.8 | 1.3 | 0.1 | 1.6 | 0.8 | 2.9 | 0.0 | 0.0 | 0.0 | 0.0 | 0.0 | 0.0 | 8.8 | 2.9 |
| 12 | 727 088 | 727 088 | 0 | 1144 | 125.7 | 40.2 | 1.9 | 2.1 | 0.1 | 40.4 | 0.0 | 0.0 | 0.0 | 0.0 | 0.0 | 0.0 | 125.7 | 40.4 |
| 13 | 1 130 843 | 1 130 843 | 0 | 1248 | 202.6 | 37.7 | 1.9 | 2.8 | 0.4 | 38.2 | 0.0 | 0.0 | 0.0 | 0.0 | 0.0 | 0.0 | 202.6 | 38.2 |
| 14 | 718 511 | 718 511 | 0 | 1162 | 129.7 | 32.4 | 1.8 | 2.1 | 0.0 | 32.6 | 0.0 | 0.0 | 0.0 | 0.0 | 0.0 | 0.0 | 129.7 | 32.6 |
| 15 | 123 780 | 123 780 | 0 | 1121 | 20.3 | 14.3 | 0.4 | 1.7 | 0.8 | 15.2 | 0.0 | 0.0 | 0.0 | 0.0 | 0.0 | 0.0 | 20.3 | 15.2 |
| 16 | 262 005 | 262 005 | 0 | 597 | 7.5 | 2.2 | 0.2 | 1.3 | 0.0 | 2.6 | 0.0 | 0.0 | 0.0 | 0.0 | 0.0 | 0.0 | 7.5 | 2.6 |
| 17 | 1 825 799 | 1 825 799 | 0 | 1288 | 53.3 | 9.9 | 0.3 | 2.2 | 0.1 | 10.3 | 0.0 | 0.0 | 0.0 | 0.0 | 0.0 | 0.0 | 53.3 | 10.3 |
| 18 | 261 357 | 261 357 | 0 | 510 | 10.4 | 2.2 | 0.1 | 1.7 | 0.0 | 2.8 | 0.0 | 0.0 | 0.0 | 0.0 | 0.0 | 0.0 | 10.4 | 2.8 |
| 19 | 367 678 | 367 678 | 0 | 540 | 44.6 | 6.9 | 0.2 | 1.9 | 0.0 | 7.1 | 0.0 | 0.0 | 0.0 | 0.0 | 0.0 | 0.0 | 44.6 | 7.1 |
| 20 | 180 072 | 180 072 | 0 | 1581 | 139.2 | 34.6 | 1.3 | 2.2 | 0.1 | 34.7 | 0.0 | 0.0 | 0.0 | 0.0 | 0.0 | 0.0 | 139.2 | 34.7 |
| 21 | 207 491 | 207 491 | 0 | 370 | 181.7 | 16.1 | 1.5 | 1.6 | 0.2 | 16.4 | 0.0 | 0.0 | 0.0 | 0.0 | 0.0 | 0.0 | 181.7 | 16.4 |
| 22 | 210 237 | 210 237 | 0 | 266 | 128.9 | 10.6 | 0.5 | 1.3 | 0.0 | 10.7 | 0.0 | 0.0 | 0.0 | 0.0 | 0.0 | 0.0 | 128.9 | 10.7 |
| 23 | 74 562 | 74 562 | 0 | 936 | 80.7 | 24.6 | 1.4 | 1.9 | 0.0 | 24.7 | 0.0 | 0.0 | 0.0 | 0.0 | 0.0 | 0.0 | 80.7 | 24.7 |
| 24 | 100 567 | 100 567 | 0 | 557 | 78.7 | 23.0 | 1.4 | 1.5 | 0.0 | 23.1 | 0.0 | 0.0 | 0.0 | 0.0 | 0.0 | 0.0 | 78.7 | 23.1 |
| 25 | 34 657 | 34 657 | 0 | 1323 | 62.4 | 35.4 | 2.4 | 1.3 | 0.0 | 35.5 | 0.0 | 0.0 | 0.0 | 0.0 | 0.0 | 0.0 | 62.4 | 35.5 |
| 26 | 42 025 | 42 025 | 0 | 1326 | 27.5 | 8.9 | 0.7 | 1.5 | 0.2 | 9.2 | 0.0 | 0.0 | 0.0 | 0.0 | 0.0 | 0.0 | 27.5 | 9.2 |
| 27 | 51 962 | 51 962 | 0 | 784 | 10.6 | 4.7 | 0.3 | 1.5 | 0.0 | 4.9 | 0.0 | 0.0 | 0.0 | 0.0 | 0.0 | 0.0 | 10.6 | 4.9 |
| Total | 12 123 881 | 12 123 881 | 0 | 22 550 | 1755.6 | 98.0 | 5.3 | 10.1 | 1.3 | 99.2 | 0.0 | 0.0 | 0.0 | 0.0 | 0.0 | 0.0 | 1755.6 | 99.2 |

**Table A4.** Detailed breakdown of the change in discharge ($\Delta D$) estimate presented in Table 2 using JPL auto-RIFT 2015 weighted average velocity (W15), $\sim 2008$ velocities from Rignot et al. (2011a), the FG1 flux gate and GL0 grounding line. d$F$ is change in flux across the grounding line and dd$V_{\mathrm{dyn}}/$d$t$ is the change in dynamic volume change for the area between FG1 and GL0. All error terms and their propagation are describe in Sect. A2.

| ID basin | Surface area (km$^2$) | | | Length (km) | $\Delta$ flux through FG1 (Gt yr$^{-1}$) | | | | | Additional $\Delta$ flux between FG1 and GL [Gt yr$^{-1}$] | Total $\Delta$ discharge (Gt yr$^{-1}$) | |
| --- | --- | --- | --- | --- | --- | --- | --- | --- | --- | --- | --- | --- |
| | GL | FG1 | GL – FG1 | FG1 | d$F$ | $\sigma$d$F_H$ | $\sigma$d$F_{\mathrm{d}H}$ | $\sigma$d$F_V$ | $\sigma$d$F_{\mathrm{no\_data}}$ | dd$V_{\mathrm{dyn}}/$d$t$ | d$D$ | $\sigma$d$D$ |
| 1 | 474 821 | 466 855 | 7967 | 1287 | 2.0 | 0.1 | 0.6 | 3.0 | 1.1 | −0.4 | 1.6 | 3.2 |
| 2 | 765 381 | 761 534 | 3847 | 370 | −0.5 | 0.0 | 0.1 | 2.0 | 3.8 | −0.3 | −0.8 | 4.3 |
| 3 | 1 556 551 | 1 553 115 | 3437 | 299 | 0.8 | 0.0 | 0.4 | 2.1 | 0.1 | −0.1 | 0.7 | 2.2 |
| 4 | 241 158 | 239 208 | 1950 | 843 | 2.2 | 0.4 | 1.0 | 2.4 | 0.4 | 0.0 | 2.2 | 2.7 |
| 5 | 185 337 | 184 737 | 600 | 489 | 0.7 | 0.1 | 0.8 | 2.0 | 0.3 | 0.1 | 0.7 | 2.2 |
| 6 | 607 737 | 604 178 | 3559 | 1080 | −0.6 | 0.1 | 0.7 | 2.5 | 0.6 | 0.2 | −0.3 | 2.7 |
| 7 | 492 518 | 492 159 | 359 | 1253 | 1.9 | 0.2 | 0.7 | 2.1 | 0.3 | 0.0 | 1.9 | 2.2 |
| 8 | 161 243 | 160 984 | 259 | 554 | 0.7 | 0.1 | 0.4 | 1.9 | 0.0 | 0.0 | 0.7 | 2.0 |
| 9 | 146 003 | 145 979 | 24 | 466 | −0.7 | 0.1 | 0.2 | 2.2 | 0.0 | −0.1 | −0.8 | 2.2 |
| 10 | 919 320 | 919 149 | 171 | 36 | −1.2 | 0.1 | 0.3 | 1.6 | 0.0 | 0.0 | −1.2 | 1.7 |
| 11 | 255 178 | 255 033 | 145 | 333 | −1.2 | 0.1 | 0.1 | 2.0 | 0.0 | 0.0 | −1.1 | 2.0 |
| 12 | 727 088 | 726 521 | 567 | 1072 | −0.3 | 0.0 | 1.3 | 2.8 | 0.4 | 0.0 | −0.3 | 3.1 |
| 13 | 1 130 843 | 1 125 684 | 5159 | 1005 | −2.3 | 0.2 | 2.3 | 3.3 | 2.8 | 0.0 | −2.3 | 4.9 |
| 14 | 718 511 | 716 677 | 1834 | 1129 | −0.4 | 0.0 | 1.5 | 2.8 | 0.5 | 0.4 | 0.0 | 3.2 |
| 15 | 123 780 | 123 620 | 160 | 1102 | 0.0 | 0.0 | 0.0 | 2.3 | 0.0 | 0.0 | 0.0 | 2.3 |
| 16 | 262 005 | 261 418 | 587 | 554 | 0.3 | 0.0 | 0.3 | 1.7 | 0.0 | 0.0 | 0.4 | 1.7 |
| 17 | 1 825 799 | 1 823 861 | 1938 | 1235 | −0.5 | 0.0 | 0.4 | 2.1 | 2.5 | 0.2 | −0.3 | 3.3 |
| 18 | 261 357 | 259 869 | 1488 | 444 | −1.5 | 0.0 | 0.0 | 1.4 | 1.2 | 0.1 | −1.4 | 1.9 |
| 19 | 367 678 | 366 585 | 1094 | 419 | 1.0 | 0.1 | 0.3 | 2.0 | 0.1 | 0.2 | 1.2 | 2.0 |
| 20 | 180 072 | 173 181 | 6891 | 1382 | 11.6 | 0.9 | 2.6 | 2.8 | 1.1 | 0.7 | 12.2 | 4.1 |
| 21 | 207 491 | 205 221 | 2271 | 326 | 9.4 | 0.6 | 2.8 | 2.1 | 0.2 | 0.1 | 9.5 | 3.6 |
| 22 | 210 237 | 208 363 | 1874 | 219 | 7.2 | 0.4 | 1.1 | 1.8 | 0.1 | −0.1 | 7.1 | 2.2 |
| 23 | 74 562 | 72 800 | 1763 | 881 | 0.1 | 0.0 | 1.7 | 2.6 | 0.5 | −0.1 | 0.0 | 3.2 |
| 24 | 100 567 | 97 297 | 3271 | 511 | 1.7 | 0.1 | 2.1 | 2.2 | 0.4 | −0.8 | 0.8 | 3.1 |
| 25 | 34 657 | 33 834 | 823 | 1360 | 3.6 | 0.3 | 0.9 | 1.6 | 4.3 | −0.6 | 3.0 | 4.7 |
| 26 | 42 025 | 41 888 | 138 | 1300 | 1.7 | 0.2 | 0.9 | 2.1 | 0.4 | 0.0 | 1.7 | 2.3 |
| 27 | 51 962 | 51 562 | 400 | 703 | 0.3 | 0.0 | 0.2 | 2.1 | 0.0 | 0.0 | 0.3 | 2.1 |
| Total | 12 123 881 | 12 071 309 | 52 572 | 20 653 | 35.9 | 1.3 | 6.2 | 11.7 | 7.3 | −0.5 | 35.4 | 15.2 |

## Appendix B: Northern Antarctic Peninsula net mass balance

Narrow deep fjords and steep spatial and temporal gradients in surface mass balance for the northern Antarctic Peninsula (B25–26) introduced large and poorly characterized uncertainties into estimates of ice discharge and $\sigma F_{\mathrm{d}v/\mathrm{d}t}$ that propagated to highly uncertain estimates of net mass change. For this reason, we derived our estimates of net mass change using previously published estimates from repeat surface elevation measurements that we added to our estimates of change in ice discharge. Work by Scambos et al. (2014), based on elevation changes and recent gravity work (Harig and Simons, 2015), suggests that the northern Antarctic Peninsula region (precise study extents vary) has seen continued mass losses at more or less a constant rate of 25–30 Gt yr$^{-1}$ for the period 2003–2015; this is further supported by examination of JPL mascon (Watkins et al., 2015) mass anomalies and RACMO surface mass budget anomalies (See Fig. B1). Estimates based on CryoSat-2 (McMillan et al., 2014) suggest a reduced mass loss for B25 and B26 (below the significance level) for the period 2010–2013, but usable data from CryoSat-2 for this rugged region are sparse.

To estimate the net mass balance for basins B25 and B26, we used estimates of glacier mass loss determined from repeat elevation measurements for the 2003–2011 period as a starting point (Scambos et al., 2014). Since this study was restricted to areas north of 66° S, we added our estimate of change in ice discharge south of 66° S (6 Gt yr$^{-1}$: Table 2) to estimate the basin-wide net mass balances for 2008–2015. The uncertainty in the net budget was calculated as the RSS of the uncertainty in the basin estimate of change in discharge, the uncertainty in the net balance estimated in Scambos et al. (2014) and the uncertainty in the surface mass budget. Basin totals and uncertainties are provided in Table 2.

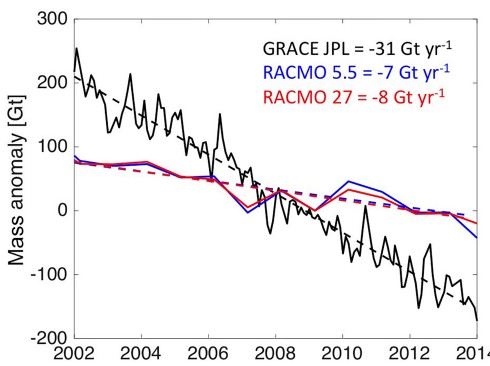

**Figure B1.** Rates of 2002–2014 mass change as derived from linear fits to cumulative anomalies of RACMO surface mass balance determined at 5.5 km (blue line) and 27 km (red line), and JPL V2.0 mascon anomalies (black line: grace.jpl.nasa.gov/data/get-data/jpl_global_mascons/) for the northern Antarctic Peninsula. Surface mass balance and JPL mascon anomalies were integrated for the seven mascons overlapping the northern Antarctic Peninsula (4324, 4325, 4372, 4373, 4374, 4415, 4416). For plotting purposes the surface mass balance anomalies were determined relative to the 1979–2003 mean. JPL mascons are corrected for changes in solid earth using the glacial isostatic adjustment (GIA) correction modeled by Geruo A and John Wahr. This figure is provided to support the argument for a relatively steady rate of northern Antarctic Peninsula mass change between the 2003 and 2015 and not to support the magnitude of that change, which is sensitive to the choice of the model used for the GIA correction.

*Author contributions.* ASG devised the study, developed the JPL auto-RIFT software, did all calculations and wrote the paper. GM was responsible for updating the grounding-line location and defining the flux gates, he also spent considerable time revising the manuscript after the lead author broke his wrist while snowboarding. TS and MF produced the LISA velocity fields; SL and MvdB provided modeled FAC and SMB output and JN produced surface elevation change rates from CryoSat-2 data. All authors discussed and commented on the manuscript at all stages.

*Competing interests.* The authors declare that they have no conflict of interest.

*Acknowledgements.* We thank TS20 B. Van Liefferinge and F. Pattyn for kindly sharing their modeled estimates of basal melt rates, B. Wouters for helpful discussions regarding Bellingshausen Sea glacier mass changes, A. Khazendar for helping to provide context for observed changes in glacier velocity, I. Joughin for helpful discussion regarding SAR velocities and sharing of data not used in this study, P. Fretwell for providing information on Bedmap-2 and E. Rignot, J. Mouginot and B. Scheuchl for making their SAR velocities publically available, without which this study would not have been possible. Ted Scambos is deeply appreciative of T. Haran and M. Klinger for all of their hard work creating the NSIDC velocity maps. This work was supported by funding from the NASA Cryosphere program. The research was conducted at the Jet Propulsion Laboratory, California Institute of Technology, under contract with the National Aeronautics and Space Administration and at University of Colorado Boulder and University of Alaska Fairbanks under NASA grant NNX16AJ88G.

Edited by: G. Hilmar Gudmundsson
Reviewed by: three anonymous referees

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

## Remarks from the language copy-editor

CE1    This abbreviation is not defined. Is it well known or should it be defined for clarity?

## Remarks from the typesetter

TS1    The composition of Figs. 1, 4 and 6–B1 has been adjusted to our standards.

TS2    2011a or b or a, b?

TS3    2011a or b or a, b?

TS4    Please confirm exponential writing throughout the text.

TS5    There is no Supplement to this paper. Please check.

TS6    2011a or b or a, b?

TS7    Should this be "to" (–)?

TS8    Please confirm.

TS9    2011a or b or a, b?

TS10    Please note that the first column in the table as it is in the manuscript ("basin") is missing here.

TS11    Please provide explanation for values in italic font.

TS12    This reference is not in the reference list. Please add it.

TS13    This reference is not in the reference list. Please add it.

TS14    This reference is not in the reference list. Please add it.

TS15    Please provide year.

TS16    There is no Supplement to this paper. Please check.

TS17    2011a or b or a, b?

TS18    Should this be $V_s$? Please check throughout.

TS19    Is this a link?

TS20    Please provide full first names throughout this section.

TS21    Please provide name.

TS22    Please provide date of last access.

TS23    Please provide all author names.

TS24    Please provide page range or article number.

TS25    Please provide date of last access.

TS26    Please provide volume number and page range or article number.

TS27    Please update if possible.

TS28    Please provide volume number and page range or article number.

TS29    Please provide all author names.

TS30    Please provide date of last access.

TS31    Please provide all author names.

TS32    Please provide volume number and page range or article number.

TS33    Please provide page range or article number.

TS34    Please provide all author names.

TS35    Please provide page range or article number.

TS36    Please provide page range and article number with DOI.

TS37    Please provide page range or article number with DOI.

TS38    Please provide place of publication.

TS39    Please provide all author names.

TS40    Please provide all author names.

TS41    Please provide publisher.

TS42    Please provide all author names.

TS43    Please provide date of last access.