# Peer review of "Increased West Antarctic and unchanged East Antarctic ice discharge over the last seven years"

_The Cryosphere, 2017_

## Referee Comment (RC1) · Anonymous Referee #1 · 10 Aug 2017

General comments

The manuscript shows the present-day Antarctic-wide surface velocities using Landsat7/8 images and an assessment of mass discharge change compared to earlier ice velocity map inferred from synthetic aperture radar. The work itself is of significance for the glaciology community to help to understand the present-day situation of Antarctic ice sheet. But the manuscript do not provide new insight to scientific community. I have five major concerns in the matter.

1. There is another manuscript in discussion in TC submitted earlier. Both papers are discussing the same issue. Although some results seem to be similar, my concern is that the both seem to draw different conclusions in term of ice discharge change in Antarctic ice sheet. The causes of the differences should be discussed in details.

Shen, Q., Wang, H., Shum, C.-K., Jiang, L., Hsu, H. T., and Dong, J.: Antarctic high-resolution ice flow mapping and increased mass loss in Wilkes Land, East Antarctica during 2006–2015, The Cryosphere Discuss., and https://doi.org/10.5194/tc-2017-34

2. The estimation of uncertainties of ice discharge changes were not rigorously based on error propagation law by an intentional and non-scientific method so that small estimates were obtained. Accordingly, uncertainties of ice discharge changes were obviously underestimated (see Table 2). For example, the uncertainty for all of Antarctica should be ±56 Gt/yr (sqrt(41^2+38^2)), not ± 15 Gt/yr (see Table 2 ) as stated in the paper. The uncertainties for individual basins and sectors (East Antarctica, West Antarctica and AP) were also underestimated. The estimate of the ice discharge change in the Antarctica should be 35±56 Gt/yr. Therefore, it is incorrect to conclude that there was a certain increased ice discharge since ~2008. There are no significant acceleration of ice discharge in West Antarctica using a correct uncertainty estimate for ice discharge change, rather than increase ice discharge as mentioned in the title.

3. The paper stated that in the calculation of ice discharge, the uncertainties were apparently reduced due to the extensive use of RES data. But I do not think that the use of RES can really reduce the uncertainties. At first, the uncertainties of dynamic volume and surface mass balance were not shown in the tables of the manuscript. In general, the uncertainty of firn densification model is relatively large during the transfer between elevation change and mass change but was not shown. Additionally, the elevation change was directly considered as the dynamic volume which is problematic, because there are many driven factors of elevation change of ice glacier/sheet, for example, firn densification, the snowfall change, basal melting etc. The surface mass balance is another large error source to the uncertainty of ice discharge using the FG2. Furthermore, the small estimates of uncertainty may result from the large number of statistical units as much as 27. For example, in the paper, the uncertainty of total ice discharge were calculated based on 27 basins, while Depoorter et al. (Nature, 2013) estimated the uncertainty based on six oceanic sectors, and Rignot et al. (Science 2013) summed the uncertainties of each calculation units (ice shelf). More importantly, calculation of ice discharge is highly sensitive to the definition of the flux gate, the intentional movement of grounding line could cause the over 20% error in individual ice discharge even if the RES data are used. Therefore,

the method for the calculation of ice discharge needs to be rigorously validated before use although this method was previously proposed by other authors.

4. The authors (and also Shen et al. in review) used first Antarctic-wide ice velocity (Rignot et al. 2011, science) as a reference map, the reference year are 2008 and 2006 respectively. The MEaSUREs Antarctic ice velocity map (v1.1) was inferred from over a long period (1996-2009) according its production statements. The data were acquired as early as 1996. Therefore the SAR-derived ice velocity map as a single year is problematic. The new MEaSUREs products have released and annual maps from 2005 to 2016 can be obtained (http://nsidc.org/data/nsidc-0720), authors should use the new products to alleviate the problem prompted.

5. The authors used different products of ice velocity (M14/15, W14/15, L750 and L124) to estimate the change in flux across FG1, but the values are apparently different. For example, there are conflicting estimates of ice discharge changes in basins 8, 12, 13, 14. In particular, the discharge is decreased for the M14/15 while it is increased for W14/15. Noted that they used the same data, only difference is that the mosaicking methods. Unfortunately, for the choice of the velocity data, the author did not present any convincible standards. The accuracy of ice velocity products should be carefully assessed using the independent surveyed data.

Specific comments

Ln 17: ' with a mean error <10 m yr$^{-1}$' . The spatial distribution of error maps should be shown, and error of ice velocity should be carefully assessed using independent data.

Ln 18: 'is 1932±38'. The ice discharge estimate is obviously smaller than the previously studies. For example 2,048 ± 149 Gt/yr for Rignot et al. (2013) in Science, 2,049 ± 87 Gt/yr for Depoorter et al. (2013) in Nature. The ice discharge for 2008 is also smaller than previous studies as above. what are the causes? As mentioned in the general comments, the uncertainties were underestimated, and should be adjusted to their correct values. In addition, authors should show the differences of uncertainties using RES data or not.

Ln 19:' 35±15 Gt/yr'. As mentioned in general comments 2. The uncertainty was apparently underestimated in Table 2. The underestimated uncertainty leads directly to a certain conclusion that there is an increased mass change since ~2008. This conclusion is obviously not convincible. It may mislead the scientific community.

Ln 19:'flow accelerations across the grounding lines of West …, account for 89% of

this increase' . A quantitative assessment of the uncertainties of ice velocities and their changes is required. We can not determine where there has a significant acceleration of ice flow from Figure 8 because most of the changes are less than 50m/yr in Figure 8. So the significance of flow acceleration should be first assessed under the consideration of a large uncertainty estimate for ice velocity (mean error of 10 m/yr and as high as 20-30m yr$^{-1}$ (in Ln 63)) .

Ln 63-64 : ' as high as 20-30m/yr locally but …(see Appendix A for validation of the velocity fields).' Authors should show where is the area with the large uncertainties of

ice velocity, in other words, should show error maps for all products. How did the authors get the conclusion of 'largely uncorrelated at basin scales'? Additionally, the Appendix A didn't show any validation of ice velocity fields, except only for ice discharge.

Ln 68 'collection0 LT1 images'. In Antarctica, the majority of images are in the processing level of L1GT, not L1T, except for some region in Antarctic Peninsula. The details of Landsat processing level can be found in the site https://landsat.usgs.gov/landsat-processing-details. 'LT1' is wrong, should be L1T.

Ln 94 :'all x and y displacements that fell outside of the range … were culled from the dataset'. From the formula. It seems to all displacements were involved to estimate ice velocity, because Q3 equal closely to 95% (3sigma) and IQR equal to Q3-Q1, which closely equal to 2sigma, and T value is set to 3. Additionally, the method is possible to exclude the valid displacements, which inferred images acquired from a longer period. A longer period, and a larger displacement is expected. Furthermore, the cloud contamination is key problem in post-processing, the authors didn't show how to deal with the issue.

Ln 106:' with median velocities <10m/yr and with >100 valid retrievals'. The threshold may be set too large as reference velocity. Additionally, the use of image-pair velocity itself to define the static reference velocity fields may be problematic.

Ln 112:'have velocities <50m/yr and …'. Same as above, the reference velocity may be set to large.

Ln 120: the threshold was set too large.

Ln 186. Why did the authors use only four weighting factors?

Ln 226-227. 'We found that FG1 was the most suitable flux gate line for estimating changes…'. why did the authors use FG2 for ice discharge change in Table2.

Ln 234-235. 'We used this flux gate line to estimate absolute discharge …, but not for assessing temporal changes in discharge'. In my view, in table2, authors used the flux gate to estimate the absolute discharge and its changes.

Ln 250. Authors didn't provide the SI.

Ln 253-255. The error of grounding line could cause that ice flow don't drain outside in some nodes in the estimate of ice discharge in some areas, so that, the directions of grounding line and ice flow vectors should also be considered.

Ln 260-265. Authors should give the differences of ice discharges using GL0 and FG2 grounding lines respectively. As mentioned in general comment, in FG2 ice discharge, authors used SMB and cryostat-2 elevation change to correct the FG2 ice discharge. In my view, at first, the elevation change used to estimate the dynamic volume change is problematic. Because the elevation change do not result from the ice flow convergence, but from snowfall, firn densification, etc. secondly, the acceleration of elevation change in the gap region should be less than 10 Gt/yr because mass balance of Antarctic ice sheet is only about -70Gt/yr. The absolute ice discharge estimates in the paper is obviously smaller than those of previous studies. the possible cause for the matter is the two terms (SMB and elevation change) could

not compensated the unmeasured ice flux due to the movement inland for grounding lines.

Ln 349. 'surface elevation changes and rates of acceleration were …'. We are skeptical over how to estimate the acceleration of elevation change because the short period (from 2011 to 2015) and the acceleration must be obvious in the time series of elevation measurements, rather than the only mathematical analysis method.

Ln 351. 'the magnitude larger than ±15 m/yr were culled'. Why did the authors use the threshold?

Ln 372, see general comment 3.

Ln 444, see general comment 2.

Ln 466. Figure 7 may be wrong, should be Figure 8?

Ln 740. Figure 6. The authors used three grounding lines for ice discharges, which make us confusing. In the figure, the FG1 ice discharges were used, while FG2 ice discharges were also used. This makes it is difficult to determine which grounding line is appropriate to estimate ice discharge. Additionally, as mentioned in general comments, the conflicting results of ice discharge change in basins 5,8,12,13,14,15,23 make it is difficult which ice flow product is correct, especially, in East Antarctica.

Ln 760. Figure 7. A mis-coregistration between the L8 ice velocity and SAR-derived ice velocity is obvious because there are apparently positive/negative pattern of change in surface velocity, especially in Marguerite Bay, Getz ice shelf. The mis-coregistraton will affect the result of ice discharge and its change.

Ln780-290. Table2 . Why did the authors used only the two JPL 2015 Landsat 8 velocity maps for 2015 ice discharge estimate. Why the other results were not included. The most important thing is that the uncertainties of ice discharge changes seems to intentionally underestimate. Although the authors attempted to give an explanation in the appendix A. The uncertainties of the changes should be estimated using the uncertainties of absolute ice discharges in 2008 and 2015. The concerns have been mentioned in general comments.

The AP seems to have a positive net mass changes (+11 Gt/yr) in Table 2, because SMB value is larger than ice discharge.

Ln 860-875. The uncertainty in Flux-change estimates should be directly calculated from the uncertainties of ice flux in 2008 and 2015, rather than another method.

---

## Author Comment (AC1) · 1 Sep 2017

**Response to Anonymous Referee #1**

Thank you greatly for taking the time to provide a thorough review of our manuscript.

Please find:
 - Reviewer comments in back
 - Responses in blue
 - *Proposed changes to manuscript in italics*
* * *
General comments
The manuscript shows the present-day Antarctic-wide surface velocities using Landsat7/8 images and an assessment of mass discharge change compared to earlier ice velocity map inferred from synthetic aperture radar. The work itself is of significance for the glaciology community to help to understand the present-day situation of Antarctic ice sheet. But the manuscript do not provide new insight to scientific community.

We're sorry that you did not glean any new insights form the paper. Here we highlight some of the scientific insights that we are excited to share with the broader community:

Despite numerous papers on the mass budget of the ice sheet (e.g. *Harig and Simons*, 2015; *Shepherd et al.*, 2012; *Velicogna*, 2009; *Wouters et al.*, 2015; *Zwally et al.*, 2015) and its total discharge (*Depoorter et al.*, 2013; *Rignot et al.*, 2008; *Rignot et al.*, 2013), its change in ice flow has not yet been directly measured on a continental scale. We feel that this in itself is a significant result as it provides the first comprehensive measurement of the dynamic response of the ice sheet that is primarily the result of changes in the rate of ocean melting and/or changes in buttressing (ice shelf back stress). Such a result allows for the disaggregation of the plethora of mass change results into the primary mechanisms of change: ice flow (ocean influenced) and precipitation (atmosphere influenced). All earlier studies attempting to do this (e.g. Harig and Simons, 2015; *Shepherd et al.*, 2012) have had to rely on large assumptions to infer such separation.

Our analysis reveals several previously undocumented and large changes in ice sheet flow. One major finding is that the glaciers feeding into the Getz Ice shelf have accelerated in response to recent ice shelf thinning and are now losing mass at a rapid rate as a result of a sharp increase ice discharge, this has not been documented elsewhere. Or analysis also reveals that the fastest speed-up of any Antarctic glacier is observed for the set of glaciers feeding into Marguerite Bay and is highly suggestive of enhanced ocean forcing in that area. Likely the most significant and novel result is our finding of a highly stable East Antarctic Ice Sheet over the period of study with virtually no change in discharge between 2008 and

2015. While recent stability has been inferred from gravity and volume change studies it has never been measured directly. Our study provides strong evidence that despite large glacier response to changes in ocean in the Amundsen Sea sector the East Antarctic has seen none. On top of all of this we also confirm the continued slowing in the rate of acceleration of the Pine Island and acceleration of Thwaites glaciers previously documented by *Mouginot et al.* (2014), that Totten glacier has not seen any recent increase in discharge as previoulsy proposed by *Li et al.*, 2016), and that the much discussed Bellingshausen Sea sector of the ice sheet has experienced only highly localized increases in flow with little change in total ice discharge over the 2008-2015 period in close agreement with *Hogg et al.*, 2017.

It is our opinion that these results will be of interest to the atmospheric and snow communities that study changes in Antarctic precipitation and to the oceanic community that is interested in studying the response of the ice sheet to changes in ocean circulation and temperature. This result is also invaluable to the ice sheet modeling community whose focus is on projecting changes in ice flow into the future.

I have five major concerns in the matter.

1. There is another manuscript in discussion in TC submitted earlier. Both papers are discussing the same issue. Although some results seem to be similar, my concern is that the both seem to draw different conclusions in term of ice discharge change in Antarctic ice sheet. The causes of the differences should be discussed in details. Shen, Q., Wang, H., Shum, C.-K., Jiang, L., Hsu, H. T., and Dong, J.: Antarctic high-resolution ice flow mapping and increased mass loss in Wilkes Land, East Antarctica during 2006–2015, The Cryosphere Discuss., and https://doi.org/10.5194/tc-2017-34

We closely followed the submission by Shen et al. who did a similar analysis to the one presented here. That paper was recently rejected so we do not feel that it is appropriate to comment on the specifics of their findings. That said, large differences in flux estimates typically result from:

   a. Differences in the velocity fields. Despite using much of the same imagery to generate the offset fields, Landsat velocity fields are highly sensitive to outlier rejection criteria and to the determination of the image pair geolocation corrections.

   b. Definition of the flux gate. Estimates of total flux are sensitive to the definition of the flux-gate cross-section. In particular, errors grow quickly with increasing reliance on interpolated estimates of ice thickness.

Access to the component velocities and the flux gate definitions used to determine the flux are required to identify the exact cause of the discrepancy, this can not be determined from the results presented in the paper alone.

2. The estimation of uncertainties of ice discharge changes were not rigorously based on error propagation law by an intentional and non-scientific method so that small estimates were obtained. Accordingly, uncertainties of ice discharge changes were obviously underestimated (see Table 2). For example, the uncertainty for all of Antarctica should be ±56 Gt/yr (sqrt(41^2+38^2)), not ± 15 Gt/yr (see Table 2 ) as stated in the paper. The uncertainties for individual basins and sectors (East Antarctica, West Antarctica and AP) were also underestimated. The estimate of the ice discharge change in the Antarctica should be 35±56 Gt/yr. Therefore, it is incorrect to conclude that there was a certain increased ice discharge since ~2008. There are no significant acceleration of ice discharge in West Antarctica using a correct uncertainty estimate for ice discharge change, rather than increase ice discharge as mentioned in the title.

We can understand the Reviewer's confusion. The appropriate propagation of errors in geophysics is non-trivial task and requires some degree of understanding of the correlation of each error term that are not always well known. For two fully independent estimates of mass flux, having normally distributed and random errors, the difference between the two terms is simply the Root Sum of Squares (RSS) of the individual errors, as suggested by the reviewer.  However, when a single definition of ice thickness is used, estimates of flux have large systematic errors that mostly cancel when differenced.

Here we provide an illustrative example of why errors should not be propagated as suggested by the reviewer.

The change in flux between times t1 and t2 and its uncertainty are a function of the change in the velocity and not the velocity magnitudes. Assuming an ice thickness of 1.0 km +/- 0.1km, a gate width of 1km, and that ice thickness is the only source of uncertainty, a depth averaged velocity increase of 1 km/yr normal to the gate cross-section would result in a 1km$^3$/yr. increase in flux with an uncertainty of +/- 0.1 km$^3$/yr .

Now if the velocity at t1 and t2 were 10km/yr. and 11km/yr., respectively, the total flux at t1 would be 10 +/- 1 km3/yr.  and 11 +/- 1.1 km3/yr. at t2. Taking the RSS of the flux magnitude uncertainties (sqrt(1.0^2 + 1.1^2) = 1.5 km$^3$/yr) overestimates the error in the change in flux by an order of magnitude.

We detail the propagation of errors in Appendix A with uncertainty in the changes in discharge specifically detailed in Section 1.2A. Hopefully this example helps to explain why errors for the change in discharge are smaller than errors in total discharge, a concern revisited by the reviewer throughout his/her comments.

3. The paper stated that in the calculation of ice discharge, the uncertainties were apparently reduced due to the extensive use of RES data. But I do not think that the use of RES can really reduce the uncertainties. At first, the uncertainties of dynamic

volume and surface mass balance were not shown in the tables of the manuscript. In general, the uncertainty of firn densification model is relatively large during the transfer between elevation change and mass change but was not shown. Additionally, the elevation change was directly considered as the dynamic volume which is problematic, because there are many driven factors of elevation change of ice glacier/sheet, for example, firn densification, the snowfall change, basal melting etc. The surface mass balance is another large error source to the uncertainty of ice discharge using the FG2. Furthermore, the small estimates of uncertainty may result from the large number of statistical units as much as 27. For example, in the paper, the uncertainty of total ice discharge were calculated based on 27 basins, while Depoorter et al. (Nature, 2013) estimated the uncertainty based on six oceanic sectors, and Rignot et al. (Science 2013) summed the uncertainties of each calculation units (ice shelf). More importantly, calculation of ice discharge is highly sensitive to the definition of the flux gate, the intentional movement of grounding line could cause the over 20% error in individual ice discharge even if the RES data are used. Therefore, the method for the calculation of ice discharge needs to be rigorously validated before use although this method was previously proposed by other authors.

OK, there is a lot to cover in the response to this comment. To be clear we do our best to account for errors resulting from uncertainties in surface mass balance, firn air content (depth averaged density), change in firn air content, surface velocity, elevation change, and the assumption that surface velocities equal depth averaged velocities. We try here to address each of the reviewer's specific comments:

I do not think that the use of RES can really reduce the uncertainties
Use of Radar Echo Sounding (RES) measurements to define ice thickness greatly reduce errors in flux gate areas relative to interpolated estimates such as BedMap2 (cf. *Li et al.*, 2016; *Rignot and Kanagaratnam*, 2006; *Rignot et al.*, 2011). Improvements come from lower errors in estimates of ice thickness (as shown Figure 4b) and higher resolution data that reduces resolution dependent systematic biases in flux (as shown in Figure 5).

At first, the uncertainties of dynamic volume and surface mass balance were not shown in the tables of the manuscript. In general, the uncertainty of firn densification model is relatively large during the transfer between elevation change and mass change but was not shown.  Additionally, the elevation change was directly considered as the dynamic volume which is problematic, because there are many driven factors of elevation change of ice glacier/sheet, for example, firn densification, the snowfall change, basal melting etc.

All error sources mentioned by the reviewer were taken into account in the initial submission. One point of note is that in applying the mass conservation approach we only rely on estimates of surface mass budget, dynamic thinning, and firn compaction for the area between the grounding line and the upstream flux gate. While uncertainties in surface mass balance (20%), elevation change (0.1 m yr$^{-1}$ or

30% of the rate of change, whichever is larger) and firn change (see Appendix) are large; estimates are integrated over a small total area (6% of the total area of the ice sheet). To address reviewer's point that not all elevation change measured between the flux gate and the grounding line can be attributed dynamic volume change is valid. To deal with this we make the assumption that only elevation change occurring over ice with a velocity > 200m/y is counted as dynamic volume change. Overall this correction is very small and insensitive to the cutoff velocity.

*For resubmission we will add an additional table to the Appendix detailing all error terms included in the analysis. We hope that this will address the review's criticisms and make our propagation of errors more transparent.*

The surface mass balance is another large error source to the uncertainty of ice discharge using the FG2. Furthermore, the small estimates of uncertainty may result from the large number of statistical units as much as 27. For example, in the paper, the uncertainty of total ice discharge were calculated based on 27 basins, while Depoorter et al. (Nature, 2013) estimated the uncertainty based on six oceanic sectors, and Rignot et al. (Science 2013) summed the uncertainties of each calculation units (ice shelf).

We felt that an error in surface mass balance of 20% that was fully correlated (systematic) within each basin was a conservative estimate. Unfortunately there is no definitive practice for estimating and propagating modeled SMB errors as there are not enough in situ observations to rigorously quantify uncertainties in SMB over the large scales relevant to this study. To calculate errors appropriately some assumption must be made as to the correlation length of the modeled estimates. Depoorter er al. 2014 estimated the error for an earlier version of RACMO2 over ice shelves by comparing to in situ observations (primarily over Ross and Filchner-Ronne) with no assessment of correlation length. In that analysis they determined an average "local" ice shelf SMB uncertainty of 28%, slight larger than the 20% grounded ice uncertainty applied here. For unsurveyed grounded ice basins (see Table S1 of their manuscript) they assign an 11% uncertainty to the climatological SMB for each of their 6 sectors. The study of *Rignot et al.,* 2008 used output from an earlier version of RACMO2 for which they estimated absolute errors in accumulation varying from 10% in dry, large basins to 30% in wet, small coastal basins. In that study the Antarctic wide error was estimated to be 6% (compared to 5% estimated in this study). *Rignot et al.,* 2013 also use output from an older version of RACMO2 and report basin scale errors ranging from 7% to 25%, with an Antarctic wide error of 14%. *Shepherd et al.,* 2012 report modeled SMB errors of 5% to 20% depending on basin size and location. Considering the improvements in modeled Antarctic SMB by RACMO2.3 (*van Wessem et al.,* 2014) and the inclusion of higher resolution output for the Antarctic Peninsula (*van Wessem et al.,* 2016) we feel that the errors applied in this study are consistent with earlier studies and well justified.

More importantly, calculation of ice discharge is highly sensitive to the definition of the flux gate, the intentional movement of grounding line could cause the over 20% error in individual ice discharge even if the RES data are used. Therefore, the method for the calculation of ice discharge needs to be rigorously validated before use although this method was previously proposed by other authors.

Our estimates are not as sensitive to the position of the flux gate as one might initially expect since we apply a mass conserving approach when extrapolating measured flux upstream of the grounding line to estimates of discharge across the grounding line. In fact, such an approach can greatly reduce errors in discharge when uncertainties in ice thickness near the grounding line are large (cf. *Li et al.*, 2016; *Rignot and Kanagaratnam*, 2006; *Rignot et al.*, 2011). These basic principles are the same as those used for reconstruction of basal topography (*Morlighem et al.*, 2011). The size of the error is solely dependent on how well the mass flux terms (primarily SMB) can be quantified between the fluxgate and the grounding line. Since the flux gate used in this study is located in close proximity to the grounding line, for most basins, errors associated with the estimation of SMB and other flux terms are smaller than the uncertainties introduced by poorly known basal topography. Taking this approach reduces the uncertainty in the total flux estimate by 64%.

*We will include an additional table in the Appendix detailing the individual errors and the magnitudes.*

4. The authors (and also Shen et al. in review) used first Antarctic-wide ice velocity (Rignot et al. 2011, science) as a reference map, the reference year are 2008 and 2006 respectively. The MEaSUREs Antarctic ice velocity map (v1.1) was inferred from over a long period (1996-2009) according its production statements. The data were acquired as early as 1996. Therefore the SAR-derived ice velocity map as a single year is problematic. The new MEaSUREs products have released and annual maps from 2005 to 2016 can be obtained (http://nsidc.org/data/nsidc-0720), authors should use the new products to alleviate the problem prompted.

Thank you for pointing this out. This dataset was made available after our original submission. Now that it is public we have assessed the implications of using the older dataset in our calculations and summarize our findings here.

These new data come with more precise time stamps but at the expense of reduced horizontal resolution (1km vs. 450), reduced spatial coverage and larger uncertainties. To ensure that our stated time period of cira-2008 is appropriate we resample (linear interpolation) the original MEASURES radar mosaic to 1km and compare to the error averaged 2007_2008 and 2008_2009 velocities from the new dataset. Differences are less than 2 Gt/yr. for all basins except for Basins 12, 13, and 14 that differ by -4, -5 and -6 Gt/yr. respectively and basin 24 by -4 Gt/yr. Some of the difference can be attributed to real differences in flow for differences in temporal sampling but also from differences in uncertainties between products (the

original MEASURES mosaic having lower errors, particularly for the East Antarctic) and from differences in horizontal resolution.

From this analysis we concluded that the best estimate of flux for the ~2008 period is still produced by the earlier MEASURES mosaic (higher spatial resolution and the lower uncertainty) that is derived from the same underlying data contained in the annual mosaics. We also determine the period "circa-2008" well characterizes the effective date of the earlier MEASURES mosaic. This data has been used previously to estimate total Antarctic discharge in Rignot et al. 2013 with a reference date of 2007 to 2008 and in *Depoorter et al.*, 2013 with a reference date of 2007 to 2009.

*We will include a paragraph in the revised manuscript summarizing our analysis.*

5. The authors used different products of ice velocity (M14/15, W14/15, L750 and L124) to estimate the change in flux across FG1, but the values are apparently different. For example, there are conflicting estimates of ice discharge changes in basins 8, 12, 13, 14. In particular, the discharge is decreased for the M14/15 while it is increased for W14/15. Noted that they used the same data, only difference is that the mosaicking methods. Unfortunately, for the choice of the velocity data, the author did not present any convincible standards. The accuracy of ice velocity products should be carefully assessed using the independent surveyed data.

Yes, there are clearly differences between mappings. Much of the difference can be attributed to product errors. As shown in Figure A2, the 2015 mosaics have the lowest uncertainties (used in this study), followed by the 2014 mosaics with the LISA products have the highest uncertainties (See Figure A2). Some difference between mappings can also be expected due to real changes in ice flow between effective dates of each map. Even so, the standard deviation between all flux change estimates is below the stated uncertainty in discharge listed in Table 2 for all 27 basins.

*In the revised manuscript we will include a sentence or two in the Figure 6 caption to this effect. We also noticed that we failed to include a legend for the Figure 6 bar plots. We will make sure to include this in the revised manuscript.*

Specific comments:
Ln 17: ' with a mean error <10 m yr-1' . The spatial distribution of error maps should be shown, and error of ice velocity should be carefully assessed using independent data.

While spatially distributed errors are informative they can be misleading without knowledge of the spatial correlation. Spatial errors are provided with MEASURES mosaic but the documentation states that "Error estimates for the velocity magnitude are located in the variable err; however these values should be used more as an indication of relative quality rather than absolute error". The most

relevant metric to qualitatively assess error in the Landsat products is image pair count, which are displayed in Figure 1.

Most important for our study is the assessment of errors in velocity along fluxgates and how these errors are correlated with distance along the gate. For this we compare Landsat velocities (all 6 mosaics) to Radar velocities (MEASURES version 1) over East Antarctica. We make the conservative assumption that all differences can be attributed to measurement error (i.e. assuming no real change in velocity between mappings). The results of this analysis are shown in Figure A2. We conclude that point-scale differences are on the order of 30m/yr. but quickly decrease with averaging distance.

*In the revised manuscript we will add a note that the image pair count can be used as a relative metric to judge quality of the velocity field. And that the assessment presented in Appendix A provides a more comprehensive assessment of absolute errors when assessing velocity change. We will also include a more explicit description of the validation analysis in the main manuscript instead of just in the Appendix.*

Ln 18: 'is 1932±38'. The ice discharge estimate is obviously smaller than the previously studies. For example 2,048 ± 149 Gt/yr for Rignot et al. (2013) in Science, 2,049 ± 87 Gt/yr for Depoorter et al. (2013) in Nature. The ice discharge for 2008 is also smaller than previous studies as above. what are the causes? As mentioned in the general comments, the uncertainties were underestimated, and should be adjusted to their correct values. In addition, authors should show the differences of uncertainties using RES data or not.

Rignot et al. (2013) used the same Radar velocity mosaic combined with Operation Ice Bridge and BEDMAP-2 ice thickness data at InSAR derived grounding lines to determine a total Antarctic grounding line flux of 2048 +/- 146 Gt/yr with upscaling accounting for 352 Gt/yr. of the total flux. This compares to 1897+/-41 Gt/yr. presented in this study. We should fist note that estimates agree within stated errors. The most obvious reason for the difference in the central estimates is the definition of the flux gates. Rignot et al. (2013) mostly rely on BEDMAP-2 data while our study draws almost entirely from flight data. Another possible reason for the difference is the upscaling of results for unmeasured basins. For these basins the total flux is assumed to be the modeled climatological average surface mass balance integrated over the upstream basin. Such estimates have not been adjusted for losses due to basal melt, are sensitive to errors in the modeled SMB and to the delineation of the contributing basin area over which SMB is integrated.

Depoorter et al. (2013) estimate a total groundingline flux of 2049 +/- 86 Gt/yr. with up scaling for unmeasured areas (same as in Rignot et al. 2013) accounting for 476 Gt/yr. This study uses a different definition of groundling but otherwise uses the same data as used in Rignot et al., 2013. This estimate is significantly higher than ours. Again the definition of ice thickness and upscaling to unmeasured basins likely accounts most of the difference.

It should also be noted that Depoorter et al. (2013) and Rignot et al. (2013) both used output from an earlier version of RACMO that produced larger total SMB than the version of the model used in our study. Since SMB is used to upscale flux, this likely contributes some to the larger flux estimates. Similar conclusions were made for updated Greenland Ice Sheet discharge estimates that were lower than previous estimates (*Enderlin et al.*, 2014).

*In the revised manuscript we will include a an additional paragraph discussing differences between our estimate of total discharge and those of Depoorter et al. (2013) and Rignot et al. (2013) along with likely sources of the discrepancy.*

Ln 19:' 35±15 Gt/yr'. As mentioned in general comments 2. The uncertainty was apparently underestimated in Table 2. The underestimated uncertainty leads directly to a certain conclusion that there is an increased mass change since ~2008. This conclusion is obviously not convincible. It may mislead the scientific community.

Hopefully our response to general comment 2 has better explained our approach to the propagating errors.

Ln 19:'flow accelerations across the grounding lines of West ..., account for 89% of this increase' . A quantitative assessment of the uncertainties of ice velocities and their changes is required. We can not determine where there has a significant acceleration of ice flow from Figure 8 because most of the changes are less than 50m/yr in Figure 8. So the significance of flow acceleration should be first assessed under the consideration of a large uncertainty estimate for ice velocity (mean error of 10 m/yr and as high as 20-30m yr-1 (in Ln 63)) .

A quantitative assessment of velocities and correlation lengths is provided in Figure A2. All error terms are defined in Appendix A and are propagated according to the equations presented in Appendix A.

*To make our propagation of errors more transparent we will include an additional table in the Appendix detailing the individual errors and their magnitudes.*

Ln 63-64 : ' as high as 20-30m/yr locally but ...(see Appendix A for validation of the velocity fields).' Authors should show where is the area with the large uncertainties of ice velocity, in other words, should show error maps for all products. How did the authors get the conclusion of 'largely uncorrelated at basin scales'? Additionally, the Appendix A didn't show any validation of ice velocity fields, except only for ice discharge.

Please see response to Specific comment "Ln 17"

Ln 68 'collection0 LT1 images'. In Antarctica, the majority of images are in the processing level of L1GT, not L1T, except for some region in Antarctic Peninsula. The details of Landsat processing level can be found in the site https://landsat.usgs.gov/landsat-processing-details. 'LT1' is wrong, should be L1T.

Thank you fro catching this.

*We will update appropriately in the revised manuscript.*

Ln 94:'all x and y displacements that fell outside of the range ... were culled from the dataset'. From the formula. It seems to all displacements were involved to estimate ice velocity, because Q3 equal closely to 95% (3sigma) and IQR equal to Q3-Q1, which closely equal to 2sigma, and T value is set to 3. Additionally, the method is possible to exclude the valid displacements, which inferred images acquired from a longer period. A longer period, and a larger displacement is expected. Furthermore, the cloud contamination is key problem in post-processing, the authors didn't show how to deal with the issue.

Our use of the word "displacements" was incorrect. Interquartile Range filtering was applied to time normalized displacements (i.e. velocities).

Time normalized displacements are filtered using 3 then 1.5 times the inter quartile range (IQR = Q3 - Q1). If the data is normally distribute the IQR $\cong$ 1.3 sigma. This means that we reject data outsize of the 4- then 2- sigma range. This approach removes ~8-10% of the data. This filtering strategy is aggressive due to the low SNR of the dataset. We hope this addresses the reviewers concerns.

For cloud filtering, or more generally for filtering areas of without matchable features, we apply a Normalized Displacement Coherence (NDC) Filter.

*In the revised manuscript we will replace "displacements" with "velocities" we will also provide a description of the cloud filtering.*

Ln 106:' with median velocities <10m/yr and with >100 valid retrievals'. The threshold may be set too large as reference velocity. Additionally, the use of image-pair velocity itself to define the static reference velocity fields may be problematic.

The reference velocity is assumed to be moving at a constant rate (not changing over time), it is not assumed to be stagnant. For this assumption to cause errors in our velocity mosaic there would need to be large areas of ice with very slow velocities that either had large trends or secular changes in velocity. We are not aware of any evidence that this assumption should cause concern for the Antarctic Ice Sheet. *Mouginot et al.*, 2017 use 20% of the lowest velocities within each displacement field. Given the vast area of Antarctic ice moving at >10 m/yr. this assumption is likely not radically different from our approach.

Ln 112:'have velocities <50m/yr and ...'. Same as above, the reference velocity may be set to large.
We find no evidence that this approach introduces significant error into the auto-RIFT velocity mosaic in a way that affects our results. This was determined through comparison to NSIDC's LISA mosaics (Figure 6), which does not adopt this criterion, and to MEASURES velocities over the East Antarctic Ice Sheet (Figure A2).

Ln 120: the threshold was set too large.
Please see response to previous comment

Ln 186. Why did the authors use only four weighting factors?
This was done for simplicity. Nearly all data used in the LISA mosaic is from image pairs separated in time by 64 days or less. As such this simplification has negligible impact on the final velocity fields.

Ln 226-227. 'We found that FG1 was the most suitable flux gate line for estimating changes...'. why did the authors use FG2 for ice discharge change in Table2. Ln 234-
Please see next comment (Ln 235)

235. 'We used this flux gate line to estimate absolute discharge ..., but not for assessing temporal changes in discharge'. In my view, in table2, authors used the flux gate to estimate the absolute discharge and its changes.
We see how this can be confusing. FG2 provides the cross-sectional area with the lowest uncertainty and is most appropriate for estimating the total discharge, even after having to account for additional mass input between the gate and the grounding line. FG1 strikes a balance between proximity to the grounding line (GL0) and the distance from ice thickness observations. This gate is best suited for estimating changes in ice discharge. Our best estimate of total discharge is computed using the 2015 autoRIFT velocities, FG2 and estimated mass flux between FG2 and GL0. We then compute the change in discharge between the 2015 and 2008 period at FG1 and subtract this form our best estimate of total discharge, accounting for dynamic volume change and changes in ice thickness between periods. Taking this approach we reduce errors in estimates of ice discharge by 64% compared to estimating ice discharge at the grounding line (GL0).

*For the revision we will add clarifying text describing the suitability of the three gate definitions for determining ice discharge.*

Ln 250. Authors didn't provide the SI.
In the original submission we did not include shapefiles with fluxgate definitions and attributes. We will make sure to include these with the final submission.

Ln 253-255. The error of grounding line could cause that ice flow don't drain outside in some nodes in the estimate of ice discharge in some areas, so that, the directions of grounding line and ice flow vectors should also be considered.

As long as flux is calculated using the component velocities (vx and vy), the ice flow vector is not needed. Here we defined the flux gate following polygon convention with the upstream side of the flux gate being defined as to the right hand side of the polygon gate vector as one move from node n to node n+1. The flux in the x-direction is then simply the flow in the x-direction multiplied by the width of the gate projected on the x-axis and the ice thickness all multiplied by the direction sign of the flux gate. The flux in the y direction is calculated following the same approach. The total flux is then the sum of the flow in the x and y directions.

*In the revised manuscript we will expand the equation shown on L255 to include the summation of the component velocities as discussed here.*

Ln 260-265. Authors should give the differences of ice discharges using GL0 and FG2 grounding lines respectively. As mentioned in general comment, in FG2 ice discharge, authors used SMB and cryostat-2 elevation change to correct the FG2 ice discharge. In my view, at first, the elevation change used to estimate the dynamic volume change is problematic. Because the elevation change do not result from the ice flow convergence, but from snowfall, firn densification, etc. secondly, the acceleration of elevation change in the gap region should be less than 10 Gt/yr because mass balance of Antarctic ice sheet is only about -70Gt/yr. The absolute ice discharge estimates in the paper is obviously smaller than those of previous studies. the possible cause for the matter is the two terms (SMB and elevation change) could not compensated the unmeasured ice flux due to the movement inland for grounding lines.

We are not confident that we follow the reviewers comment. Accounting for dynamic volume change and SMB between the upstream gate and the grounding line does introduce additional error into our estimate of ice discharge. Since these corrections are integrated over relatively small areas, their contribution to the total discharge error term is relatively small for most basins. It is also true that ice sheet volume change is a combination of dynamic mass change, surface mass balance anomalies and changes in the firn air content. We have separated dynamic volume change from SMB related volume change by applying a 200 m/yr. threshold to our CryoSat-2 elevation change results. Because this is an imperfect assumption we add a large uncertainty to the dynamic volume change of (0.1 m yr$^{-1}$ or 30% of the correction, whichever is larger).

*In the revised manuscript we will provide an additional table in the Appendix listing all correction magnitudes and associated uncertainties for all basins. We will also include a comparison of total discharge estimates using the three different flux gate definitions.*

Ln 349. 'surface elevation changes and rates of acceleration were ...'. We are skeptical over how to estimate the acceleration of elevation change because the short period (from 2011 to 2015) and the acceleration must be obvious in the time

series of elevation measurements, rather than the only mathematical analysis method.

We do not have altimetry data for the 2008-2010 period so we need to extrapolate rates of elevation change determined from CryoSat-2 data to the period of study. Our choice is to either apply the constant rate measured over the 2010-2015 period or to include some estimate of the change in rate through time (acceleration). Here we chose to apply a linear rate of acceleration since acceleration is expected in areas of rapid dynamics. Including an acceleration term has negligible impact on our total flux estimates since data is only being extrapolated over the 2-year period from 2008-2010. We hope that this better justifies our use of an acceleration term when estimating the dynamic volume change.

Ln 351. 'the magnitude larger than ±15 m/yr were culled'. Why did the authors use the threshold?

This is done to remove gross outliers (*Nilsson et al.*, 2016). 5-year elevation changes rates are not expected to exceed this threshold.

Ln 372, see general comment 3. Ln 444, see general comment 2.
Response provided above.

Ln 466. Figure 7 may be wrong, should be Figure 8?
Thank you for spotting this.

*We will change this to Figure 8 in the revised manuscript.*

Ln 740. Figure 6. The authors used three grounding lines for ice discharges, which make us confusing. In the figure, the FG1 ice discharges were used, while FG2 ice discharges were also used. This makes it is difficult to determine which grounding line is appropriate to estimate ice discharge. Additionally, as mentioned in general comments, the conflicting results of ice discharge change in basins 5,8,12,13,14,15,23 make it is difficult which ice flow product is correct, especially, in East Antarctica.

We see how this can be confusing. FG2 provides the cross-sectional area with the lowest uncertainty and is most appropriate for estimating the total discharge, even after having to account for additional mass input between the gate and the grounding line. FG1 strikes a balance between proximity to the true grounding line (GL0) and the distance from ice thickness observations. This gate is best suited for estimating changes in ice discharge. Our best estimate of total discharge is computed using the 2015 autoRIFT velocities, FG2 and computed mass fluxes between FG2 and GL0. We then compute the change in discharge between the 2015 and 2008 period at FG1 and subtract this form our best estimate of total discharge, accounting for dynamic volume change and changes in ice thickness. This is why results for FG1 are shown in Figure 6 (assessment of change in discharge). Taking

this approach greatly reduces errors in estimates of total discharge (error of 5.6% for *GL0*, 4.4% for *FG1*, and 2.0% for *FG2*). GL0 is only used to determine the area between the gate and the grounding line for which corrections need to be applied. Temporal changes in the position of the groundling line only affect the area for which flux gate to grounding line corrections are determined. For the 7-year period of this study changes in in grounding line position have negligible impact on our results.

Our response to general comment 5 addresses the rest of these concerns.

*For the revision we will add clarifying text describing the suitability of the three gate definitions for determining ice discharge.*

Ln 760. Figure 7. A mis-coregistration between the L8 ice velocity and SAR-derived ice velocity is obvious because there are apparently positive/negative pattern of change in surface velocity, especially in Marguerite Bay, Getz ice shelf. The mis-coregistraton will affect the result of ice discharge and its change.

Geolocation errors that most likely originate in the radar data introduce noise into our analysis but are unlikely to significant biases our estimates of flux or flux change because: 1. Errors will somewhat cancel when integrated across the entire glacier cross section (speedup has corresponding slowdown on opposite side of glacier), 2. flux-gate nodes located in problematic areas are assigned a Landsat flux value and are assumed to be constant between time periods (see Figure 2), 3. Geolocation errors are represented in our assessment of the velocity error (see Figure A2).

*We will include a statement to this effect in the revised manuscript.*

Ln780-290. Table2 . Why did the authors used only the two JPL 2015 Landsat 8 velocity maps for 2015 ice discharge estimate. Why the other results were not included. The most important thing is that the uncertainties of ice discharge changes seems to intentionally underestimate. Although the authors attempted to give an explanation in the appendix A. The uncertainties of the changes should be estimated using the uncertainties of absolute ice discharges in 2008 and 2015. The concerns have been mentioned in general comments.

Hopefully our response to general comments 2 & 5 has addressed these concerns.

The AP seems to have a positive net mass changes (+11 Gt/yr) in Table 2, because SMB value is larger than ice discharge.
Knowledge of ice thickness, circa-2008 ice velocities and SMB all have large uncertainties for the AP. For this reason we do not use estimates of flux and SMB to determine the mass balance of the AP and instead rely on earlier estimates derived from ice volume change estimates (*Scambos et al.*, 2014) that we correct for measured changes in discharge. Our assumption of stable rates of mass loss for the AP is supported by repeat gravity measurements from the GRACE satellites as

presented in Figure B1. For a detailed description of how the AP mass change was estimated we refer the reader to Appendix B.

*We will include clarifying text in the caption of Table 2 to address this confusion.*

Ln 860-875. The uncertainty in Flux-change estimates should be directly calculated from the uncertainties of ice flux in 2008 and 2015, rather than another method. Hopefully our response to general comment 2 has better explained our approach to the propagating errors.

References:

Depoorter, M. A., J. L. Bamber, J. A. Griggs, J. T. M. Lenaerts, S. R. M. Ligtenberg, M. R. van den Broeke, and G. Moholdt (2013), Calving fluxes and basal melt rates of Antarctic ice shelves, *Nature*, *advance online publication*, doi:10.1038/nature12567.

Enderlin, E. M., I. M. Howat, S. Jeong, M.-J. Noh, J. H. van Angelen, and M. R. van den Broeke (2014), An improved mass budget for the Greenland ice sheet, *Geophys. Res. Lett.*, *41*(3), 866-872, doi:10.1002/2013GL059010.

Harig, C., and F. J. Simons (2015), Accelerated West Antarctic ice mass loss continues to outpace East Antarctic gains, *Earth Planet. Sci. Lett.*, *415*, 134-141, doi:http://dx.doi.org/10.1016/j.epsl.2015.01.029.

Hogg, A. E., et al. (2017), Increased ice flow in Western Palmer Land linked to ocean melting, *Geophys. Res. Lett.*, *44*(9), 4159-4167, doi:10.1002/2016GL072110.

Li, X., E. Rignot, J. Mouginot, and B. Scheuchl (2016), Ice flow dynamics and mass loss of Totten Glacier, East Antarctica, from 1989 to 2015, *Geophys. Res. Lett.*, *43*(12), 6366-6373, doi:10.1002/2016GL069173.

Morlighem, M., E. Rignot, H. Seroussi, E. Larour, H. Ben Dhia, and D. Aubry (2011), A mass conservation approach for mapping glacier ice thickness, *Geophys. Res. Lett.*, *38*(19), n/a-n/a, doi:10.1029/2011GL048659.

Mouginot, J., E. Rignot, B. Scheuchl, and R. Millan (2017), Comprehensive Annual Ice Sheet Velocity Mapping Using Landsat-8, Sentinel-1, and RADARSAT-2 Data, *Remote Sensing*, *9*(4), 364.

Nilsson, J., A. Gardner, L. Sandberg Sørensen, and R. Forsberg (2016), Improved retrieval of land ice topography from CryoSat-2 data and its impact for volume-change estimation of the Greenland Ice Sheet, *The Cryosphere*, *10*(6), 2953-2969, doi:10.5194/tc-10-2953-2016.

Rignot, E., J. L. Bamber, M. R. van den Broeke, C. Davis, Y. Li, W. J. van de Berg, and E. van Meijgaard (2008), Recent Antarctic ice mass loss from radar interferometry and regional climate modelling, *Nature Geosci*, *1*(2), 106-110.

Rignot, E., S. Jacobs, J. Mouginot, and B. Scheuchl (2013), Ice-shelf melting around Antarctica, *Science*, *341*(6143), 266-270.

Rignot, E., and P. Kanagaratnam (2006), Changes in the Velocity Structure of the Greenland Ice Sheet, *Science*, *311*(5763), 986-990, doi:10.1126/science.1121381.

Rignot, E., I. Velicogna, M. R. van den Broeke, A. Monaghan, and J. Lenaerts (2011), Acceleration of the contribution of the Greenland and Antarctic ice sheets to sea level rise, *Geophys. Res. Lett.*, *38*(5), L05503, doi:10.1029/2011gl046583.

Scambos, T. A., E. Berthier, T. Haran, C. A. Shuman, A. J. Cook, S. R. M. Ligtenberg, and J. Bohlander (2014), Detailed ice loss pattern in the northern Antarctic Peninsula: widespread decline driven by ice front retreats, *The Cryosphere*, *8*(6), 2135-2145, doi:10.5194/tc-8-2135-2014.

Shepherd, A., et al. (2012), A Reconciled Estimate of Ice-Sheet Mass Balance, *Science*, *338*(6111), 1183-1189, doi:10.1126/science.1228102.

van Wessem, J. M., et al. (2016), The modelled surface mass balance of the Antarctic Peninsula at 5.5 km horizontal resolution, *The Cryosphere*, *10*(1), 271-285, doi:10.5194/tc-10-271-2016.

van Wessem, J. M., et al. (2014), Improved representation of East Antarctic surface mass balance in a regional atmospheric climate model, *J. Glaciol.*, *60*(222), 761-770, doi:10.3189/2014JoG14J051.

Velicogna, I. (2009), Increasing rates of ice mass loss from the Greenland and Antarctic ice sheets revealed by GRACE, *Geophys. Res. Lett.*, *36*(19), L19503, doi:10.1029/2009gl040222.

Wouters, B., A. Martin-Español, V. Helm, T. Flament, J. M. van Wessem, S. R. M. Ligtenberg, M. R. van den Broeke, and J. L. Bamber (2015), Dynamic thinning of glaciers on the Southern Antarctic Peninsula, *Science*, *348*(6237), 899-903, doi:10.1126/science.aaa5727.

Zwally, H. J., J. Li, J. W. Robbins, J. L. Saba, D. Yi, and A. C. Brenner (2015), Mass gains of the Antarctic ice sheet exceed losses, *J. Glaciol.*, *61*(230), 1019-1036, doi:10.3189/2015JoG15J071.

---

## Referee Comment (RC2) · Anonymous Referee #2 · 19 Sep 2017

This study is a useful benchmark contribution to Antarctic mass balance research in particular because it uses very large data archives to achieve high-resolution, near-comprehensive coverage and improved uncertainty reduction in ice flow measurements. In this way (and with updated SMB products and a new approach to flux gate comparisons), it compliments and improves upon earlier pioneering continent-scale flux studies, and allows recent flux changes to be calculated. As such, it marks a maturation in mass-balance auditing and points the way towards the regular, operational big-data measurement of Antarctic mass change.

Specific comments: Most of my queries have been covered in the authors' responses to the other review.

In title and throughout: I suggest avoiding using the term stability (or stable, re-

stabilization) to mean unchanging flux because stability has other particular connotations for ice sheet mass balance.

Abstract: I suggest rewording the final sentences, e.g. "The modest increase in ice discharge over the past 7 years but ongoing high rates of ice sheet mass loss and distinct patterns of elevation lowering suggest that the recent pattern of mass loss in Antarctica is part of a longer-term phase of enhanced glacier flow initiated in the decades leading up to the first continent-wide radar mapping of ice flow."

For the uncertainty associated with the assumption of surface velocity being equal to depth-averaged velocity ($\sigma$F v-bar), the authors convincingly explain that this term is small, however it is a bias term of a particular sign which suggests that it should be corrected for or otherwise added to one side of the uncertainty range rather than being combined in quadrature.

Section 3.1.2: line 405, replace 'certainty' with 'confidence'. Can the authors please be more specific in this section – do they consider the 56 Gt to be incorrect and the real imbalance to be close to zero?

Figure 4 caption: Please clarify the y-axis units.

There are two Figure 6s. The second one (now Figure 7) needs a legend and also more discussion of the range of values yielded by the various tracking methods for some basins, e.g. 13 (as mentioned in the response to the other review).

Conclusions: I suggest adding a statement on how best to improve and continue ice sheet mass balance monitoring in this way, e.g., by adding to the time series of high-resolution Peninsula velocity fields, improving the flux-gate RES coverage, improving the SMB fields, continuing Landsat-like and Cryosat/ICESat-like datasets etc. – where do the biggest potential improvements lie? Emphasise the value of this study as the potential starting point for routine ongoing assessments, and the potential importance of this in diagnosing unstable behaviour.

Detailed comments: Line 182: 'mean mean' Line 272: 'See appendix A for the ...' Line 383 and onwards: Figure 7 instead of 6 etc. Line 431: 'Groundling' Line 440: '...Totten Glacier increased in ...' Line 445: '79% of the increase comes from glaciers...and another 11% comes from...' Line 509: 'that that' References: Fretwell et al repeated. Figure 5 caption: '...along-flux-gate...' Figure 9 caption: '...all 2015 image-pair displacements...' Line 819: '...assumed to be indicative of...'

---

## Referee Comment (RC3) · Anonymous Referee #3 · 24 Oct 2017

Summary The key results presented in this paper are measurements of Antarctic ice flow from Landsat 8, velocity change computed for a ∼6-year period between ∼2014 and ∼2008, and Antarctic mass balance calculated for ∼2014 using the input output method.

This paper is poorly written and uninteresting to read, largely due to the tedious methods section where two very similar feature tracking techniques are documented at great length. This could be of value if a robust inter-comparison between the two approaches was performed, or if the differences between the two results was analysed in detail, however, given that this work isn't done, there is really no scientific justification for presenting the two Landsat methods in this paper. Overall the paper would greatly benefit from a thorough re-write, which should mainly consist of condensing the methods

text, which is unnecessarily long and often repetitive. In addition to writing style, this manuscript must be edited to properly cite previous publications. I have noted throughout the methods and results sections that the authors have done a very cursory job of this, with many directly relevant papers not acknowledged in the text. Aside from reporting the new dataset, this paper doesn't deliver any novel science about the spatial pattern or magnitude of ice velocity changes in Antarctica, because regional case studies covering the present day time period have already been published in areas experiencing the largest change. This is however, is the first time Antarctic ice speed for the 2013-2015 period has been presented, along with ice sheet wide velocity change since 2008, so these results are novel. Again the discussion of these new results would be considerably improved if the continent wide signal was assessed in the context of previously published regional case studies.

Despite these major criticisms, which can only be addressed by significantly editing the existing manuscript, this paper does describe a new dataset that will likely be used by the scientific community.

Specific Edits L1 – Ice sheet instability and imbalance are not the same thing. The authors have shown that East Antarctica is not negatively out of balance during their study period, but their results don't prove stability. Replace this word in the title, and check use of the word 'stability' throughout the rest of the paper. L17 – New velocity map does not provide complete inland coverage of ice velocity as there is a data gap south of 82.4°. Edit wording in abstract to be factually correct. L20 – In the abstract, Marguerite Bay, West Antarctic Peninsula is flagged up as a key region with one of the most rapid velocity change, however the velocity change for this full region isn't visible in Figure 8. Edit fig 8 to show velocity change map in zoom for this region. L22 – Incorrect use of term stable. Edit throughout paper. L32 – Check sentence wording. Doesn't read smoothly. L36 – Mass change can be measured by multiple techniques with high precision and accuracy, e.g. gravimetry and altimetry in addition to velocity. Edit sentence to reflect that one technique, not all, require ice velocity to measure

mass change. L45 – Edit sentence to reflect that Landsat-8 measurements are only acquired during the summer. Use of the word annual implies that it is a true yearly average, when it is in fact just a summer mean so the speeds could be biased high. If the authors believe their measurements are representative of the annual mean, then evidence should be presented to support this. L67 – Attribution of author contribution to the paper should be listed in the acknowledgements, not the main body of a paper. L80 – The authors have clearly stated their adaptive window size used for velocity tracking. Add sentence to also state the step size. L84 – State the method used to correct the scale distortion, and provide some statistics evidencing that the error has been reduced. L86 – Is the variability of the ice speeds measured with all window sizes, less than the stated accuracy of the velocity measurements, (i.e. $\sim$10 m/yr)? L92 – State the threshold ice speed that was used to identify 'stable' (or rather stationary/slow flowing) ice surfaces. Were all areas classed as stationary used to improve the image co-registration, or was it a subset? If the later please edit the text to clarify rational for selecting ground control sites. Again the authors should also re-evaluate their use of the term 'stable'. L135 – The authors have used a shorter epoch for the raw data used as input to the NSIDC LISA processing technique. Why do this? If the purpose of the paper is to provide a present day assessment of Antarctic ice speeds and compare this with historical data, then only one processing technique is required. If alternatively, the authors aim to inter-compare multiple techniques to asses their respective merits, then the study period has to be the same for any meaningful inter-comparison to be performed. Either process data over the same time period or remove the poorer Landsat-8 method info and results. L145 – Again state the step size used. L145 – The authors used chip sizes ranging from 16 to 128 pixels in the JPL method, and 20 pixels in the NSIDC method. This will have a measurable impact on the output velocity measurements, as ice speed derived from larger window sizes will be biased lower than if a smaller window size was used on the same image pair. The authors should demonstrate how they have accounted for this. L150 – Quantify 'fairly strong', or amend writing style. L152 – Edit double full stop. L166 – State the maximum temporal baseline used for the image pairs. L170 – Justify why different post-processing methods have been applied to the output from the JPL and NSIDC velocity processing chains once the velocity measurements have been obtained. L177 – Provide the statistics for this interomparison with Rignot et al 2011 for all three surface types, (rock, zero flow, slow flow). L180 – The paragraph structure in this paper must be reorganised, it's completely arbitrary in its current form. For example, why have the authors introduced vr, vz and vl in the previous paragraph (which started off describing the NSIDC post processing method), and then discussed use of these variables in the following two short paragraphs? It's not great throughout the rest of the paper, but its particularly infuriating in the methods section as the paper would read much more clearly if each paragraph did a specific job, i.e. explained a distinct aspect of the work. L185 – The 750m output grid size for the NSIDC dataset is substantially larger than the 240 m resolution of the JPL dataset, and this will impact any intercomparison between the two. The authors must state how they have accounted for this. L192 – The authors have chosen flux gates based on some fairly straightforward rationale. A bit of time should be spent making the description more concise, as two sides of A4 is unnecessarily long. L262 – There are known issues with assuming that firn corrected elevation change rates are 100% dynamic (Zwally et al 2015, Wouters et al 2013), so this assumption is not valid. Moreover, not all ice flowing at >200m/yr is dynamic either, so the authors should edit the paper to state their rationale, and cite published literature that have demonstrated the complexity of this issue if there is no alternative to this assumption. L265 – What's the logic for choosing 0.1m/yr or 30%? These numbers seem arbitrary, so assuming there is some justification, edit the paper to state rationale. L275 – Relevant mass flux literature should be cited through the methods section. For example, Rignot et al, Mouginot et al 2014, Chuter et al 2017. The authors have not invented a new technique, so previous publications should be acknowledged in the text. L349 – Nilsson et al 2016 was not the first publication to apply the surface fit solver to Cryosat data, therefore the authors should edit text to cite previous publications where this technique was developed. Moreover, the Nilsson et al 2016 paper documents a method for esti-

mating altimetry mass change of Greenland, not Antarctica, where the firn processes and therefore processing challenges associated with it, are not the same, as shown by Nilsson et al 2015. The Antarctica method should be explained in full, or an appropriate citation should be provided. L353 – Edit the manuscript to explain how the authors have extrapolated elevation change at the ice sheet margins, where interpolation between two data points isn't possible. It's in this area that the highest rates of elevation change are located, therefore although the area is small, the numbers are significant, particularly given the way the authors are using this result in this paper. L374 – The authors have presented two separate Landsat datasets, JPL and NSIDC. Please choose a nomenclature and stick to it throughout the paper as readers do not know which dataset is referred to by 'Landsat' alone. This should be edited throughout the paper. L377 – Figure 8 in this paper shows that there is large spatial variability in the velocity change parameter, therefore its not correct to assume that velocity change at FG1 is the same as at FG2. The error associated with this assumption must be sensibly quantified, or better, don't use this unsatisfactory approach at all. L384 – This one sentence does not constitute a rigorous inter-comparison between the JPL and NSIDC datasets. Aside form the fact that the epoch covered by each datasets is not temporally contiguous, the authors provide no discussion about the respective merits of each method, the statistical differences between the two datasets, or geographical regions over which one method might out perform the other. It is immensely frustrating to have had to read through lengthy methods description of two marginally different techniques, only to have one dataset discarded with no apparent logical basis other than the personal preference of the authors. This paper should be edited to remove the description of one of the Landsat datasets, or, the authors should to a formal inter-comparison. L390 – The time period covered by the JPL dataset is only $\sim$ 1 year longer than the epoch covered by more recent data in Mouginot et al 2014. Edit paper to state how these results differ from Mouginot et al paper during the time period they overlap, not just the period where they don't. L398 – Edit paper to comment how these Getz results compare to the Chuter et al 2017 result, and cite the relevant paper. L405 – Edit paper

to comment how these Bellingshausen results compare to the Hogg et al 2017 result, and cite the relevant paper. L418 – The authors state that Scar Inlet Ice Shelf has sped up, however in the lengthy methods section of this paper, there has been no mention of how tidally induced velocity changes have been removed from the new dataset. The authors should remove this statement about the cause of Leppard and Flask Glacier velocity changes, or demonstrate quantitatively in this manuscript that tidally induced velocity change has been removed from both the Landsat 8 and historical SAR dataset. L437 – The spatial pattern of speedup on Law dome looks like its associated with the spatial distribution of image tracks. Can the authors demonstrate that this speedup is not just an artefact caused by a processing error? L440 – Edit increase'd' L715 – Fig 1. Change figure to show inland ice speed (in the 'Pole hole') in the Rignot et al 2011 full Antarctic velocity map, or explicitly state in the figure caption that this area has been masked out to fit the spatial extent of the new JPL ice velocity datasets. It is missrepresation of the Rignot et al 2011 dataset to imply that there is large a data gap in areas where one does not exist, particularly when the authors have actually used their velocity measurements from this region in their assessment. L715 – Add spatially variable error map for each velocity dataset shown in Figure 1. Input data density is interesting, but the error estimate has practical value. L740 – Edit figure caption to state more clearly which Landsat dataset corresponds to each color in the bar charts. L760 – The spatial pattern of change in ice speed on Pine Island Glacier, shown in Figure 8, isn't in agreement with change in speed presented elsewhere, and published in Mouginot et al (2014). The authors should discuss if the pattern, (specifically the two separate patches of high speedup), is a real signal, or if it is due to an error in one of the datasets?

---

## Author Comment (AC2) · 30 Oct 2017

**Response to Anonymous Referee #2**

Thank you kindly for taking the time to provide a thoughtful review of our manuscript.
Please find:
- Reviewer comments in back
- Responses in blue
- *Proposed changes to manuscript in italics*
* * *
This study is a useful benchmark contribution to Antarctic mass balance research in particular because it uses very large data archives to achieve high-resolution, near-comprehensive coverage and improved uncertainty reduction in ice flow measurements. In this way (and with updated SMB products and a new approach to flux gate comparisons), it compliments and improves upon earlier pioneering continent-scale flux studies, and allows recent flux changes to be calculated. As such, it marks a maturation in mass-balance auditing and points the way towards the regular, operational big-data measurement of Antarctic mass change.
Specific comments: Most of my queries have been covered in the authors' responses to the other review.

In title and throughout: I suggest avoiding using the term stability (or stable, re-stabilization) to mean unchanging flux because stability has other particular connotations for ice sheet mass balance.

The title of our manuscript was "Increased West Antarctic ice discharge and East Antarctic stability over the last seven years". We understand the reviewers concern with the usage of the term "stability", particularly that it could be taken out of context to infer that the mass balance of East Antarctic Ice Sheet will be resistant to future environmental change. We felt that we had provided sufficient context for correct interpretation of "stability" by indicating the quantity (discharge) and period of time (7 years) that "stability" is referring to. Given the reviewer's comments *we have changed the title to "Increased West Antarctic and unchanged East Antarctic ice discharge over the last seven years". We have also modified text in the main manuscript changing "stable" to "steady", "constant", and "unchanged".*

Abstract: I suggest rewording the final sentences, e.g. "The modest increase in ice discharge over the past 7 years but ongoing high rates of ice sheet mass loss and distinct patterns of elevation lowering suggest that the recent pattern of mass loss

in Antarctica is part of a longer-term phase of enhanced glacier flow initiated in the decades leading up to the first continent-wide radar mapping of ice flow."

*We will change the text to the following:*
*The West Antarctic ice sheet is experiencing high rates of mass loss and displays distinct patterns of elevation lowering that point to a dynamic imbalance. We find modest increase in ice discharge over the past 7 years that suggest that the recent pattern of mass loss in Antarctica is part of a longer-term phase of enhanced glacier flow initiated in the decades leading up to the first continent-wide radar mapping of ice flow.*

For the uncertainty associated with the assumption of surface velocity being equal to depth-averaged velocity ($\sigma F$ v-bar), the authors convincingly explain that this term is small, however it is a bias term of a particular sign which suggests that it should be corrected for or otherwise added to one side of the uncertainty range rather than being combined in quadrature.

*Good point. We have added this term as a bias to both sides of the error budget to retain symmetry… this has little impact on our total error budget.*

Section 3.1.2: line 405, replace 'certainty' with 'confidence'. Can the authors please be more specific in this section – do they consider the 56 Gt to be incorrect and the real imbalance to be close to zero?

This is a good question. We can say with confidence that any mass anomaly observed by satellite gravimeters or altimeters was not the consequence of a change in ice flow over our period of study. We also find that the net mass balance of Basins 23 and 24 to be negative, though the uncertainty is large (-27 +/- 24 Gt/yr.). An estimated loss of 56 G/yr. falls slightly outside of our uncertainty envelope.

*We have changed certainty to confidence and added a reference to a recent paper that examines longer-term changes in ice dynamics of this region:*
*This result agrees with a recent investigation of longer-term (1995-2016) changes in ice discharge for this region (Hogg et al., 2017). In that study they found that the region's glacier experienced an increase in ice discharge between 1995 and 2008 and almost no change in discharge between 2008 and 2016.*

Figure 4 caption: Please clarify the y-axis units.
*We have modified the caption to:*
*Histograms of ice equivalent thickness (a), uncertainty in ice equivalent thickness (b), year of ice thickness measurement (c), Firn Air Content (d), uncertainty in Firn Air*

*Content (e), surface velocity (f), change rate of ice equivalent thickness (g), and uncertainty in change rate of ice equivalent thickness (h) for GL0, FG1 and FG2 flux gates. The y-axis is the percentage of flux nodes that fall within each histogram bin.*

There are two Figure 6s. The second one (now Figure 7) needs a legend and also more discussion of the range of values yielded by the various tracking methods for some basins, e.g. 13 (as mentioned in the response to the other review).

*Thank you for catching this! In the revised manuscript we will add a legend to the new Figure 7 for the bar plots and we will expand our discussion of the flux change differences between results as per our response to Reviewer 1's point number 5.*

Conclusions: I suggest adding a statement on how best to improve and continue ice sheet mass balance monitoring in this way, e.g., by adding to the time series of high-resolution Peninsula velocity fields, improving the flux-gate RES coverage, improving the SMB fields, continuing Landsat-like and Cryosat/ICESat-like datasets etc. – where do the biggest potential improvements lie? Emphasise the value of this study as the potential starting point for routine ongoing assessments, and the potential importance of this in diagnosing unstable behaviour.

*Thank you for the suggestion.*

*We have added the following paragraph to the conclusions:*
*Glaciology is rapidly transitioning from an observationally constrained environment to one with ample high quality, high volume satellite datasets suitable for mapping ice flow on continental scales (e.g. Landsat 8, Sentinel 2a/b, Sentinel 1a/b). This study provides a foundation for continued assessment of ice sheet flow and discharge that will allow researches to observe both large and subtle changes ice sheet flow that may indicate early signs of ice sheet instability with low latency. Such a capability would help to diagnose unstable flow behavior and, in conjunction with high accuracy measurements of ice sheet elevation and mass change, would lead to improved assessment ice sheet surface mass balance and ice shelf melt rates. Low latency monitoring of ice flow and discharge would also allow field programs, flight planning and satellite tasking to coordinate the collection complimentary observations in areas of changing ice behavior. These advances will ultimately lead to a deeper understanding of the causal mechanisms resulting in observed and future ice sheet instabilities. Any substantial improvement in our assessment of ice sheet discharge will require more detailed knowledge of ice thickness just upstream of the grounding line, particularly for areas of complex flow such as the Antarctic Peninsula and Victoria Land. Errors in discharge estimates can be greatly reduced if thickness profiles are*

*acquired perpendicular to ice flow. Improved estimates of net mass change calculated using the mass budget approach will come from continued refinement of regional climate models and better estimates of basal melt.*

Detailed comments:
Line 182: 'mean mean'
Corrected.

Line 272: 'See appendix A for the ...'
Corrected.

Line 383 and onwards: Figure 7 instead of 6 etc.
Corrected.

Line 431: 'Groundling'
Corrected.

Line 440: '...Totten Glacier increased in ...'
Corrected.

Line 445: '79% of the increase comes from glaciers. . .and another 11% comes from. . .'
Corrected.

Line 509: 'that that'
Corrected.

References: Fretwell et al repeated.
Corrected.

Figure 5 caption: '. . .along-flux-gate. . .'
Changed.

Figure 9 caption: '. . .all 2015 image-pair displacements. . .'
Corrected.

Line 819: '. . .assumed to be indicative of. . .'
Corrected.

---

## Author Comment (AC3) · 30 Oct 2017

Thank you kindly for taking the time to provide a review of our manuscript.
Please find:
- Reviewer comments in back
- Responses in blue
- *Proposed changes to manuscript in italics*
* * *
This paper is poorly written and uninteresting to read, largely due to the tedious methods section where two very similar feature tracking techniques are documented at great length. This could be of value if a robust inter-comparison between the two approaches was performed, or if the differences between the two results was analysed in detail, however, given that this work isn't done, there is really no scientific justification for presenting the two Landsat methods in this paper. Overall the paper would greatly benefit from a thorough re-write, which should mainly consist of condensing the methods text, which is unnecessarily long and often repetitive. In addition to writing style, this manuscript must be edited to properly cite previous publications. I have noted through-out the methods and results sections that the authors have done a very cursory job of this, with many directly relevant papers not acknowledged in the text. Aside from reporting the new dataset, this paper doesn't deliver any novel science about the spatial pattern or magnitude of ice velocity changes in Antarctica, because regional case studies covering the present day time period have already been published in areas experiencing the largest change. This is however, is the first time Antarctic ice speed for the 2013-2015 period has been presented, along with ice sheet wide velocity change since 2008, so these results are novel. Again the discussion of these new results would be considerably improved if the continent wide signal was assessed in the context of previously published regional case studies.

We paraphrase the reviewer concerns as:
1. Lengthy methods section that describes separate methods for independent but similar velocity processing chains are described.
   *To address this concerns we have greatly reduced the length of the methodology text and removed redundancy. Most importantly we have added text clarifying that multiple velocity products were included to assess the sensitivity of the results to choice of processing methodology. We have also provided a more in-depth discussion of the differences between products and given justification for choosing to focus results to a single mapping.*

2. Properly cite previous publications
   *We have added additional citations where appropriate.*

Specific Edits L1 – Ice sheet instability and imbalance are not the same thing. The authors have shown that East Antarctica is not negatively out of balance during their study period, but their results don't prove stability. Replace this word in the title, and

check use of the word 'stability' throughout the rest of the paper.

The title of our manuscript was "Increased West Antarctic ice discharge and East Antarctic stability over the last seven years". We understand the reviewers concern with the usage of the term "stability", particularly that it could be taken out of context to infer that the mass balance of East Antarctic Ice Sheet will be resistant to future environmental change. We felt that we had provided sufficient context for correct interpretation of "stability" by indicating the quantity (discharge) and period of time (7 years) that "stability" is referring to. Given the reviewer's comment, and the earlier comments of reviewer 2, *we have changed the title to "Increased West Antarctic and unchanged East Antarctic ice discharge over the last seven years". We have also modified text in the main manuscript changing "stable" to "steady", "constant", and "unchanged".*

L17 – New velocity map does not provide complete inland coverage of ice velocity as there is a data gap south of 82.4 mass change. Edit wording in abstract to be factually correct.

Thank you for catching this.
*Text will be changed to: "inland coverage of ice velocity north of 82.4°S"*

L20 – In the abstract, Marguerite Bay, West Antarctic Peninsula is flagged up as a key region with one of the most rapid velocity change, however the velocity change for this full region isn't visible in Figure 8. Edit fig 8 to show velocity change map in zoom for this region.

The original Figure 8 did include a zoom of Marguerite Bay just above the frame showing the Amundsen Sea Sector. *We have made the Marguerite Bay frame larger in the revised figure.*

L22 –Incorrect use of term stable. Edit throughout paper.

*We have modified text in the main manuscript changing "stable" to "steady", "constant", and "unchanged".*

L32 – Check sentence wording. Doesn't read smoothly.

*Agreed. We've changed this to "Recent studies indicate significant mass loss from the Antarctic Ice Sheet that is likely accelerating"*

L36 – Mass change can be measured by multiple techniques with high precision and accuracy, e.g. gravimetry and altimetry in addition to velocity. Edit sentence to reflect that one technique, not all, require ice velocity to measure

While we agree that there are several measurement approaches for determining ice sheet mass change, this sentence specifically refers to the "difficulty in resolving continent-wide ice discharge". To the authors' knowledge, velocity is required to directly measure ice discharge separate from other sources of mass change.

L45 – Edit sentence to reflect that Landsat-8 measurements are only acquired during the summer. Use of the word annual implies that it is a true yearly average, when it is in fact just a summer mean so the speeds could be biased high. If the authors believe their measurements are representative of the annual mean, then evidence should be presented to support this.

*Good point, will change "annual" to "yearly".* For some regions imagery can be matched with 1 year of separation making it a truly annual measurement of ice velocity. Time of acquisition may bias estimates of ice flow but unlike Greenland it is unclear if there is a significant seasonal cycle in Antarctic ice discharge.

L67 – Attribution of author contribution to the paper should be listed in the acknowledgements, not the main body of a paper.
*Thanks for catching this. This will be removed.*

L80 – The authors have clearly stated their adaptive window size used for velocity tracking. Add sentence to also state the step size.
*We have clarified this in the following sentence:*
*Results from the sparse search guide a dense search with search centers spaced such that there is no overlap between adjacent template search chips (i.e. distance between template centers is equal to the template size).*

L84 – State the method used to correct the scale distortion, and provide some statistics evidencing that the error has been reduced.
Scale distortion is an artifact of all map-projected data and is simply an artifact of projecting a warped surface onto a plane. Corrections are derived from the projection equations.

*We now include a reference to projection equations:*
*We corrected for this scale distortion when converting from pixel displacement to velocity following the equations presented in Snyder, 1987*

L86 – Is the variability of the ice speeds measured with all window sizes, less than the stated accuracy of the velocity measurements, (i.e.10 m/yr)?
The stated uncertainty average of 10 m/yr. is for the merged product only and is more dependent on number of cloud free image acquisition and persistence of surface features than on the selected chip size.

L92 –State the threshold ice speed that was used to identify 'stable' (or rather stationary/slow flowing) ice surfaces. Were all areas classed as stationary used to improve the image co-registration, or was it a subset? If the later please edit the text to clarify rational for selecting ground control sites. Again the authors should also re-evaluate their use of the term 'stable'.

All data was used to generate the "reference velocity" to which the annual mosaics were registered. *We have clarified this sentence as follows:*

*This was done by stacking all time-normalized displacements (velocities), co-registering them over stationary/slow flowing surfaces with low variability as described in the next section…*

L135 – The authors have used a shorter epoch for the raw data used as input to the NSIDC LISA processing technique. Why do this? If the purpose of the paper is to provide a present day assessment of Antarctic ice speeds and compare this with historical data, then only one processing technique is required. If alternatively, the authors aim to inter-compare multiple techniques to asses their respective merits, then the study period has to be the same for any meaningful inter-comparison to be performed. Either process data over the same time period or remove the poorer Landsat-8 method info and results. The aim here was simply to demonstrate the robustness of the discharge results. Using different epochs, different processing methodologies (though similar), different chip sizes, different sample spacing and different resolutions has minimal impact on the conclusion of the manuscript (see original Figure 6). *We have added text clarifying the purpose of including multiple velocity products and greatly shortened the methods section.*

L145 – Again state the step size used.
*Now included.*

L145 – The authors used chip sizes ranging from 16 to 128 pixels in the JPL method, and 20 pixels in the NSIDC method. This will have a measurable impact on the output velocity measurements, as ice speed derived from larger window sizes will be biased lower than if a smaller window size was used on the same image pair. The authors should demonstrate how they have accounted for this. The goal here was not to homogenize datasets and approached but rather to demonstrate the robustness of the discharge estimates using two independent approaches to measuring velocity. Using different chip sizes does not seem to have any impact on the discharge results (see original Figure 6).

L150 – Quantify 'fairly strong', or amend writing style.
*This paragraph has been re-written.*

L152 – Edit double full stop.
*Fixed, thanks.*

L166 – State the maximum temporal baseline used for the image pairs.
Now provided.

L170 – Justify why different post-processing methods have been applied to the output from the JPL and NSIDC velocity processing chains once the velocity measurements have been obtained.
Please see response to L135 comment.

L177 – Provide the statistics for this interomparison with Rignot et al 2011 for all three surface types, (rock, zero flow, slow flow).
This paragraph describes the offset corrections applied to the PyCorr velocity fields. The Rignot et al 2011 data is used to determine the magnitude of the correction, not to assess the quality or differences between products. This paragraph was largely redundant so has now been merged with the preceding paragraph.

L180 – The paragraph structure in this paper must be reorganised, it's completely arbitrary in its current form. For example, why have the authors introduced vr, vz and vl in the previous paragraph (which started off describing the NSIDC post processing method), and then discussed use of these variables in the following two short paragraphs? It's not great throughout the rest of the paper, but its particularly infuriating in the methods section as the paper would read much more clearly if each paragraph did a specific job, i.e. explained a distinct aspect of the work.
*This section has been re-written.*

L185 – The 750m output grid size for the NSIDC dataset is substantially larger than the 240 m resolution of the JPL dataset, and this will impact any intercomparison between the two. The authors must state how they have accounted for this.
Using different grid resolution was simply to demonstrate the impact of differing resolutions on discharge estimates. The impact was minor as shown in the original Figure 6. Please see response to L135.

L192 – The authors have chosen flux gates based on some fairly straightforward rationale. A bit of time should be spent making the description more concise, as two sides of A4 is unnecessarily long.
*We have shortened this section in the revised manuscript*

L262 – There are known issues with assuming that firn corrected elevation change rates are 100% dynamic (Zwally et al 2015, Wouters et al 2013), so this assumption is not valid. Moreover, not all ice flowing at >200m/yr is dynamic either, so the authors should edit the paper to state their rationale, and cite published literature that have demonstrated the complexity of this issue if there is no alternative to this assumption.
We have added the following for increased transparency:
*A velocity cut off 200 m yr$^{-1}$ was selected to separate volume changes resulting from changes surface mass balance and those resulting from changes in dynamics. This threshold is arbitrary. Even so, the dynamic volume change correction is very small and largely insensitive to the selected cut off velocity.*

L265 – What's the logic for choosing 0.1m/yr or 30%? These numbers seem arbitrary, so assuming there is some justification, edit the paper to state rationale.
We have added the following for increased transparency:
*Uncertainty in the dynamic volume change can not be rigorously quantified and are therefore conservatively assumed to be 0.1 m yr$^{-1}$ times the area between the grounding line and the flux gate having a surface velocity >200 m yr$^{-1}$ or 30% of the magnitude of the estimated dynamic volume change, whichever is larger.*

L275 – Relevant mass flux literature should be cited through the methods section. For example, Rignot et al, Mouginot et al 2014, Chuter et al 2017. The authors have not invented a new technique, so previous publications should be acknowledged in the text. *Additional references added.*

L349 – Nilsson et al 2016 was not the first publication to apply the surface fit solver to Cryosat data, therefore the authors should edit text to cite previous publications where this technique was developed. Moreover, the Nilsson et al 2016 paper documents a method for estimating altimetry mass change of Greenland, not Antarctica, where the firn processes and therefore processing challenges associated with it, are not the same, as shownby Nilsson et al 2015. The Antarctica method should be explained in full, or an appropriate citation should be provided.
We agree with the reviewer that Nilsson et al was not the first to apply surface fits to CryoSat-2 data. There are however many approaches to applying surface fits. More importantly Nilsson et al. (2016) describe the full process chain used to go from the ESA L1b waveform data to the JPL L2 elevations and elevation changes. For this reason we feel that Nilsson et al. (2016) is the most relevant citation. Citations to other approaches of extracting elevation changes from CS2 data can be found in Nilsson et al. (2016).

L353 – Edit the manuscript to explain how the authors have extrapolated elevation change at the ice sheet margins, where interpolation between two data points isn't possible. It's in this area that the highest rates of elevation change are located, therefore although the area is small, the numbers are significant, particularly given the way the authors are using this result in this paper.
We are not sure we fully understand the reviewer's request. This paragraph states that "The edited data was then interpolated onto a 1 km grid using the weighted average of the 16 closest grid points, weighted by their standard error from the least squares solution and distance.)

L374 – The authors have presented two separate Landsat datasets, JPL and NSIDC. Please choose a nomenclature and stick to it throughout the paper as readers do not know which dataset is referred to by 'Landsat' alone. This should be edited throughout the paper.
*This has been corrected throughout.*

L377 – Figure 8 in this paper shows that there is large spatial variability in the velocity change parameter, therefore its not correct to assume that velocity change at FG1 is the same as at FG2. The error associated with this assumption must be sensibly quantified, or better, don't use this unsatisfactory approach at all.
We measure flux change at the FG1 grounding line and add this to the total flux estimated at FG2. This approach reduces errors by ~50%. We have added clarifying language to the methods section to better justify our approach.

L384 – This one sentence does not constitute a rigorous inter-comparison between the JPL and NSIDC datasets. Aside form the fact that the epoch covered by each datasets is

not temporally contiguous, the authors provide no discussion about the respective merits of each method, the statistical differences between the two datasets, or geographical regions over which one method might out perform the other. It is immensely frustrating to have had to read through lengthy methods description of two marginally different techniques, only to have one dataset discarded with no apparent logical basis other than the personal preference of the authors. This paper should be edited to remove the description of one of the Landsat datasets, or, the authors should to a formal inter-comparison.
Please see response to general comment 1 and to L135

L390 – The time period covered by the JPL dataset is only 1 year longer than the epoch covered by more recent data in Mouginot et al 2014. Edit paper to state how these results differ from Mouginot et al paper during the time period they overlap, not just the period where they don't.
Thank you for pointing this out. I am not sure why we had omitted this in the submitted paper. *We have changed the text accordingly:*
*This implies an average discharge increase of 2.4 Gt yr$^{-2}$ for 2008-2015 that is considerably lower than the 6.5 Gt yr$^{-2}$ previously estimated for 1994-2008 (Mouginot et al., 2014). This recent slowing in the rate of acceleration is in excellent agreement with the previously published temporally dense history of ice discharge that gave a rate of discharge increase for this region of 2.3 Gt yr$^{-2}$ for overlapping but shorter period of 2010-2013 period (Mouginot et al., 2014).*

L398 – Edit paper to comment how these Getz results compare to the Chuter et al 2017 result, and cite the relevant paper.
Thank you for suggesting this. *We have included the following:*
*This result is in broad agreement with Chuter et al. (2017) that observed increases in ice velocity during the 2007-2013 period alongside 2010-2013 dynamic thinning rates of 0.7 m yr$^{-1}$ for the glaciers feeding the Abbot and Getz ice shelves.*

L405 – Edit paper to comment how these Bellingshausen results compare to the Hogg et al 2017 result, and cite the relevant paper.
Thank you for the suggestion. *We have added the following sentence:*
*This result agrees with a recent investigation of longer-term (1995-2016) changes in ice discharge for this region (Hogg et al., 2017) that found that the region's glacier experienced an increase in ice discharge between 1995 and 2008 and almost no change in discharge between 2008 and 2016.*

L418 – The authors state that Scar Inlet Ice Shelf has sped up, however in the lengthy methods section of this paper, there has been no mention of how tidally induced velocity changes have been removed from the new dataset. The authors should remove this statement about the cause of Leppard and Flask Glacier velocity changes, or demonstrate quantitatively in this manuscript that tidally induced velocity change has been removed from both the Landsat 8 and historical SAR dataset.

*We have removed the reference to our data and instead rely on the citation to Khazendar et al. 2015.*

L437 – The spatial pattern of speedup on Law dome looks like its associated with the spatial distribution of image tracks. Can the authors demonstrate that this speedup is not just an artefact caused by a processing error?
Point well taken. The radar tracks are clearly visible in the Figure 8 inset image for this region. Underwood and Bond glaciers acceleration (~40 m/yr. in places) signals exceed the radar errors that are like due to residual ionosphere effects (~20 m/yr.). *Even so we have added cautionary language to account for the increased error in this region: "The region to the west of Law Dome, including Underwood and Bond glaciers, shows evidence of some increased flow speed and ice discharge, though the signal is near the limit of detection."*

L440 – Edit increase'd' L715 –
Changed

Fig 1. Change figure to show inland ice speed (in the 'Pole hole') in the Rignot et al 2011 full Antarctic velocity map, or explicitly state in the figure caption that this area has been masked out to fit the spatial extent of the new JPL ice velocity datasets. It is missrepresation of the Rignot et al 2011 dataset to imply that there is large a data gap in areas where one does not exist, particularly when the authors have actually used their velocity measurements from this region in their assessment.
Agreed. This was a mistake on our part. Figure has been corrected.

L715 – Add spatially variable error map for each velocity dataset shown in Figure 1. Input data density is interesting, but the error estimate has practical value.
*We now include both count and error.*

L740 – Edit figure caption to state more clearly which Landsat dataset corresponds to each color in the bar charts.
Thank you for catching this. *We have added a legend for the bar plots.*

L760 – The spatial pattern of change in ice speed on Pine Island Glacier, shown in Figure 8, isn't in agreement with change in speed presented elsewhere, and published in Mouginot et al (2014). The authors should discuss if the pattern, (specifically the two separate patches of high speedup), is a real signal, or if it is due to an error in one of the datasets?
The reviewer makes a very keen observation. Our map of velocity change shows an area of peak velocity change at 50 km upstream of the grounding line and a secondary peak at 110 km from the gl. We see no such peak when comparing between Landsat products, which makes us confident that the secondary peak is not an artifact of the Landsat processing. One possible non-geophysical explanation is that the radar mosaic includes data from a period significantly earlier than 2008 for this area. *We have include a mention of this in the revised manuscript.*

---

## Author Response (AR1)

[revised manuscript text omitted]
. Velocities are further filtered For the remaining unmasked velocities, weby examininged the difference between 185 the velocities  $\gamma\gamma\gamma$  at the assessed pixel with the eight surrounding values speed values in the speed output grid ( $\gamma\gamma\gamma$ ). These surrounding grid locations may or may not have a velocity value (given our noise filtering, above). Velocities with If a grid location had no output velocity neighbors were, it was masked (i.e. not included in the final composite). Velocities with If only one of the eight neighbor neighbor locations populated, we were -masked the center pixel if the absolute difference between the two prevalues was greater than 365 m/yr1-m/day. Velocities with With two neighbors neighbours were masked if 190 they exceeded, we required that the center vv value be within three standard deviations of the mean of its neighbors, or it was Finally Once the above masking operation was complete, the standard deviation of each 3x3 region was masked.

computed, and the center pixel of each region was masked if the corresponding standard deviation is greater than  $\frac{365 \text{ m/yr}}{1000 \text{ m/day}}$ .

195 Image-pair In compositing the masked velocity, we also sought to adjust false offsets of ice motion due to geolocation errors in the image pairs. This is particularly problematic for closely-spaced pairs (16 and 32 day separations), which in fact were the bulk of the image pairings in our compilation. To correct the effects of imprecise ggeolocation errors were corrected using , we computed three sets of x-y velocity offsets and a corresponding percent coverage associated with that velocity image and that correction algorithm. Each set of offsets and coverages arewere computed using one of three mask images: a "over rock (www.add.scar.org)k" mask (*vrmsk*) and, a "near-zero ice velocity(<20 m yr-1), " mask (*vzmsk*) and a "low ice velocity (<40 and >20 m yr-1) areas" mask (*vlmsk*) ac.-cording to Rignot et al. (2011aThe 
[revised manuscript text omitted]

---

## Author Response (AR2)

**Response to Anonymous Referee #3 – Second Response**

Thank you kindly for this second review

**Please find:**

- Reviewer comments in back
- Responses in blue
- Proposed changes to manuscript in italics

\_\_\_\_\_

Review of: Increased West Antarctic ice discharge and East Antarctic stability over the last seven years. Gardner et al., 2017

Summary

The authors have significantly revised this manuscript and incorporated most of the original reviewer requests. This effort is appreciated and the manuscript is very much improved because of it. However, there are some outstanding points where I do not feel that the authors response is satisfactory. I recommend minor changes to the manuscript to address these few remaining items.

**Specific Edits**

L36 – Mass change can be measured by multiple techniques with high precision and accuracy, e.g. gravimetry and altimetry in addition to velocity. Edit sentence to reflect that one technique, not all, require ice velocity to measure mass change. While we agree that there are several measurement approaches for determining ice sheet mass change, this sentence specifically refers to the "difficulty in resolving continent- wide ice discharge". To the authors' knowledge, velocity is required to directly measure ice discharge separate from other sources of mass change.

The original comment still stands because although the authors response says their statement refers to mass discharge, the sentence in the paper is in not discharge specific. Its trivial to edit the paper to clarify this.

**We have modified this sentence to read:**

A major hurdle for improved attribution of mass changes determined from gravimetry and/or altimetry, and in determining mass changes themselves from the mass balance approach, is the difficulty in resolving continent-wide changes in ice discharge at high precision and accuracy for multiple epochs.

L349 – Nilsson et al 2016 was not the first publication to apply the surface fit solver to Cryosat data, therefore the authors should edit text to cite previous publications where this technique was developed. Moreover, the Nilsson et al 2016 paper documents a method for estimating altimetry mass change of Greenland, not Antarctica, where the firn processes and therefore processing challenges associated with it, are not the same, as shown by Nilsson et al 2015. The Antarctica method should be explained in full, or an appropriate citation should be provided.

We agree with the reviewer that Nilsson et al was not the first to apply surface fits to CryoSat-2 data. There are however many approaches to applying surface fits. More importantly Nilsson et al. (2016) describe the full process chain used to go from the ESA L1b waveform data to the JPL L2 elevations and elevation changes. For this reason we feel that Nilsson et al. (2016) is the most relevant citation. Citations to other approaches of extracting elevation changes from CS2 data can be found in Nilsson et al. (2016).

Fine that the lower level processing is documented in the Nilsson et al 2016 paper, but the method used to go from elevation change to mass change from altimetry in a dry snow Antarctic environment is not documented at all in this paper. A citation must be provided for this step of the methodology, or if the authors haven't followed a previously published method, their technique for this part of the processing chain must be documented in full. The methods used for radar altimetry in Greenland have very different challenges from Antarctica, so it is not good enough to just cite a Greenland methods paper here.

We apply the methods as described in the Nilsson et al 2016 paper. We understand the reviewers concern that it is more challenging to extract elevation changes for low-density glacier surfaces where changes in volume scattering can contaminate the elevation change signals derived from radar altimetry. This is especially true with conventional radar altimeters and in the interior of the ice sheets. This is not as much of a concern in our application since we are only using elevation change result for the periphery of the ice sheet (between GL0 and FG1) where CryoSat-2 operates in SARIn mode, change signals are larger and volume scattering effects are reduced. In addition, our retracking approach is much less sensitive to changes in volume scattering than other waveform retrackers (See Figure 2 in Nilsson et al., 2016) and is similar to the methodology used by *Helm et al.*, 2014 to process CryoSat-2 data over both the Antarctic and Greenland Ice Sheets. It should be noted that Helm et al., 2014 did not apply any ice sheet specific changes to their methodology nor any empirical correction for changes in waveform shape. In addition our results and conclusions are insensitive to centimeter scale errors in the estimated elevation and volume changes. For these reasons we argue that we have provided sufficient documentation of our elevation change analysis.

L353 – Edit the manuscript to explain how the authors have extrapolated elevation change at the ice sheet margins, where interpolation between two data points isn't possible. It's in this area that the highest rates of elevation change are located, therefore although the area is small, the numbers are significant, particularly given the way the authors are using this result in this paper.

We are not sure we fully understand the reviewer's request. This paragraph states that "The edited data was then interpolated onto a 1 km grid using the weighted average of the 16 closest grid points, weighted by their standard error from the least squares solution and distance.) In regions of imbalance, the rate of elevation change typically increases from its maximum level at the grounding line, to decreasing magnitude inland. At the ice sheet edge, if the 16 closest pixels are inland, as is often the case because steeply sloping topography at the grounding line, filling the margin with the mean of the inland data points will lead to an underestimation of the real thinning rate at the grounding line. Consequently, interpolating across data gaps where the 16 closest pixels are distributed all around the gap, should ideally be handled in a different way where the data gap exists to one side of the 16 closest measurements. The authors should account for this in their method to avoid a marginal underestimation of the elevation change rates.

The reviewer is concerned that steep gradients in elevation changes near the grounding line are not being properly resolved with the distance weighted interpolation of the 16 nearest points. This is a classic problem of interpolation and is not easily satisfied, as changes in elevation are not always well correlated with ancillary information that might be used to improve the interpolation (e.g. mean velocity as used by *Hurkmans et al.*, 2012). Here we have chosen to use a distance + error weighted approach using the 16 closest neighbors which we feel is a reasonable approach that does not introduce significant uncertainty into our analysis as the distribution of SARIn elevations are relatively dense compared to earlier conventional altimetry that had more difficulty in high-slope areas. While imperfect, we feel that our approach is sufficient for our purposes and can likely be improved upon in future investigations. As can be seen in Table A1 the dynamic volume change correction for the entire Antarctic is 16 Gt/vr or less than 1% of total discharge so any further improvement in the estimate of dynamic volume change will have minimal impact on our results. To provide the reviewer with a better sense of what the elevation change patterns look like we have included a figure here showing a close up of the area with the majority of the dynamic volume change (Amundsen Sea Sector):

Figure 1: Zoom of Amundsen Sea Sector elevation changes derived from CryoSat-2 data. The integrated area between *GL0* and *FG2* having ice velocities > 200 m/yr were used to compute the dynamic volume change correction.

L437 – The spatial pattern of speedup on Law dome looks like its associated with the spatial distribution of image tracks. Can the authors demonstrate that this speedup is not just an artefact caused by a processing error?

Point well taken. The radar tracks are clearly visible in the Figure 8 inset image for this region. Underwood and Bond glaciers acceleration (~40 m/yr. in places) signals exceed the radar errors that are like due to residual ionosphere effects (~20 m/yr.). Even so we have added cautionary language to account for the increased error in this region: "The region to the west of Law Dome, including Underwood and Bond glaciers,

**shows evidence of some increased flow speed and ice discharge, though the signal is near the limit of detection."**

This isn't sufficient. The processing artefact isn't in the error estimate, so it wouldn't be possible for users of the data to filter bad data out based on the available quality information. Along with this, the magnitude of this velocity error is the same, and in some cases larger, than velocity change elsewhere that the reader is expected to trust is real. So the author's statement that the signal is at the limit of detection is wrong, unless other 'real' signal is below the limit of detection. The authors should remove known errors in their dataset, or transparently state in the paper that their data shouldn't be trusted in this region.

The stripes in the velocity differencing are largely a result of artifacts in the previously published radar velocity mosaic of Rignot et al. 2011. This dataset is provided with an error estimate but "these estimates should be used more as an indication of relative quality rather than absolute error". Maximum errors are all